# Gradient Descent with Large Step Sizes: Chaos and Fractal Convergence Region

**Shuang Liang**
UCLA
liangshuang@g.ucla.edu

**Guido Montúfar**
UCLA & MPI MiS
montufar@math.ucla.edu

## Abstract

We examine gradient descent in matrix factorization and show that under large step sizes the parameter space develops a fractal structure. We derive the exact critical step size for convergence in scalar-vector factorization and show that near criticality the selected minimizer depends sensitively on the initialization. Moreover, we show that adding regularization amplifies this sensitivity, generating a fractal boundary between initializations that converge and those that diverge. The analysis extends to general matrix factorization with orthogonal initialization. Our findings reveal that near-critical step sizes induce a chaotic regime of gradient descent where the training outcome is unpredictable and there are no simple implicit biases, such as towards balancedness, minimum norm, or flatness.

## 1 Introduction

Understanding the properties of gradient descent in non-convex overparametrized optimization has been a central pursuit in modern machine learning. The step size, or learning rate, is a critical factor determining the dynamics and convergence of gradient descent optimization. In particular, it has a major influence on the returned solution and its generalization performance (Nar and Sastry, 2018; Jastrzebski et al., 2020; Lewkowycz et al., 2020; Cohen et al., 2021). Large step sizes have been associated with flat and balanced minimizers of the training objective (Wu et al., 2018; Wang et al., 2022; Menon, 2025), sparse feature representations (Nacson et al., 2022; Andriushchenko et al., 2023), smooth solution functions (Mulayoff and Michaeli, 2020; Nacson et al., 2023), and improved generalization (Ba et al., 2022; Qiao et al., 2024; Sadrtdinov et al., 2024). Yet, the theoretical understanding of large step sizes remains limited, even in simple convex settings. Our investigation is motivated by two fundamental questions:

*Given an initial parameter, what is the critical (largest) step size that allows convergence?*

*What kind of implicit biases are induced by gradient descent with near-critical step size?*

Addressing these questions is challenging, since large step sizes can produce highly complex, non-monotonic, and even chaotic trajectories. In particular, trajectories may not converge to stationary points but instead enter periodic or chaotic oscillations (Chen and Bruna, 2023; Chen et al., 2024b; Ghosh et al., 2025), or converge to a statistical distribution (Kong and Tao, 2020); trajectories that eventually converge to a minimizer may still undergo chaotic oscillations during early training (Zhu et al., 2023; Kreisler et al., 2023; Song and Yun, 2023); and trajectories with nearby initializations can diverge exponentially from one another (Herrmann et al., 2022; Jiménez-González et al., 2025). Moreover, empirically, the set of step sizes and the set of initializations leading to convergence can form fractal structures (see, respectively, Sohl-Dickstein, 2024; Zhu et al., 2023). In this work, we provide precise answers to the above questions in the context of matrix factorization problems with rigorous theoretical characterizations.

We begin by examining gradient descent in a simplified problem to factor a scalar target as the inner product of two vectors. We show that two striking phenomena emerge at near-critical step sizes: (i) initializations in arbitrarily small sets can converge to global minimizers with arbitrarily large norm, sharpness or imbalance, or to a saddle, indicating a lack of simple implicit biases; and (ii) the set of initializations that converge, that is, the convergence region, has a fractal structure, indicating that the convergence is unpredictable near the boundary (see Figure 1). Interestingly, while

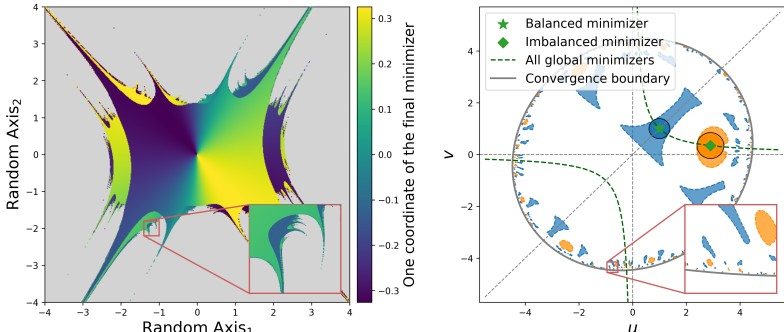

Figure 1: The training outcome of gradient descent depends sensitively on initializations near the convergence boundary. Left: Gradient descent applied to $L(u, v) = (u^\top v - 1)^2 + 0.3(\|u\|_2^2 + \|v\|_2^2)$ with $(u, v) \in \mathbb{R}^{10}$. Shown is a random two-dimensional slice of $\mathbb{R}^{10}$. Gray points are initializations leading to divergence; other points are colored by a coordinate value of the converged minimizer. The convergence boundary has a fractal structure. Right: Gradient descent applied to $L(u, v) = (uv - 1)^2$ with $(u, v) \in \mathbb{R}^2$. Shown are a balanced minimizer $p$ (green star), its neighborhood $O_p$ (blue disk), and the preimage of $O_p$ under $\text{GD}^6$ (blue region with dashed boundary). An imbalanced minimizer (green diamond) is shown similarly, with its neighborhood and preimages depicted in orange. The convergence boundary is smooth but the converged point is sensitive to initialization.

in the unregularized setting the convergence region is regular (in the almost everywhere sense), it becomes fractal once regularization is introduced. Further quantifying the unpredictability of the training outcome, we show that the topological entropy of the gradient descent system is at least $\log 3$. Also, we show that the fractal nature of the boundary of the convergence region is captured by a self-similar curve whose fractal dimension is estimated as $1.249$. To our knowledge, beyond the univariate training loss setting studied by Kong and Tao (2020); Chen et al. (2024b), this is the first rigorous characterization of chaos in gradient descent optimization.

We then extend our analysis to general matrix factorization by showing that, when the initialization lies in a subspace defined by a set of orthogonal conditions, the gradient descent dynamics decouples into several independent scalar factorization dynamics. Hence, all results established for scalar factorization remain valid on this subspace. This includes the commonly used identity initialization, as well as the training of linear residual networks (Hardt and Ma, 2017; Bartlett et al., 2018).

We further analyze the mechanisms underlying these phenomena and trace them to a folding behavior of the update map $\text{GD}(\theta) = \theta - \eta \nabla L(\theta)$: the map GD sends a region $\mathcal{C}$ onto a superset of $\mathcal{C}$ in a multi-fold covering manner. Consequently, if $\mathcal{C}$ contains a convergence boundary that is invariant under GD, then the boundary exhibits self-similarity; moreover, GD is mixing on the boundary, giving rise to the sensitivity to initialization. We show that for general neural networks with polynomial activations GD indeed acts as a covering map on the parameter space outside a measure-zero set.

Overall, our results imply that near-critical step sizes place gradient descent in a chaotic regime where the training outcome is unpredictable: infinitesimal perturbations on the initial conditions can lead to substantially different outcomes. This contrasts sharply with the stable dynamics observed at smaller step sizes. We empirically validate the presence of chaos in matrix factorization with general initializations, deep matrix factorization, and deep ReLU networks trained on real-world datasets.

## 1.1 Main Contributions

The goal of this article is to provide rigorous insights into the dynamics of gradient descent with large step sizes in matrix factorization. Our contributions can be summarized as follows:

- We derive the exact critical step size for convergence in scalar factorization and show that the convergence region is equal almost everywhere to a bounded and smooth domain. At critical step sizes, we show that infinitesimal perturbations to the initialization can lead to global minimizers with arbitrarily large norm, or to a saddle point. Moreover, for initializations randomly sampled from arbitrarily small sets, the distribution of the sharpness at the converged minimizers has a

support containing all possible values below $2/\eta$, which evidences chaos at a distributional level. Also, we show that the topological entropy of gradient descent system is at least $\log 3$.

- We show that when using $\ell_2$ regularization, gradient descent selects either the minimal distance solution or the maximal distance solution among all global minimizers. At critical step size, infinitesimal perturbations of the initialization can switch this selection from one to the other. Furthermore, adding $\ell_2$ regularization yields a fractal convergence boundary whose geometry is captured by a self-similar shape in $\mathbb{R}^2$ after symmetry reduction.

- We extend these results to general matrix factorization with initializations in a subspace that includes the identity initialization. In particular, gradient descent exhibits chaos and fractal convergence boundary on this subspace.

- We reason that chaos arises from a folding behavior of the gradient descent update map. For general neural networks with polynomial activations, we show that gradient descent is a covering map on each connected component of the parameter space after removing a measure-zero set. We empirically validate these chaotic phenomena in deep ReLU networks trained on real-world data.

## 1.2 RELATED WORK

**Gradient Descent Dynamics Under Large Step Sizes** A main line of research on large-step-size gradient descent focuses on the non-monotonic convergence of the loss and its impact on the final model. Key perspectives include the *Edge of Stability* (Cohen et al., 2021; Ma et al., 2022; Agarwala et al., 2023; Damian et al., 2023; Ahn et al., 2022; 2023; Zhu et al., 2023; Wang et al., 2023) and the catapult phenomenon (Lewkowycz et al., 2020; Kalra and Barkeshli, 2023; Meltzer and Liu, 2023; Zhu et al., 2024a;b). Compared to these works, our analysis extends to even larger (near-critical) step sizes. Another line of work shows how a large step size can enhance feature learning in one step of gradient descent (Ba et al., 2022; Dandi et al., 2024; Moniri et al., 2025), also comparing different parametrizations (Sonthalia et al., 2025). Similar observations about the role of the step size in feature learning have also been made for SGD (Andriushchenko et al., 2023; Lu et al., 2024) and pre-training (Sadrtdinov et al., 2024). Ziyin et al. (2022) observed that for a certain range of step sizes, SGD can have undesirable behavior, such as convergence to local maxima. For linear networks, Kreisler et al. (2023) identified a monotonically decreasing quantity (sharpness) along the gradient descent trajectories. Wang et al. (2022) showed large step size induces an implicit bias towards balanced minimizers in matrix factorization. Crăciun and Ghoshdastidar (2024) proved the existence of a step size threshold above which the algorithm diverges. A similar problem is studied by Marion and Chizat (2024). Large-step-size gradient descent has also been investigated in logistic regression (Wu and Su, 2023; Wu et al., 2024; Meng et al., 2024), and some of the analysis has been further extended to shallow networks (Cai et al., 2024).

**Chaos in Optimization** Van Den Doel and Ascher (2012) empirically observed chaos, specifically positive finite-time Lyapunov exponents, for several variants of steepest descent methods. The phenomenon named *period-doubling bifurcation route to chaos* has been widely observed in recent literature (Kong and Tao, 2020; Chen and Bruna, 2023; Chen et al., 2024b; Meng et al., 2024; Danovski et al., 2024; Ghosh et al., 2025). Among them, only Kong and Tao (2020); Chen et al. (2024b) provided rigorous analyses for the chaotic dynamics. They showed the emergence of *Li-Yorke chaos*, i.e., the existence of periodic orbits of arbitrary periods, for univariate training losses. In comparison, our setting is high-dimensional. Additionally, we establish not only the existence of all periodic orbits, but also the sensitivity of the limiting point to initialization, which is more relevant to practical optimization, particularly the implicit bias of the optimization algorithm.

## 2 PRELIMINARIES

We focus on the following shallow matrix factorization problem with $\ell_2$ regularization:

$$\min_{\theta=(U,V)} L(\theta) = \frac{1}{2} \left\| U^\top V - Y \right\|_F^2 + \frac{\lambda}{2} (\|U\|_F^2 + \|V\|_F^2), \tag{1}$$

where $\lambda \geq 0$, $U, V \in \mathbb{R}^{d \times d_y}$ and the target matrix $Y \in \mathbb{R}^{d_y \times d_y}$ is a diagonal matrix. The diagonality of $Y$ is a weak assumption that can be achieved by reparametrization. Specifically, for arbitrary $Y$ consider the singular value decomposition $Y = P_Y \Sigma_Y Q_Y^\top$ and the rotations $U = \tilde{U} P_Y^\top$ and

$V = \tilde{V} Q_Y^\top$. The objective then becomes $\tilde{L}(\tilde{U}, \tilde{V}) = \frac{1}{2}\|\tilde{U}^\top \tilde{V} - \Sigma_Y\|_F^2 + \frac{\lambda}{2}(\|\tilde{U}\|_F^2 + \|\tilde{V}\|_F^2)$, where the target is diagonal. Moreover, the optimization dynamics in minimizing $\tilde{L}(\tilde{U}, \tilde{V})$ are identical to those in minimizing $L(U, V)$ up to the rotation (see details in Appendix C).

Gradient optimization in problem (1) has been extensively studied, especially in small-step-size regimes (see, e.g., Saxe et al., 2014; Arora et al., 2019; Yun et al., 2021; Du et al., 2018; Li et al., 2021; Min et al., 2023; Chen et al., 2024a). Its landscape enjoys a favorable structure: the global minimum is attained and every stationary point is either a global minimizer or a strict saddle (Li et al., 2019b; Valavi et al., 2020; Zhou et al., 2022). Nevertheless, this problem retains a key complexity typical of neural network optimization: global minimizers can differ substantially (although they yield the same end-to-end matrix). In particular, in the unregularized case, both the parameter norm $\|U\|_F^2 + \|V\|_F^2$ and the imbalance $\|UU^\top - VV^\top\|_F^2$ can be arbitrarily large on the set of minimizers. This makes the problem a natural testbed for studying how hyperparameter choices affect the implicit biases of parameter optimization algorithms. We note that other forms of regularization have also been studied in the literature, such as $\|UU^\top - VV^\top\|_F^2$ (Tu et al., 2016; Ge et al., 2017).

We consider gradient descent with constant step size $\eta$ to solve problem (1):
$$U_{t+1} = U_t - \eta V_t(V_t^\top U_t - Y^\top) - \eta\lambda U_t, \quad V_{t+1} = V_t - \eta U_t(U_t^\top V_t - Y) - \eta\lambda V_t.$$

We define the gradient descent update map $\mathrm{GD}_\eta(\theta) = \theta - \eta\nabla L(\theta)$ so that $(U_{t+1}, V_{t+1}) = \mathrm{GD}_\eta(U_t, V_t)$. The *basin of attraction* of a stationary point $\theta^*$ of $L$ is the set of all initializations that converge to $\theta^*$, $\{\theta\colon \lim_{N\to\infty} \mathrm{GD}_\eta^N(\theta) = \theta^*\}$.[1] The *convergence region* for step size $\eta$, $\mathcal{D}_\eta$, is the union of the basins of attraction of all global minimizers. The *critical step size* $\eta^*(\bar{U}, \bar{V})$ for an initialization $(\bar{U}, \bar{V})$ is defined as $\eta^*(\bar{U}, \bar{V}) = \sup\{\eta\colon \lim_{N\to\infty} \mathrm{GD}_\eta^N(\theta) \in \mathcal{M}\}$, i.e., the supremum of the step sizes that allow convergence to a global minimizer, where $\mathcal{M} = \{\theta\colon L(\theta) = \min_{\theta'} L(\theta')\}$ denotes the set of all global minimizers. A set $S$ is said to be invariant under $\mathrm{GD}_\eta$ if $\mathrm{GD}_\eta(S) \subset S$.

We introduce notions for describing fractal geometry. A fractal is typically defined as a shape that exhibits self-similarity and fine structure at arbitrarily small scales. Formally, we say a set $S \subset \mathbb{R}^n$ is *self-similar with degree $k$* if there exist $k$ homeomorphisms, $\phi_i\colon S \to S, i = 1, \cdots, k$, that satisfy (i) $S = \cup_{i=1}^k \phi_i(S)$ and (ii) there exists an open set $O \subset S$ such that $\cup_{i=1}^k \phi_i(O) \subset O$ and $(\phi_i(O))_{i=1}^k$ are pairwise disjoint. Condition (i) states that $S$ can be covered by $k$ smaller copies of itself. Condition (ii), which is known as the open set condition, ensures that those copies do not overlap much. This definition is closely related to an Iterated Function System (IFS), a standard tool for analyzing fractals (see, e.g., Hutchinson, 1981; Falconer, 2013). However, unlike IFS where the maps are required to be contractive and the set $S$ to be compact, the shapes considered in our study may be unbounded.

Finally, we introduce notions related to chaos. Although there is no universal definition of chaos, one common characterization of chaos is the sensitivity to initialization, which is often known as the *butterfly effect*. In the context of optimization, this means that infinitesimal perturbations of the initial condition can lead to substantially different training outcomes (e.g., turning convergence into divergence, or shifting convergence from one minimizer to another qualitatively different minimizer). In dynamical systems, a key measure of chaos is the *topological entropy*. Informally, the topological entropy $h(F)$ of a dynamical system $F$ measures the exponential growth rate of the number of distinct trajectories of $F$ as a function of the trajectory length. We defer the formal definition to Appendix B.1. A positive topological entropy is widely regarded as a hallmark of chaos (see, e.g., Katok et al., 1995; Robinson, 1998; Vries, 2014). In this paper, we adopt the above notions for fractals and chaos. We note that different definitions and settings exist, and discuss their relation to our study in Appendix A.

## 3  SIMPLIFIED MATRIX FACTORIZATION

In this section, we study gradient descent in the special case of problem (1) where $d_y = 1$, i.e., factorizing a scalar $y$ as the inner product of two vectors as $u^\top v$. This and similar scalar factorization settings have served as canonical models for understanding large-step-size dynamics (Lewkowycz et al., 2020; Wang et al., 2022; Kreisler et al., 2023; Ahn et al., 2023; Zhu et al., 2023; Kalra et al., 2025). Compared to these, our analysis extends to the full spectrum of step sizes, rather than restricting to bounded step sizes. Proofs for results in this section are in Appendix E.

---

[1] In dynamical systems the basin of attraction is often defined for attractors. Here we extend the terminology to include all stationary points, such as saddles, for simplicity of presentation.

## 3.1 CHAOS AT LARGE STEP SIZE

Consider the unregularized scalar factorization problem:

$$\min_{\theta=(u,v)\in\mathbb{R}^{2d}} L(\theta) = \frac{1}{2}(u^\top v - y)^2, \tag{2}$$

where $u, v \in \mathbb{R}^d$, $d \geq 1$ and $y \in \mathbb{R}$. Problem (2) retains several key complexities of the general problem (1), including non-convexity, high-dimensionality, non-Lipschitz gradients, and an unbounded set of global minimizers $\mathcal{M} = \{\theta\colon u^\top v = y\}$.

In the following result, we characterize the critical step size for problem (2) and the emergence of chaos under critical step sizes. We use $B(\theta, \varepsilon)$ to denote the ball of radius $\varepsilon$ centered at $\theta$.

**Theorem 1.** *Consider gradient descent with step size $\eta$ for solving problem* (2). *The following holds:*

- ***Critical Step Size:** For almost all initializations $(\bar{u}, \bar{v}) \in \mathbb{R}^{2d}$, the algorithm converges to a global minimizer if $\eta < \eta^*(\bar{u}, \bar{v})$ and fails to converge to any minimizer if $\eta > \eta^*(\bar{u}, \bar{v})$, where the critical step size is given by (when $y = 0$, we adopt the convention $1/0 = +\infty$):*

$$\eta^*(\bar{u}, \bar{v}) = \min\left\{ \frac{1}{|y|}, \frac{8}{\|\bar{u}\|_2^2 + \|\bar{v}\|_2^2 + \sqrt{(\|\bar{u}\|_2^2 + \|\bar{v}\|_2^2)^2 - 16y(\bar{u}^\top\bar{v} - y)}} \right\}. \tag{3}$$

  *Therefore, when $\eta$ satisfies $\eta|y| < 1$, the convergence region $\mathcal{D}_\eta$ is equal almost everywhere to $\mathcal{D}'_\eta = \left\{ (u, v) \in \mathbb{R}^{2d}\colon \|u\|_2^2 + \|v\|_2^2 + \sqrt{(\|u\|_2^2 + \|v\|_2^2)^2 - 16y(u^\top v - y)} < \frac{8}{\eta} \right\}.$*

- ***Sensitivity to Initialization:** Fix a step size $\eta$ that satisfies $\eta|y| < 1$. Let $\gamma_{\min} = \min\{\|\theta\|\colon \theta \in \mathcal{M}\}$ be the minimal norm over all global minimizers. Given arbitrary $\theta \in \partial\mathcal{D}'_\eta$, $\varepsilon, K > 0$ and $\gamma \in [\gamma_{\min}, \infty)$, there exist $\theta', \theta'', \theta''' \in B(\theta, \varepsilon)$ such that, as $N$ tends to infinity, $\mathrm{GD}_\eta^N(\theta')$ converges to a global minimizer with norm $\gamma$, $\mathrm{GD}_\eta^N(\theta'')$ converges to a global minimizer with $\|uu^\top - vv^\top\|_F > K$, and $\mathrm{GD}_\eta^N(\theta''')$ converges to $(\mathbf{0}, \mathbf{0})$, which is a saddle when $y \neq 0$.*

- ***Trajectory Complexity:** Assume $\eta|y| < 1$. The topological entropy of the gradient descent system $\mathrm{GD}_\eta$ satisfies $h(\mathrm{GD}_\eta) \geq \log 3$. Moreover, $\mathrm{GD}_\eta$ has periodic orbits of any positive integer period.*

Theorem 1 provides a necessary and sufficient convergence condition for gradient descent in problem (2). The critical step size (3) consists of two terms. The first term arises because: when $\eta|y| > 1$, all global minimizers become unstable and only attract a measure-zero set. The second term characterizes the full convergence region, which is equal almost everywhere to an ellipsoid $\mathcal{D}'_\eta$ (see right panel of Figure 1). Note that this convergence result is global: it holds for all step sizes $\eta$ and initializations $\theta_0$, except for a null set in the $(\eta, \theta_0)$-space. In contrast, the previous work of Wang et al. (2022) only showed convergence for step sizes in the range $\eta < 1/(3|y|)$, and initializations in a strict subset of $\mathcal{D}'_\eta$ (see Figure 7). A comparison between the proof techniques used for Theorem 1 and those used by Wang et al. (2022) is provided in Appendix E.1.2.

By Theorem 1, gradient descent in problem (2) exhibits a strong form of sensitivity to initial condition. Note that in problem (2), the squared norm of the parameter coincides with the loss sharpness $\lambda_{\max}(\nabla^2 L)$ at global minimizers (see Appendix E.1.2). Hence, Theorem 1 shows that at critical step size, infinitesimal perturbations of the initialization can send the trajectory to a minimizer with arbitrarily large norm, sharpness or imbalance, or to a saddle. This is a hallmark of unpredictability: it is impossible to reduce the error in the prediction of the converging point by improving the precision in the specification of the initialization. We remark that this form of reachability from an arbitrarily small range of initial values is familiar in chaos theory, such as the Julia sets in complex systems and the Wada basin boundaries (see, e.g., Devaney and Eckmann, 1987; Nusse and Yorke, 1996; Aguirre et al., 2001). However, these classical frameworks depend on properties not satisfied by our setting, such as complex differentiability or invertibility of the system map, and thus do not apply here.

Theorem 1 also quantitatively measures chaos in gradient descent in problem (2): the topological entropy is positive and is at least $\log 3$. Roughly speaking, the number of distinct gradient descent trajectories of length $N$ grows at a rate of $3^N$ (further interpretation for topological entropy is provided

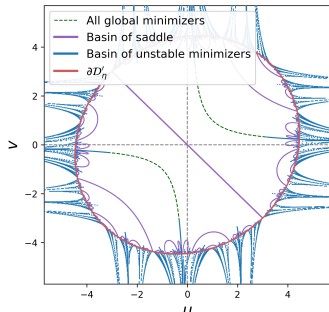 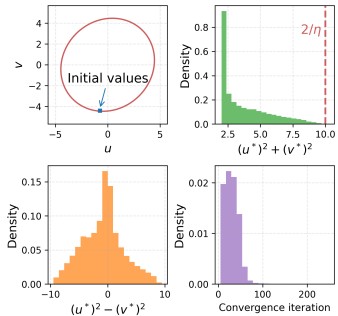

Figure 2: Gradient descent applied to $L(u, v) = (uv - 1)^2$ with $(u, v) \in \mathbb{R}^2$. Left: Blue lines and purple lines represent the basins of attraction of unstable minimizers and of the saddle $(\mathbf{0}, \mathbf{0})$, respectively. Right: Initializations are evenly sampled from a small set intersecting $\partial \mathcal{D}'_\eta$ (the blue square). However, the distributions of the squared norm and imbalance of the converged minimizer $(u^*, v^*)$, and of the number of iterations to reach a loss below $10^{-8}$, have wide supports.

in Appendix B.1). Beyond entropy, another aspect of trajectory complexity is shown: gradient descent admits periodic orbits of any positive integer period. This is closely related to Li-Yorke chaos and was shown for gradient descent in univariate loss functions (Kong and Tao, 2020; Chen et al., 2024b).

While Theorem 1 identifies the convergence region up to a measure-zero set, we now provide a complete description of the region. In Appendix E.1.1, we show that every minimizer with squared norm larger than $2/\eta$ is Lyapunov-unstable and that the basins of attraction of unstable minimizers and of the saddle have measure zero. By Theorem 1, these measure-zero basins intersect $\partial \mathcal{D}'_\eta$ in arbitrarily small neighborhoods and exhibit fractal structure (see left panel of Figure 2). The full convergence region $\mathcal{D}_\eta$ is then the union of the smooth domain $\mathcal{D}'_\eta$ and the basins of unstable minimizers, excluding the basin of the saddle $(\mathbf{0}, \mathbf{0})$ when $y \neq 0$.

The next result shows that gradient descent in problem (2) exhibits chaos even after removing the measure-zero basins of attraction of unstable minimizers, i.e., it is chaotic at the distributional level.

**Theorem 2.** *Under the same conditions and notations as Theorem 1 and assuming $\eta|y| < 1$, given arbitrary $\theta \in \partial \mathcal{D}'_\eta, \varepsilon > 0$ and open sub-interval $E \subset [\gamma_{\min}, 2/\eta]$, there exists a set of positive measure $O \subset B(\theta, \varepsilon)$ such that for any $\theta' \in O$, $\mathrm{GD}_\eta^N(\theta')$ converges to a solution with norm in $E$.*

Consider initializations randomly sampled from a distribution whose support contains an *arbitrarily small* neighborhood of the convergence boundary. Theorem 2 then implies that the sharpness (or norm) of the converged minimizer must have a support containing the *entire* interval $(\gamma_{\min}, 2/\eta)$ (see right panel of Figure 2). In other words, at near-critical step size and even after discarding the basins of unstable minimizers, the best prediction for the final sharpness is the *entire* interval $(\gamma_{\min}, 2/\eta)$.

## 3.2 REGULARIZATION INDUCES FRACTAL CONVERGENCE BOUNDARY

Consider the scalar factorization problem with $\ell_2$ regularization:

$$\min_{\theta=(u,v)\in\mathbb{R}^{2d}} L(\theta) = \frac{1}{2}(u^\top v - y)^2 + \frac{\lambda}{2}(\|u\|_2^2 + \|v\|_2^2), \tag{4}$$

where $u, v \in \mathbb{R}^d$, $d \geq 1$, $\lambda \geq 0$ and $y \in \mathbb{R}$. The added regularization makes the set of global minimizers a bounded set. In particular, for problem (4), $\mathcal{M} = \{u = \mathrm{sgn}(y)v, \|u\|_2^2 = |y| - \lambda\}$ when $\lambda < |y|$ and $\mathcal{M} = \{(\mathbf{0}, \mathbf{0})\}$ when $\lambda \geq |y|$. Regularization is commonly used to mitigate unbounded minimizers and to establish convergence results (Cabral et al., 2013; Ge et al., 2017; Li et al., 2019a). However, and rather remarkably, we will show that for the regularized problem the global dynamics of gradient descent becomes even more unpredictable than for the unregularized problem: not only is the limiting point of convergent trajectories unpredictable but also the convergence itself.

The predictability of convergence depends on the geometry of the boundary of the convergence region. Two difficulties arise in analyzing this geometry: (i) the presence of the basin of the saddle; and (ii) the high-dimensionality of the convergence boundary. Specifically, we observe that the basin

of attraction of the saddle point intricately penetrates $\mathcal{D}_\eta$ and creates topological boundaries "within" $\mathcal{D}_\eta$ (see Figure 8). However, such boundaries are not of interest, since they do not separate points inside $\mathcal{D}_\eta$ from points outside, i.e., both sides lie in $\mathcal{D}_\eta$. This motivates us to instead consider the boundary of $\mathcal{D}_\eta'' = \mathcal{D}_\eta \cup \mathcal{S}_\eta$, where $\mathcal{S}_\eta$ is the basin of the saddle. Note, the smooth domain $\mathcal{D}_\eta'$ in the unregularized problem plays an analogous role in clarifying the geometry of convergence region.

The second difficulty is the high dimensionality of the boundary $\partial \mathcal{D}_\eta'' \subset \mathbb{R}^{2d}$. To address this, we identify and reduce the symmetry in $\partial \mathcal{D}_\eta''$. We introduce the map $T \colon \mathbb{R}^{2d} \to \mathbb{R}^2$, $T(u, v) = (u^\top v, \|u\|_2^2 + \|v\|_2^2)$. The fiber of $T$, i.e., the preimage of a point in $T(\mathbb{R}^{2d})$, is generically a manifold diffeomorphic to $\mathbb{S}^{d-1} \times \mathbb{S}^{d-1}$ and hence has a regular shape (see Appendix D). In the following, we show that gradient descent dynamics are captured by their evolution across these fibers.

**Proposition 3.** *Let $(u_t, v_t)_{t \geq 0}$ denote the gradient descent trajectory in problem* (4) *with $\lambda \geq 0$. Let $(z_t, w_t) = T(u_t, v_t)$. There exists a planar map $F \colon \mathbb{R}^2 \to \mathbb{R}^2$ that only depends on $\eta, \lambda, y$ such that $(z_{t+1}, w_{t+1}) = F(z_t, w_t)$ holds for all $t \geq 0$. In particular, $(u_t, v_t)$ converges to $\mathcal{M}$ if and only if $(z_t, w_t)$ converges to $T(\mathcal{M})$, and it converges to $(\mathbf{0}, \mathbf{0})$ if and only if $(z_t, w_t)$ converges to $(0, 0)$.*

The map $F$ describes how the gradient descent trajectory evolves across the fibers of $T$. Its formulation is given in Appendix D. By Proposition 3, all points lying in the same fiber share the same convergence behavior. Therefore, roughly, the boundary $\partial \mathcal{D}_\eta''$ can be constructed by attaching fibers of $T$ to the projected boundary $T(\partial \mathcal{D}_\eta'')$; an example will be given below. Note, as the fibers are generically smooth manifolds, any geometric complexity of $\partial \mathcal{D}_\eta''$ will be captured by $T(\partial D_\eta'')$.

In the following result, we show that the convergence boundary of problem (4) has a self-similar structure and gradient descent exhibits sensitivity to initialization near the convergence boundary.

**Theorem 4.** *Consider gradient descent with step size $\eta$ for problem* (4) *with $0 < \lambda \leq \min\{(1/\eta) - |y|, 1/(2\eta)\}$. Consider the map $T(u, v) = (u^\top v, \|u\|_2^2 + \|v\|_2^2)$. Let $\mathcal{S}_\eta$ be the basin of attraction of $(\mathbf{0}, \mathbf{0})$, which is a saddle if $\lambda < |y|$, and let $\mathcal{D}_\eta'' = \mathcal{D}_\eta \cup \mathcal{S}_\eta$. The following holds:*

- *Self-similarity: $\mathcal{S}_\eta$ has measure zero and $T(\partial \mathcal{D}_\eta'')$ is self-similar with degree three.*

- *Unboundedness: When $y = 0$, there exist constants $a, b > 0$ such that almost all initializations $(\bar{u}, \bar{v})$ with $|\bar{u}^\top \bar{v}| < a \exp(-b(\|\bar{u}\|_2^2 + \|\bar{v}\|_2^2))$ converge to a global minimizer.*

- *Sensitivity to Initialization: For any $\theta \in \mathcal{D}_\eta$, the algorithm converges either to the closest global minimizer $p^-(\theta) = \arg\min_{p \in \mathcal{M}} \|p - \theta\|^2$, or the farthest $p^+(\theta) = \arg\max_{p \in \mathcal{M}} \|p - \theta\|^2$.[2] Moreover, there exist infinitely many points on $\partial \mathcal{D}_\eta''$ such that for any open set $O$ containing such a point, there exist $\theta', \theta'' \in O$ such that $\mathrm{GD}_\eta^N(\theta')$ converges to $p^-(\theta')$ and $\mathrm{GD}_\eta^N(\theta'')$ converges to $p^+(\theta'')$, as $N$ tends to infinity.*

Theorem 4 indicates that the convergence boundary for problem (4), $\partial \mathcal{D}_\eta''$, can be constructed by attaching fibers of $T$ to a self-similar set $T(\partial \mathcal{D}_\eta'')$ with degree three. $T(\partial \mathcal{D}_\eta'')$ is displayed in the left panel of Figure 3, and its box-counting dimension is estimated to be $1.249$, as shown in the right panel. The case of $d = 1$, i.e., $u, v \in \mathbb{R}$, is shown in the middle panel of Figure 3. There, the fibers of $T$ generically consist of four discrete points. $\partial \mathcal{D}_\eta''$ is then constructed by attaching four points to $T(\partial \mathcal{D}_\eta'')$. Hence, $\partial \mathcal{D}_\eta''$ is simply the union of four copies of $T(\partial \mathcal{D}_\eta'')$. The fractal boundary marks unpredictability in convergence: given an initial point near the boundary, it is almost impossible to determine whether the point is inside or outside the convergence region. This unpredictability is also quantified by the box-counting dimension, as explained in Appendix B.2.

Theorem 4 shows that when $y = 0$, the convergence region has an unbounded interior up to a null set. This sharply contrasts with the convergence region in the unregularized case, which coincides almost everywhere with a bounded domain. By Theorem 4, gradient descent converges provided that the $u^\top v$ decays exponentially fast as a function of the squared norm $\|u\|_2^2 + \|v\|_2^2$. Geometrically, this creates an outward spike in the convergence region. Then, the self-similarity replicates this spike infinitely many times and at multiple scales, giving rise to the spiky convergence boundary observed in Figure 3. Although Theorem 4 shows the unboundedness only for the case $y = 0$, we observe qualitatively the same geometry for general targets (for an example, see left panel in Figure 1).

---

[2]When $\theta \in \mathcal{D}_\eta$, both $\min_{p \in \mathcal{M}} \|p - \theta\|^2$ and $\max_{p \in \mathcal{M}} \|p - \theta\|^2$ admit unique solutions. Moreover, $p^+(\theta) \neq p^-(\theta)$ when $\lambda < |y|$.

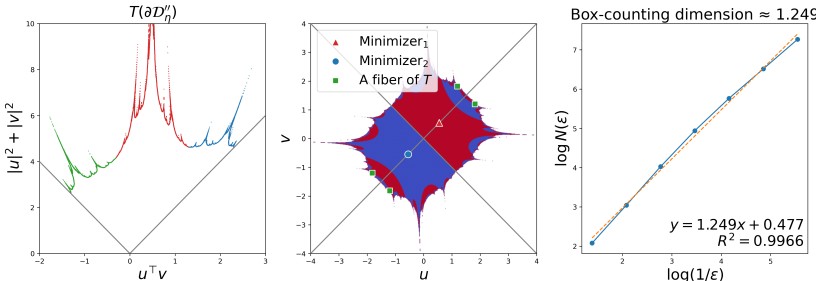

Figure 3: Gradient descent is applied to $L(u, v) = (uv - 0.5)^2/2 + 0.1(u^2 + v^2)$, where $(u, v) \in \mathbb{R}^2$. Left: The projected convergence boundary $T(\partial\mathcal{D}''_\eta)$ is self-similar with degree three: it is covered by three smaller copies of itself (green, red, blue). Middle: The convergence boundary $\partial\mathcal{D}''_\eta$ consists of four replicates of $T(\partial\mathcal{D}''_\eta)$, separated by gray lines. The only two minimizers are shown as a red triangle and a blue circle. Points are colored red if they converge to the red triangle, and blue if they converge to the blue circle. Right: The box-counting dimension of $T(\partial\mathcal{D}''_\eta)$ is estimated as $1.249$.

Fractal basin boundaries have been extensively studied in dynamical systems (see, e.g., Grebogi et al., 1983b; McDonald et al., 1985). However, these classical approaches are either largely case-specific or rely on properties that do not hold in our settings, for instance, invertibility of the system map. A more detailed discussion is in Appendix A. To our knowledge, our result provides the first rigorous characterization of a fractal convergence region in the context of machine learning optimization.

Theorem 4 also shows that, although regularization eliminates unbounded global minimizers, the selected minimizer remains unpredictable. Specifically, while the algorithm always selects either the minimal distance solution or the maximal distance solution, this selection becomes unpredictable near the convergence boundary, as both choices occur in arbitrarily small sets (see middle panel of Figure 3). This stands in contrast with gradient descent under small step sizes, which typically exhibits a distance-minimization bias (see, e.g., Gunasekar et al., 2018; Boursier et al., 2022). Indeed, in the following, we show that with sufficiently small step size, gradient descent in problem (4) always selects the minimal distance solution.

**Theorem 5.** *Under the same conditions and notations as Theorem 4 and letting $Q(u, v) = \|u\|_2^2 + \|v\|_2^2 + \sqrt{(\|u\|_2^2 + \|v\|_2^2)^2 - 16y(u^\top v - y)}$, the following holds for almost all initializations $(\bar{u}, \bar{v})$: If $\eta < 8/(4\lambda + Q(\bar{u}, \bar{v}))$, then gradient descent converges to a global minimizer; If $\eta < 4/(4\lambda + Q(\bar{u}, \bar{v}))$, then the particular minimizer it converges to is $p^-(\bar{u}, \bar{v})$.*

Finally, we present another implication of the chaos in gradient descent. In Appendix F, we show for the case $d = 1$, i.e., $L(u, v) = \frac{1}{2}(uv - y)^2 + \frac{\lambda}{2}(u^2 + v^2)$, $u, v \in \mathbb{R}$, $\lambda \geq 0$, that, any continuous *dynamical invariant* must be constant. In particular, the *imbalance* $u^2 - v^2$, which is known to be (approximately) preserved under gradient descent with small step sizes (Du et al., 2018; Arora et al., 2019), fails dramatically under large step sizes. Although this result does not directly extend to the case $d \geq 2$, we anticipate that the chaos strongly constrains the form of dynamical invariants.

## 4 MATRIX FACTORIZATION AND BEYOND

In this section, we first extend results in Section 3 to general matrix factorization with orthogonal initializations. We then analyze the underlying mechanism that gives rise to the chaotic phenomena, and discuss to what extent this mechanism may extend to more general settings. Finally, we present experiments showing chaos in a real-world machine learning setting.

### 4.1 MATRIX FACTORIZATION WITH ORTHOGONAL INITIALIZATIONS

Consider gradient descent in matrix factorization (1) with initializations in the following subspace:

$$\mathcal{W} = \left\{ (U, V) \in \mathbb{R}^{2d \cdot d_y} : \langle u^i, u^j \rangle = \langle u^i, v^j \rangle = \langle v^i, v^j \rangle = 0, \ \forall i \neq j \right\}, \tag{5}$$

where $u^i, v^i$ denote the $i$th column of $U, V$. The subspace $\mathcal{W}$ includes several commonly studied initialization schemes, such as the scaled identity initializations $\bar{U} = \alpha I_d, \bar{V} = \beta I_d$ (Chou et al.,

2024; Ghosh et al., 2025), zero-asymmetric initialization (Wu et al., 2019), and those used in training linear residual networks (Hardt and Ma, 2017; Bartlett et al., 2018). Note that in the low-rank setting $d < d_y$, constraints in (5) require at least $d_y - d$ pairs of $(u^i, v^i)$ to be initialized to zero.

A key observation is that results of Section 3 precisely characterize the gradient descent dynamics on $\mathcal{W}$: with initialization in $\mathcal{W}$, the trajectory remains in $\mathcal{W}$, and the dynamics decouple column-wise: each pair $(u^i, v^i)$ evolves independently according to the scalar factorization dynamics with target $y_i$, the $i$th diagonal entry of $Y$. Thus, all results in Section 3 extend verbatim, and gradient descent exhibits chaos and a fractal convergence boundary on $\mathcal{W}$ (see details in Appendix G).

A full characterization of the dynamics outside $\mathcal{W}$ requires additional investigations, for instance, whether $\mathcal{W}$ attracts or repels nearby trajectories. However, we point out that, the presence of chaos on $\mathcal{W}$ already suggests that the global dynamics can be unpredictable. For instance, due to the continuity of $\mathrm{GD}_\eta$, we expect that initializations in the vicinity of the convergence boundary in $\mathcal{W}$, will inherit the sensitivity to initial conditions, at least during the initial phase before potentially escaping $\mathcal{W}$. This phenomenon is known as *transient chaos* in dynamical systems (see, e.g., Tél, 1990). Experimentally, we observe that the chaotic phenomena indeed persist for general initializations, and also in *deep* matrix factorization (Appendix I). A rigorous characterization of gradient descent dynamics in these settings is an intriguing direction that we leave for future work.

### 4.2 General Mechanism Behind Chaos

We now explain the mechanism giving rise to chaos in scalar factorization. For simplicity, consider the case $d = 1$: $L(u, v) = (uv - y)^2/2 + \lambda(u^2 + v^2)/2$ with $(u, v) \in \mathbb{R}^2$. In this setting, there exists a set $\mathcal{C} \subset \mathbb{R}^2$ such that $\mathrm{GD}_\eta$ behaves as a 3-covering map from $\mathcal{C}$ onto $\mathrm{GD}_\eta(\mathcal{C}) \supset \mathcal{C}$. Roughly speaking, $\mathrm{GD}_\eta$ stretches and folds $\mathcal{C}$ to cover $\mathrm{GD}_\eta(\mathcal{C})$ three times. Furthermore, $\mathcal{C}$ contains the convergence boundary $\partial \mathcal{D}_\eta$. The boundary satisfies $\mathrm{GD}_\eta(\partial \mathcal{D}_\eta) = \partial \mathcal{D}_\eta$, meaning that $\partial \mathcal{D}_\eta$ can be stretched and folded three times to cover *itself*, thereby exhibiting self-similarity. Additionally, due to its folding behavior, $\mathrm{GD}_\eta$ is *transitive* on $\partial \mathcal{D}_\eta$, i.e., points on the boundary $\partial \mathcal{D}_\eta$ are mixed under the iterations of $\mathrm{GD}_\eta$ (via multiple stretches and folds). This mixing leads to the sensitivity to initializations near the boundary. A visualization of these properties are provided in Figure 6.

The key ingredient above is the existence of a set $\mathcal{C}$ in the parameter space on which $\mathrm{GD}_\eta$ acts as a covering map. In fact, the described chaotic phenomena arise given such a set $\mathcal{C}$ and any invariant subset $\mathcal{A} \subset \mathcal{C}$ that satisfies $\mathrm{GD}_\eta(\mathcal{A}) = \mathcal{A}$, a condition typically satisfied by the boundary of a basin of attraction. Then we ask: *When does such a set $\mathcal{C}$ exist?* The following result partially addresses this for general neural networks with polynomial activation functions.

**Proposition 6.** *Let $f_\theta(\cdot)$ be a polynomial neural network of arbitrary depth and width, parametrized by $\theta \in \mathbb{R}^p$. Consider a loss function $L(\theta) = \sum_{j=1}^m \ell(f_\theta(x_j), y_j)$ with polynomial loss $\ell$ and training data $(x_j, y_j)_{j=1}^m$. For all $\eta > 0$ except for at most finitely many values, there exists a measure-zero set $\mathcal{K}_\eta \subset \mathbb{R}^p$ such that $\mathrm{GD}_\eta$ is a covering map on any connected component of $\mathbb{R}^p \setminus \mathcal{K}_\eta$.*

The proof is provided in Appendix G. We remark that, while Proposition 6 suggests a general folding behavior of $\mathrm{GD}_\eta$, it does not address whether $\mathrm{GD}_\eta(\mathcal{C})$ contains $\mathcal{C}$ and whether $\mathcal{C}$ contains a basin boundary. We leave these intriguing questions for future research.

### 4.3 Chaos and Fractals in Neural Networks

We show that the chaotic phenomena persist in real-world training settings. We trained a depth-three ReLU network on a 2-class subset of CIFAR-10 (Krizhevsky et al., 2009). In Figure 4 we consider the mean squared error and report how the step size affects the norm and sharpness of the parameter that is returned at the end of training. For gradient descent without momentum, two distinct regimes of step sizes can be observed: (i) EoS regime: both the final norm and sharpness lie close to a smooth curve, indicating predictability. In particular, the final sharpness is close to $2/\eta$, aligning with Cohen et al. (2021). (ii) Chaotic regime: the norm and sharpness become sensitive to the step size, indicating chaos and unpredictability. In particular, the final sharpness spans almost all values below $2/\eta$, aligning with our Theorem 2. A similar phenomenon appears for gradient descent with Polyak momentum (Polyak, 1964). However, in the small-step-size regime, the final sharpness forms a cluster rather than aligning with $(2 + 2\beta)/\eta$ as predicted by Cohen et al. (2021). Experiments with cross-entropy loss are provided in Appendix J, where a similar two-regime phenomenon is observed.

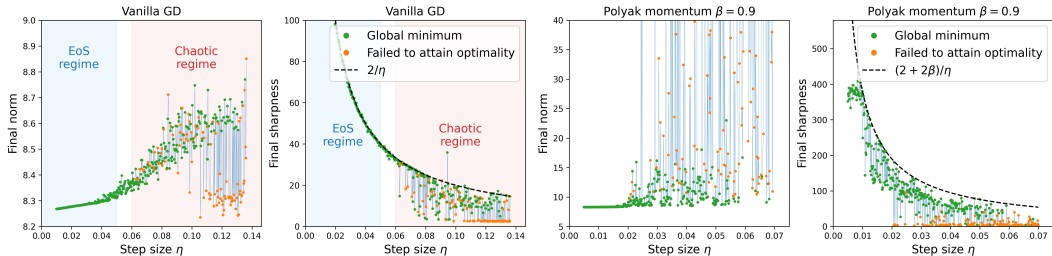

Figure 4: GD without and with momentum in training a depth-3 ReLU network on a subset of CIFAR-10 for 5000 iterations under mean squared error. The initialization is randomly sampled once and then kept fixed across all panels. At large step sizes, the final norm and final sharpness are sensitive to step size. Dashed black lines show the final sharpness predicted by Cohen et al. (2021).

In Figure 5, we show how the parameter initialization of gradient descent affects the final loss and sharpness when using weight decay. For both quantities, we observe fractal structures in the parameter space, indicating that the training outcome is highly sensitive to the initialization. Further experiments in Appendix J show qualitatively similar fractal patterns also appear without weight decay. Experiment details of Figures 4 and 5 are provided in Appendix J.

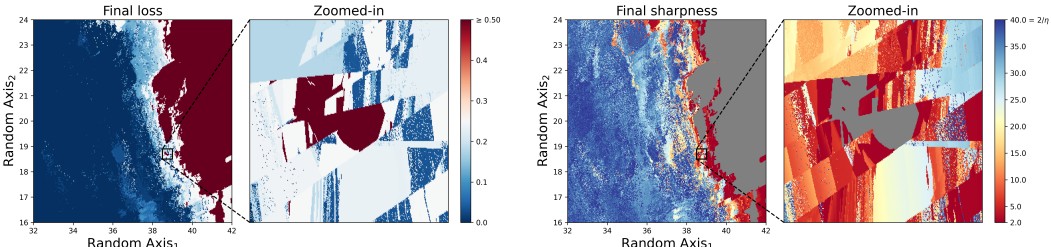

Figure 5: GD with weight decay in training a depth-3 ReLU network on a subset of CIFAR-10 for 3000 iterations. Shown is a random two-dimensional slice of the parameter space. Each initial parameter is colored by the value of the final loss and final sharpness, respectively. Left two panels: fractal basins of global minimizers (dark blue), sub-optimal solutions (white), and a region leading to divergence (dark red). Right two panels: the final sharpness is sensitive to the initialization, spanning a wide range of values below $2/\eta$. Gray points are initializations from which the algorithm diverged.

## 5 CONCLUSION

We offered a rigorous characterization of gradient descent with large step sizes in matrix factorization. Our results reveal two striking phenomena: near the convergence boundary, the selection of the minimizer is unpredictable, and adding regularization can induce a fractal convergence boundary that makes the convergence itself unpredictable. As a driver of this complexity, we suggested a covering map structure exhibited by the gradient descent update map on the parameter space.

**Limitations**    Although our characterizations substantially expand the state of knowledge in non-convex overparametrized optimization in the particular setting of matrix factorization, further research is needed to rigorously characterize the dynamics of large-step-size gradient descent in other settings, such as general initializations, deep matrix factorization, or neural networks with nonlinear activation functions. We believe the contributed insights can aid in the development of such programs.

**Future Directions**    We showed that at large step sizes there may not exist any simple algorithmic biases, but observed that biases could still be studied in a distribution sense. Further analyzing the properties of the distribution over global minimizers that is induced by a distribution of initializations is an interesting direction for future work. In particular, are there cases in which the distribution is uniform over a subset of minimizers, or cases in which it will concentrate in a predictable way?

**Reproducibility Statement**   Code to reproduce our experiments is made available at `https://github.com/shuangliang15/chaos-matrix-factorization`.

ACKNOWLEDGMENT

This project has been supported by NSF CCF-2212520 and NSF DMS-2145630. GM was partially supported by DARPA AIQ project HR00112520014, DFG SPP 2298 project 464109215, and BMFTR in DAAD project 57616814 (SECAI). We thank Ryan Tibshirani for insightful discussions.

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

## APPENDIX

The appendix is organized into the following sections.

- Appendix A: Relation to classical theory for chaos and fractals
- Appendix B: Measure of chaos in dynamical systems
- Appendix C: Diagonality of the target matrix
- Appendix D: Quotient dynamics of gradient descent
- Appendix E: Proofs for Section 3
- Appendix F: Non-existence of continuous dynamical invariant
- Appendix G: General matrix factorization
- Appendix H: Experiment details
- Appendix I: Additional experiments on matrix factorization
- Appendix J: Additional experiments on real-world data

## A  RELATION TO CLASSICAL THEORY FOR CHAOS AND FRACTALS

### A.1  CHAOTIC DYNAMICAL SYSTEMS

A common definition of chaotic dynamical systems is as follows (Devaney and Eckmann, 1987). A dynamical system $F\colon X \to X$, where $X$ is the state space, is chaotic if: (i) it is sensitive to initialization, (ii) it is topological transitive, and (iii) periodic points are dense in $X$. Here, sensitivity to initialization requires that there exists $\delta > 0$ such that for any $x \in X$ and any neighborhood of $x$, there exists $N > 0$ and $y$ in the neighborhood such that $\mathrm{dist}(F^N(x), F^N(y)) > \delta$. This is weaker than the property we presented in Theorem 1 and Theorem 4: when the converged points of two trajectories are different, the trajectories must differ by a positive difference at some time $N$, but not vice versa. In Proposition 15, we show that the boundary $\partial\mathcal{D}'_\eta$, as defined in Theorem 1, is invariant under gradient descent. In Proposition 18, we show that when restricted to $\partial\mathcal{D}'_\eta$, the gradient descent system is semi-conjugate to a one-dimensional system that is precisely Devaney chaotic. However, we show in Proposition 20 that the original gradient descent system is not Devaney chaotic when $d \geq 2$ as it fails to be topological transitive.

A system $F\colon X \to X$ is topological transitive if for any pair of non-empty open sets $U, V$, there exists $N$ such that $F^N(U) \cap V \neq \varnothing$. A family of dynamical systems that exhibit transitivity is the family of *Axiom-A diffeomorphisms* (Smale, 1967). These are dynamical systems where the set of non-wandering points is hyperbolic and is equal to the closure of the set of periodic points. A closed set $\Lambda$ is hyperbolic if it is forward invariant and at each point $x \in \Lambda$ the tangent space of the ambient space splits as a direct sum of stable and unstable subspaces. It is known that an Axiom-A diffeomorphism is always transitive on each of its *basic sets* (Bowen et al., 2008, Chapter 3). However, gradient update maps typically are not expected to satisfy this definition as they in general are not global diffeomorphisms.

Another definition of chaotic dynamical system is by Li and Yorke (1975). They considered a dynamical system $F\colon X \to X$ with $X \subset \mathbb{R}$ being an interval chaotic if $F$ has a periodic orbit with period three. They showed that if such a periodic orbit exists, then (i) $F$ has periodic orbit with any period; (ii) there exists an uncountable set $S \subset J$ such that, for every $p, q \in S$ with $p \neq q$,

$$\limsup_{N \to \infty} |F^N(p) - F^N(q)| > 0, \quad \liminf_{N \to \infty} |F^N(p) - F^N(q)| = 0,$$

and (iii) for every $p \in S$ and a periodic point $q \in J$,

$$\limsup_{N \to \infty} |F^N(p) - F^N(q)| > 0.$$

In Proposition 18, we showed that the restricted system $\mathrm{GD}_\eta|_{\partial\mathcal{D}'_\eta}$ is semi-conjugate to a one-dimensional system that is Li-Yorke chaotic. In general, Devaney chaos and Li-Yorke chaos do not imply each other. For a detailed comparison between different notions of chaotic dynamical systems, see the work of Elaydi (2007).

## A.2 Fractal basin boundary

The fractal convergence boundary studied in this work falls into a more general notion, called a fractal basin boundary. The seminal classification given by McDonald et al. (1985) divides fractal basin boundaries into three categories: quasicircles, locally connected but not quasicircles, and locally disconnected. The most regular type, quasicircle, is typical in the Julia sets of complex analytic maps. However, as noted by McDonald et al. (1985), properties of complex analytic maps do not generalize to real maps, and hence quasicircle is uncommon in real systems. We observe that the convergence boundary in our study falls in the second category, locally connected but not quasicircles. This type of boundary has been observed in several planar maps, i.e., dynamical systems defined on regions of $\mathbb{R}^2$. A well-known example is the following system:

$$x_{n+1} = \lambda_x x_n \mod (1), \quad y_{n+1} = \lambda_y y_n + \cos(2\pi x_n).$$

The basin boundary of this system is precisely the Weierstrass curve. McDonald et al. (1985) argued that a typical characteristic of this type of boundaries is the local stratification structure, which also appears in the convergence region in our case (see left panel in Figure 1). To our knowledge, however, all examples of locally connected boundaries appearing in the literature, including those presented by McDonald et al. (1985); Hunt et al. (1999); Rosa Jr and Ott (1999), are bounded, whereas the convergence region in our case is shown to be unbounded. Classical approaches do not apply to our study, as most of those theoretical studies are case-specific. The last category has the most complicated structure and, as noted by Aguirre et al. (2009), turns out to appear more commonly in physical systems. Boundaries in this category typically exhibit a Cantor set structure. Examples include the famous Hénon map and the horseshoe map. For a recent review of the fractal boundaries, we refer readers to Aguirre et al. (2009).

## B  Measure of chaos in dynamical systems

We introduce two measures of chaos in dynamical systems. In Appendix B.1 we introduce the topological entropy of a dynamical system and, in Appendix B.2 we discuss how the fractal dimension of the basin boundary implies unpredictability.

## B.1  Topological entropy

Let $F\colon X \to X$ be a dynamical system, where $X$ is the state space with a metric $d$. The idea behind topological entropy is to measure how fast the number of "distinct" trajectories increases as the trajectory length increases. To measure the difference between two trajectories of length $N$, consider

$$d_N(x, y) = \max_{0 \leq i \leq N-1} d(F^i(x), F^i(y)).$$

Then, the number of "distinct" trajectories of length $N$ is measured by

$$r(N, \varepsilon) = \max\left\{ |S|\colon d_N(x, y) > \varepsilon, \forall x, y \in S, x \neq y \right\},$$

where $|S|$ is the number of elements in $S$. The topological entropy of $F$, denoted $h(F)$, measures the exponential growth rate of $r(N, \varepsilon)$ as $N$ increases. Specifically, $h(F)$ is defined as follows:

$$h(F) = \lim_{\varepsilon \to 0^+} \limsup_{N \to \infty} \frac{\log r(N, \varepsilon)}{N}.$$

We give an example to provide more intuition. Consider the following symbolic dynamical systems:

$$\sigma\colon \{0, 1\}^\infty \to \{0, 1\}^\infty, \ \sigma(s_0 s_1 s_2 \cdots) = (s_1 s_2 \cdots).$$

Here the state space $\{0, 1\}^\infty$ denotes the set of all infinite sequence of two symbols 0 and 1, whose metric is defined by

$$d((s_0 s_1 \cdots), (s_0' s_1' \cdots)) = \sum_{j=0}^\infty \frac{|s_j - s_j'|}{2^j}.$$

The system $\sigma$ is called the *full-shift on two symbols*. Despite its simple definition, this system is unpredictable and chaotic (see, e.g., Devaney and Eckmann, 1987). In particular, periodic points are dense in the state space, and, there exists a trajectory that is dense in the state space, i.e., there is a

single trajectory that can come arbitrarily close to any point. In terms of predictability, consider two points $\boldsymbol{s} = (s_0 s_1 \cdots), \boldsymbol{s}' = (s_0' s_1' \cdots)$ that have the same first $m$ elements, but differ starting from the $(m + 1)$th element. By the definition of the distance, we have $d(\boldsymbol{s}, \boldsymbol{s}') \leq \sum_{j=0}^{\infty} 1/2^{m+j} = 1/2^{m+1}$. However, for all $N \geq m$, we have $d(\sigma^N(\boldsymbol{s}), \sigma^N(\boldsymbol{s})) > 1/2$. Therefore, even if two initial points are arbitrarily close to each other, one can not make any prediction on how close their trajectories will remain in the long term. This unpredictability stems from the richness of "distinct" trajectories. In fact, one can show that $h(\sigma) = \log 2 > 0$ (see, e.g., Vries, 2014). Note, the topological entropy of the gradient descent in matrix factorization is at least $\log 3$ (Theorem 1).

## B.2 Box-counting dimension and unpredictability

There have been numerous investigations discussing how a non-integer fractal dimension implies unpredictability in dynamical systems (see, e.g., Tél, 1990; Aguirre et al., 2009). Here we provide a brief introduction to this topic.

Recall that the *box-counting dimension* of a set $S$ is defined as the following limit if it exists:

$$D_B(S) = \lim_{\varepsilon \to 0} \frac{\log(N(\varepsilon))}{\log(1/\varepsilon)},$$

where $N(\varepsilon)$ is the number of boxes of side length $\varepsilon$ needed to cover the set $S$. For a dynamical system $F \colon X \to X$ where $X \subset \mathbb{R}^D$ is the state space, let $D_B$ be the box-counting dimension of a basin boundary and $D$ be the topology dimension of the state space. Consider a collection of trajectories and randomly perturb their initial points by a scale $\varepsilon$. Let $f(\varepsilon)$ denote the fraction of the trajectories that converge to a different point, i.e., whose initial point lies in a different basin of attraction after the perturbation. Thus, $f(\varepsilon)$ can be roughly viewed as the chance of making an error in predicting the converged point when the precision in specifying the initial point is $\varepsilon$. In general, the following scaling relation holds (Grebogi et al., 1983a):

$$f(\varepsilon) \sim \varepsilon^{D - D_B},$$

where $D - D_B$ is known as the *uncertainty exponent*. When the boundary is smooth, we have $D_B = D - 1$ and thus $f(\varepsilon) \sim \varepsilon$, i.e., the accuracy of the prediction of the converged point is proportional to the precision on the initial point. In contrast, when the boundary has a non-integer dimension, we have $D - 1 < D_B < D$ and hence $D - D_B < 1$. This implies that a substantial increase in the precision in specifying the initial point leads to only a very small increase in the accuracy of the prediction. This marks sensitivity to initialization and unpredictability. Note, the box-counting dimension of the projected boundary $T(\partial \mathcal{D}_\eta'')$ is estimated as $1.249$, yielding an uncertainty exponent $2 - 1.249 = 0.751$ (see Section 3.2).

## C Diagonality of the target matrix

We show that in matrix factorization (1), one may assume without loss of generality that the target matrix is diagonal. This simplification is a standard technique that has been widely adopted in the literature.

Let $Y = P_Y \Sigma_Y Q_Y^\top$ be the singular value decomposition of $Y$, where $P_Y, Q_Y \in O(d_y)$ and $\Sigma_Y \in \mathbb{R}^{d_y \times d_y}$ is diagonal. Consider the change of coordinates $U = \tilde{U} P_Y^\top$ and $V = \tilde{V} Q_Y^\top$. Recall that the $U$-update in minimizing $L(U, V)$ is given by

$$U_{t+1} = U_t - \eta V_t(V_t^\top U_t - Y^\top) - \eta \lambda U_t.$$

In the new coordinate, we have

$$
\begin{aligned}
\tilde{U}_{t+1} &= U_{t+1} P_Y \\
&= U_t P_Y - \eta V_t(V_t^\top U_t - Y^\top) P_Y - \eta \lambda U_t P_Y \\
&= \tilde{U}_t - \eta \tilde{V}_t Q_Y^\top (Q_Y \tilde{V}_t^\top \tilde{U}_t - Q_Y \Sigma_Y^T) - \eta \lambda \tilde{U}_t \\
&= \tilde{U}_t - \eta \tilde{V}_t(\tilde{V}_t^\top \tilde{U}_t - \Sigma_Y^T) - \eta \lambda \tilde{U}_t.
\end{aligned}
\tag{6}
$$

On the other hand, since the Frobenius norm is invariant under left- or right-multiplication by orthogonal matrices, the loss function in the new coordinates is given by

$$\tilde{L}(\tilde{U}, \tilde{V}) = \frac{1}{2}\|P_Y\tilde{U}^\top\tilde{V}Q_Y^\top - Y\|_F^2 + \frac{\lambda}{2}(\|\tilde{U}P_Y^\top\|_F^2 + \|\tilde{V}Q_Y^\top\|_F^2)$$

$$= \frac{1}{2}\|\tilde{U}^\top\tilde{V} - \Sigma_Y\|_F^2 + \frac{\lambda}{2}(\|\tilde{U}\|_F^2 + \|\tilde{V}\|_F^2).$$

Note the update iteration (6) coincides with the $\tilde{U}$-update in minimizing $\tilde{L}(\tilde{U}, \tilde{V})$ with gradient descent. An analogous calculation shows the same holds for the $\tilde{V}$-update. Therefore, one may directly study the gradient descent dynamics in minimizing $\tilde{L}(\tilde{U}, \tilde{V})$.

## D    QUOTIENT DYNAMICS OF GRADIENT DESCENT

We show that the gradient descent dynamics in the scalar factorization problem can be described by a quotient system, and we further establish key properties of this system.

Consider the map

$$T\colon \mathbb{R}^{2d} \to \mathbb{R}^2, \ T(u, v) = (u^\top v - y, \|u\|_2^2 + \|v\|_2^2).$$

Note that this definition differs from the one introduced in Section 3.2 by a constant shift of $-y$, where $y \in \mathbb{R}$ is target scalar of problem (2). This adjustment is made purely for convenience in presenting the proof. All results stated here extend trivially to the original formulation.

We will show that the gradient descent dynamics are fully captured by their evolution across the fibers of the map $T$. In other words, different initializations in the same fiber produce qualitatively identical trajectories. This reflects an inherent symmetry of the system. The quotient system factors out this symmetry and describes the fiber-wise dynamics. The term *quotient dynamical system* is borrowed from the theory of equivariant dynamical systems (see, e.g., Golubitsky and Stewart, 2003).

### D.1    QUOTIENT DYNAMICAL SYSTEM

We first introduce two properties of the map $T$: (i) the preimage of any measure-zero set has measure zero and (ii) the fiber of $T$ is generically a smooth manifold.

**Proposition 7.** *The preimage of any measure-zero set under the map $T$ is a measure-zero set.*

**Proof.** By Ponomarev (1987), it suffices to show the map $T$ is a submersion almost everywhere, i.e., the Jacobian of $T$ has rank two almost everywhere. Notice that

$$JT(u, v) = \begin{pmatrix} v & u \\ 2u & 2v \end{pmatrix}.$$

Hence, $\text{rank}(JT) < 2$ if and only if there exists $c \neq 0$ such that $cv = 2u$ and $cu = 2v$. This gives $c^2 v = 2cu = 4v$, and hence, $c = \pm 2$ or $v = 0$. The set $\{v = 0\}$ has zero measure. When $c = \pm 2$, we have $u = \pm v$ which also yields measure-zero set. This completes the proof.  $\square$

**Proposition 8.** *For almost all $(z, w) \in T(\mathbb{R}^{2d}) = \{(z, w) \in \mathbb{R}^2\colon w \geq 2|z + y|\}$, $T^{-1}(z, w)$ is diffeomorphic to $\mathbb{S}^{d-1} \times \mathbb{S}^{d-1}$.*

**Proof.** Notice that

$$\|u + v\|_2^2 = w + 2(z + y), \quad \|u - v\|_2^2 = w - 2(z + y).$$

Consider the linear bijection $p = u + v$ and $q = u - v$. It follows that

$$T^{-1}(z, w) = \left\{(p, q)\colon \|p\|_2^2 = w + 2(z + y), \|q\|_2^2 = w - 2(z + y)\right\}.$$

Thus, the fiber is diffeomorphic to $\mathbb{S}^{d-1} \times \mathbb{S}^{d-1}$ whenever $w \pm 2(z + y) \neq 0$. Note this only fails at a measure-zero set, which completes the proof.  $\square$

Next, we prove Proposition 3. In the terminology of dynamical systems, this result shows that the gradient descent system $\text{GD}_\eta$ is semi-conjugate to a planar system $f$ under the map $T$.

**Proposition 9** (Proposition 3)**.** *Let $(u_t, v_t)_{t \geq 0}$ denote the gradient descent trajectory in problem* (4) *with $\lambda \geq 0$. Let $(z_t, w_t) = T(u_t, v_t)$. Consider the map $f \colon \mathbb{R}^2 \to \mathbb{R}^2$ defined by*

$$f \begin{pmatrix} z \\ w \end{pmatrix} = \begin{pmatrix} \eta^2 z^3 + \eta^2 y z^2 + ((1 - \eta\lambda)^2 - \eta w + \eta^2 \lambda w)z + y\eta^2\lambda^2 - 2y\eta\lambda \\ ((1 - \eta\lambda)^2 + \eta^2 z^2)w - 4\eta z(1 - \eta\lambda)(z + y) \end{pmatrix}. \qquad (7)$$

*We have that $(z_{t+1}, w_{t+1}) = f(z_t, w_t)$ holds for all $t \geq 0$. In particular, $(u_t, v_t)$ converges to $\mathcal{M}$ if and only if $(z_t, w_t)$ converges to $T(\mathcal{M})$, and it converges to $(\mathbf{0}, \mathbf{0})$ if and only if $(z_t, w_t)$ converges to $(-y, 0)$.*

**Proof.** To ease the notation, let $(z, w) = (z_t, w_t)$ and $(z', w') = (z_{t+1}, w_{t+1})$ for arbitrary $t$. We have that

$$\begin{aligned}
z' &= (u')^\top v' - y \\
&= (u - \eta z v - \eta\lambda u)^\top (v - \eta z u - \eta\lambda v) - y \\
&= z - \eta z w - 2\eta\lambda u^\top v + \eta^2 z^2 u^\top v + \eta^2 \lambda z w + \eta^2 \lambda^2 u^\top v \\
&= z - \eta z w - 2\eta\lambda(z + y) + \eta^2 z^2(z + y) + \eta^2 \lambda z w + \eta^2 \lambda^2(z + y) \\
&= \eta^2 z^3 + \eta^2 y z^2 + ((1 - \eta\lambda)^2 - \eta w + \eta^2 \lambda w)z + y\eta^2\lambda^2 - 2y\eta\lambda.
\end{aligned}$$

Also, we have that

$$\begin{aligned}
w' &= \|u'\|^2 + \|v'\|^2 \\
&= (u - \eta z v - \eta\lambda u)^\top (u - \eta z v - \eta\lambda u) + (v - \eta z u - \eta\lambda v)^\top (v - \eta z u - \eta\lambda v) \\
&= ((1 - \eta\lambda)^2 + \eta^2 z^2)w - 4\eta z(1 - \eta\lambda)(z + y).
\end{aligned}$$

Note that, the loss function solely depends on $u^\top v - y$ and $\|u\|_2^2 + \|v\|_2^2$. Thus $(u_t, v_t)$ converges to $\mathcal{M}$ if and only if $(z_t, w_t)$ converges to $T(\mathcal{M})$. Also, note that, $T(u, v) = (-y, 0)$ if and only if $u = v = \mathbf{0}$. Thus $(u_t, v_t)$ converges to $(\mathbf{0}, \mathbf{0})$ if and only if $(z_t, w_t)$ converges to $(-y, 0)$. This completes the proof. $\qquad \square$

To further simplify the analysis, we consider the change of coordinates $\phi(z, w) = (\eta z, \eta w)$. Note that, under the map $\phi(z, w) = (\eta z, \eta w)$, the system $f$, as defined in (7), is topologically conjugate to

$$\begin{aligned}
F \begin{pmatrix} z \\ w \end{pmatrix} = \phi \circ f \circ \phi^{-1} \begin{pmatrix} z \\ w \end{pmatrix} &= \begin{pmatrix} z^3 + \eta y z^2 + ((1 - \eta\lambda)^2 - w + \eta\lambda w)z + y\eta^3\lambda^2 - 2y\eta^2\lambda \\ ((1 - \eta\lambda)^2 + z^2)w - 4z(1 - \eta\lambda)(z + \eta y) \end{pmatrix} \\
&= \begin{pmatrix} z^3 + \mu z^2 + ((1 - \nu)^2 - w + \nu w)z + \nu^2\mu - 2\mu\nu \\ ((1 - \nu)^2 + z^2)w - 4z(1 - \nu)(z + \mu) \end{pmatrix},
\end{aligned}$$
$$\qquad (8)$$

where we let $\mu = \eta y$ and $\nu = \eta\lambda$. With Cauchy-Schwartz inequality, it is straightforward to verify that the state space of $F$ is

$$\Omega = \left\{ (z, w) \in \mathbb{R}^2 \colon w \geq 2|z + \mu| \right\}.$$

The system $F$ has two parameters, $\mu$ and $\nu$, whereas $f$ has three, $\eta, y, \lambda$. Therefore, we instead study the system $F$. Note, trajectories of $F$ and those of $f$ only differ by a scale. Thus all results for $F$ extend trivially to $f$.

### D.2    PROPERTIES OF THE QUOTIENT DYNAMICS

We show that the map $F$, as defined in (8), is a proper map, i.e., the preimage of any compact set is compact.

**Proposition 10** (Properness)**.** *When $0 \leq \nu < 1 - |\mu|$, the map $F$ is proper on $\Omega$.*

**Proof.** Consider $\|(z_k, w_k)\| \to \infty$ for a sequence of points $(z_k, w_k)$. Let $(z'_k, w'_k) = F(z_k, w_k)$. Assume $(z'_k, w'_k)$ stays bounded. Since $(z_k, w_k)$ is unbounded and $\Omega$ is a cone, one must have $w_k \to \infty$. Notice that

$$w'_k = ((1 - \nu)^2 + z_k^2)w_k - 4z_k(1 - \nu)(z_k + \mu).$$

To make $w'_k$ bounded, $z_k$ has to be unbounded. However, as $w_k \geq 2|z_k + \mu|$,

$$\begin{aligned} w'_k &\geq ((1-\nu)^2 + z_k^2)w_k - 4|z_k| \cdot |z_k + \mu| \cdot |1 - \nu| \\ &\geq ((1-\nu)^2 + z_k^2)w_k - 2|z_k| \cdot w_k \cdot |1 - \nu| \\ &\geq w_k(|z_k| - (1-\nu))^2. \end{aligned}$$

Since $w_k, |z_k|$ are unbounded, $w'_k$ has to be unbounded, which yields a contradiction. This completes the proof. $\qquad\square$

Consider the function $Q$ defined as follows

$$Q \colon \Omega \to \mathbb{R}, \; Q(z, w) = w + \sqrt{w^2 - 16\mu z}.$$

We will frequently use $Q$ as a Lyapunov-like function to study the dynamics of $F$.

In the following result, we describe the level set structure of the function $Q$.

**Lemma 11** (Level-set structure). *Consider $Q(z, w) = w + \sqrt{w^2 - 16\mu z}$. Then, $Q(z, w) \geq 4|\mu|$ for all $(z, w) \in \Omega$. Moreover, we have that*

- *If $r = 4|\mu|$, then for all $(z, w) \in \Omega$, $Q(z, w) = r$ if and only if $w = 2\mathrm{sgn}(\mu)(z + \mu)$ and $w \leq 4|\mu|$; $Q(z, w) > r$ holds for all other points.*

- *If $r > 4|\mu|$, then for all $(z, w) \in \Omega$, $Q(z, w)$ is less than, equal to, larger than $r$ if and only if $-16\mu z - r^2 + 2rw$ is less than, equal to, larger than $0$, respectively.*

**Proof.** When $w \geq 2|z + \mu|$,

$$w^2 - 16\mu z \geq 4(z + \mu)^2 - 16\mu z = 4(z - \mu)^2 \geq 0.$$

Therefore, $Q$ is well-defined in $\Omega$.

When $\mu = 0$, we have that $Q(z, w) = 2w$. The claimed results clearly hold. In the sequel, consider $\mu \neq 0$. Let $r = Q(z, w) = w + \sqrt{w^2 - 16\mu z}$ and $s = w - \sqrt{w^2 - 16\mu z}$. Then we have $z = (rs)/(16\mu)$ and $w = (r + s)/2$. Notice that

$$r^2 - s^2 = (r + s)(r - s) = 2w \cdot 2\sqrt{w^2 - 16\mu z} \geq 0.$$

Since $w \geq 2|z + \mu|$, we have $w^2 \geq 4(z + \mu)^2$ and hence

$$\begin{aligned} (\frac{r + s}{2})^2 &\geq 4(\frac{rs}{16\mu} + \mu)^2 \\ &\Leftrightarrow (r^2 - 16\mu^2)(s^2 - 16\mu^2) \leq 0 \\ &\Leftrightarrow r \geq 4|\mu|, \; |s| \leq 4|\mu|. \end{aligned} \tag{9}$$

We have that for $(z, w) \in \Omega$,

$$\begin{aligned} w + \sqrt{w^2 - 16\mu z} &= 4|\mu| \\ &\Leftrightarrow \sqrt{w^2 - 16\mu z} = 4|\mu| - w \\ &\Leftrightarrow w^2 - 16\mu z = (4|\mu| - w)^2, \; w \leq 4|\mu| \\ &\Leftrightarrow w = 2\mathrm{sgn}(\mu)(z + \mu), \; w \leq 4|\mu|. \end{aligned}$$

Therefore,

$$\{Q(z, w) = 4|\mu|\} = \{w = 2\mathrm{sgn}(\mu)(z + \mu), \; w \leq 4|\mu|\} \subset \partial\Omega,$$

and $\{Q(z, w) > 4|\mu|\} = \Omega \setminus \{Q(z, w) = 4|\mu|\}$.

Now we consider $r > 4|\mu|$. When $w = 2(z + \mu)$, we have

$$-16\mu z - r^2 + 2rw = 0 \Leftrightarrow w = \frac{r + 4\mu}{2}.$$

When $w = -2(z + \mu)$, we have

$$-16\mu z - r^2 + 2rw = 0 \Leftrightarrow w = \frac{r - 4\mu}{2}.$$

Therefore, the line $-16\mu z - r^2 + 2rw = 0$ intersect $\partial\Omega$ at two points, whose $w$ coordinates are $\frac{r \pm 4\mu}{2}$. Since $r > 4|\mu|$, we have that $\frac{r \pm 4\mu}{2} < r$ always holds. This implies that, for all $(z, w) \in \Omega \cap \{-16\mu z - r^2 + 2rw \le 0\}$, we have $r - w > 0$. Thus, we have that for $(z, w) \in \Omega$ and $r > 4|\mu|$,

$$w + \sqrt{w^2 - 16\mu z} < r$$
$$\Leftrightarrow \sqrt{w^2 - 16\mu z} < r - w$$
$$\Leftrightarrow w^2 - 16\mu z < (r - w)^2$$
$$\Leftrightarrow -16\mu z - r^2 + 2rw < 0.$$

The above clearly holds when $<$ is changed to $=$ or $>$. This completes the proof. $\qquad\square$

We identify three invariant sets of the quotient system $F$. A set $S \subset \Omega$ is said to be an invariant set under $F$ if $F(S) \subset S$.

**Lemma 12** (Invariant boundary). *The boundary $\partial\Omega$ consists of two lines: $\{w = 2(z + \mu), w \ge 0\}$ and $\{w = -2(z + \mu), w \ge 0\}$. Each of the lines is an invariant set of $F$. Meanwhile, when $0 \le \nu < 1 - |\mu|$, the set $\{Q = 4|\mu|\}$ is invariant under $F$.*

**Proof.** Let $(z', w') = F(z, w)$. By direct computation, we have that

$$w' - 2(z' + \mu) = (w - 2(z + \mu))(1 + z - \nu)^2,$$
$$w' + 2(z' + \mu) = (w + 2(z + \mu))(-1 + z + \nu)^2. \tag{10}$$

It follows that if $(z, w) \in \partial\Omega = \{w = \pm 2(z + \mu)\}$, $F(z, w) \in \partial\Omega$.

According to Lemma 11, $\{Q = 4|\mu|\} = \{w = 2\mathrm{sgn}(\mu)(z + \mu), w \in [0, 4|\mu|]\}$. When $w = 2\mathrm{sgn}(\mu)(z + \mu)$, we have that the $w$-update is given by

$$w' = w\left(\left(\mathrm{sgn}(\mu)\frac{w}{2} - \mu\right)^2 + (1 - \nu)^2\right) - 2w\left(\mathrm{sgn}(\mu)\frac{w}{2} - \mu\right)(1 - \nu)$$
$$= w(\frac{w}{2} - 1 + \nu - |\mu|)^2$$
$$\triangleq \kappa(w).$$

We will analyze the image set of $\kappa([0, 4|\mu|])$. Clearly, the minimum of $\kappa([0, 4|\mu|])$ is $\kappa(0) = 0$. Let $A = -1 + \nu - |\mu|$. We have that $\kappa'(w) = 0$ if $w = -2A$ or $w = -2A/3$. Notice that when $0 \le \nu < 1 - |\mu|$, we have $4|\mu| \le -2A$. Therefore, the maximum of $\kappa([0, 4|\mu|])$ is either $\kappa(4|\mu|)$ or $\kappa(-2A/3)$. When $4|\mu| > -2A/3$, we have $(1 - \nu)/5 < |\mu| \le 1 - \nu$. Notice that

$$\kappa(\frac{-2A}{3}) = \frac{8(1 - \nu + |\mu|)^3}{27}.$$

Viewing $\kappa(\frac{-2A}{3})$ as a cubic function of $|\mu|$, we have that, as $1 - \nu > 0$, $\kappa(\frac{-2A}{3})$ is convex on $(1 - \nu)/5 < |\mu| \le 1 - \nu$. Therefore, to show $\kappa(\frac{-2A}{3}) < 4|\mu|$ for $(1 - \nu)/5 < |\mu| \le 1 - \nu$, it suffices to show this holds when $\mu = (1 - \nu)/5$ and $\mu = 1 - \nu$. Notice that

$$\frac{8}{27}((1 - \nu) + \frac{1 - \nu}{5})^3 \le 4 \cdot \frac{1 - \nu}{5} \Leftrightarrow (1 - \nu)^2 \le \frac{25 \cdot 27}{2 \cdot 6^3} \approx 1.56,$$

and that

$$\frac{8}{27}(1 - \nu + 1 - \nu)^3 < 4(1 - \nu) \Leftrightarrow (1 - \nu)^2 \le \frac{27}{16},$$

which are all satisfied. Therefore, $\kappa(-2A/3) \le 4|\mu|$ when $4|\mu| > -2A/3$. Meanwhile, we have

$$\kappa(4|\mu|) = 4|\mu|(|\mu| - 1 + \nu)^2.$$

Since $\nu < 1 - |\mu|$, we have that $-1 < |\mu| - 1 + \nu < 0$ and that $\kappa(4|\mu|) \le 4|\mu|$. Therefore, the image set of $\kappa([0, 4|\mu|])$ is contained $[0, 4|\mu|]$. This means that the set $\{Q = 4|\mu|\}$ is invariant under $F$, which completes the proof. $\qquad\square$

In the sequel, we present two important properties of the map $F$, which will be used in the proof of our main results. In the following result, we identify the region on which a single update of $F$ leads to a decrease, or an increase in the value of $Q$.

**Lemma 13** (Monotonicity region). *Assume $0 \leq \nu < 1 - |\mu|$. When $\nu = 0$, we have that, for $(z, w) \in \Omega$: (i) $Q(F(z, w)) = Q(z, w)$ if and only if $(z, w)$ lies in the set*

$$Z \triangleq \{w = \mu z + 4\} \cup \{z = 0\} \cup \{w = 2\text{sgn}(\mu)(z + \mu), w \leq 4|\mu|\};$$

*and (ii) If $(z, w) \notin Z$, we have $\Big(Q(F(z, w)) - Q(z, w)\Big) \cdot (w - \mu z - 4) > 0$.*

*When $\nu > 0$, we have that, for $(z, w) \in \Omega$, (i) $Q(F(z, w)) \leq Q(z, w)$ if and only if $(z, w)$ lies in the set*

$$\{z^2 \leq -\nu^2 + 2\nu\} \cup \left\{ w < -\frac{\mu(z^2 - 2\nu + \nu^2)}{z(\nu - 1)} - \frac{4z^2(\nu - 1)}{\nu^2 - 2\nu + z^2}, z^2 > -\nu^2 + 2\nu \right\},$$

*which contains $\{Q(z, w) < 8 - 4\nu\}$; and (ii) $Q(F(z, w)) = Q(z, w)$ if and only if $(z, w)$ lies in the set*

$$Z \triangleq \{Q(z, w) = 4|\mu|\} \cup \left\{ w = -\frac{\mu(z^2 - 2\nu + \nu^2)}{z(\nu - 1)} - \frac{4z^2(\nu - 1)}{\nu^2 - 2\nu + z^2}, z^2 > -\nu^2 + 2\nu \right\}.$$

**Proof.** Let $(z', w') = F(z, w)$. Assume that $\mu > 0$. Note the case of $\mu < 0$ can be proved via an analogous procedure. Let $r = Q(z, w)$ and let $s = w - \sqrt{w^2 - 16\mu z}$. When $r = 4|\mu|$, we have that $Q(F(z, w)) = Q(z, w)$ always holds, by Lemma 12. Consider $r > 4|\mu|$. Using Lemma 11, we have that the sign of $Q(z', w') - Q(z, w)$ is the same as that of the inner product between the vector pointing from $(z, w)$ to $(z', w')$ and the normal vector $(-8\mu, Q(z, w))$ of the line $-16\mu z - Q(z, w)^2 + 2Q(z, w)w = 0$, which is given by

$$
\begin{aligned}
&(-8\mu)(z' - z) + Q(z, w)(w' - w) \\
= &- 8\mu(z^3 + \mu z^2 + (\nu^2 - 2\nu - w + \nu w)z + \nu^2\mu - 2\mu\nu) + \\
&\quad Q(z, w)((\nu^2 - 2\nu + z^2)w - 4z(1 - \nu)(z + \mu)) \\
\propto &\ \mu^2(r^2 - 16\mu^2)(r^2 s^2 + 8rs^2(-1 + \nu) + 256\mu^2(-2 + \nu)\nu) \\
\propto &\ r^2 s^2 + 8\nu r s^2 - 8rs^2 + 256\mu^2\nu^2 - 512\mu^2\nu.
\end{aligned}
\tag{11}
$$

When $\nu = 0$, the above is equal to $s^2 r(r - 8)$. By noticing that $r > 4|\mu| > 0$, that the sign of $r - 8$ is the same as that of $w - \mu z - 4$ by Lemma 11, and that $s = 0$ if and only if $z = 0$, we have all the results for $\nu = 0$.

When $\nu > 0$, (11) has the same sign as

$$2\mu(z^2 + \nu^2 - 2\nu) - (1 - \nu)z(w - \sqrt{w^2 - 16\mu z}).$$

We have that for $(z, w) \in \Omega$,

$$
\begin{aligned}
&\{2\mu(z^2 + \nu^2 - 2\nu) - (1 - \nu)z(w - \sqrt{w^2 - 16\mu z}) \leq 0\} \\
=&\{z^2 \leq -\nu^2 + 2\nu\} \cup \left\{ w < -\frac{\mu(z^2 - 2\nu + \nu^2)}{z(\nu - 1)} - \frac{4z^2(\nu - 1)}{\nu^2 - 2\nu + z^2}, z^2 > -\nu^2 + 2\nu \right\},
\end{aligned}
$$

and

$$
\begin{aligned}
&\{2\mu(z^2 + \nu^2 - 2\nu) - (1 - \nu)z(w - \sqrt{w^2 - 16\mu z}) = 0\} \\
=&\left\{ w = -\frac{\mu(z^2 - 2\nu + \nu^2)}{z(\nu - 1)} - \frac{4z^2(\nu - 1)}{\nu^2 - 2\nu + z^2}, z^2 > -\nu^2 + 2\nu \right\},
\end{aligned}
$$

Next, we show that $\{Q < 8 - 4\nu\}$ is contained in the above set. Notice that $8 > 4(|\mu| + \nu)$ always holds when $0 \leq \nu < 1 - |\mu|$. So $8 - 4\nu > 4|\mu|$ and, by Lemma 11, the level set $\{Q = 8 - 4\nu\}$ is on the line

$$w = \frac{8\mu}{8 - 4\nu}z + \frac{8 - 4\nu}{2}.$$

Then it suffices to show that when $z^2 > -\nu^2 + 2\nu$, the following holds

$$-\frac{\mu(z^2 - 2\nu + \nu^2)}{z(\nu - 1)} - \frac{4z^2(\nu - 1)}{\nu^2 - 2\nu + z^2} - (\frac{8\mu}{8 - 4\nu}z + \frac{8 - 4\nu}{2}) \geq 0. \tag{12}$$

By direct computation, we have that the level set $\{Q = 8 - 4\nu\}$ intersects $\partial\Omega$ at $z = \pm(2 - \nu)$. Hence, by the cone structure of $\Omega$, we have $z^2 < (2 - \nu)^2$ if $Q < 8 - 4\nu$. Therefore, multiplying $z(z^2 + \nu^2 - 2\nu)$ to both sides of the above inequality and assuming $z > 0$, we have that the inequality is equivalent to

$$\frac{\nu}{(\nu - 2)(\nu - 1)} \cdot (-z^2 + (2 - \nu)^2) \cdot (-\mu z^2 + (4 - 6\nu + 2\nu^2)z - \mu(-2\nu + \nu^2)) \geq 0$$
$$\Leftrightarrow (-z^2 + (2 - \nu)^2) \cdot (-\mu z^2 + (4 - 6\nu + 2\nu^2)z - \mu(-2\nu + \nu^2)) \geq 0 \tag{13}$$
$$\Leftrightarrow -\mu z^2 + (4 - 6\nu + 2\nu^2)z - \mu(-2\nu + \nu^2)) \geq 0.$$

The symmetry axis of the parabola is $(\nu - 1)(\nu - 2)/\mu > 0$. Since $1 - \nu > \mu$, the symmetry axis lies in $(2 - \nu, +\infty)$. Notice that

$$-\mu z^2 + (4 - 6\nu + 2\nu^2)z - \mu(-2\nu + \nu^2)|_{z = \sqrt{-\nu^2 + 2\nu}} \geq 0$$
$$\Leftrightarrow 2(1 - \nu)\sqrt{-\nu^2 + 2\nu} \geq 0,$$

which is satisfied. Therefore, (13) holds and (12) holds. The case of $z < 0$ can be proved with a similar procedure. Thus, we have that $\{Q < 8 - 4\nu\}$ is inside the set $\{Q(z, w) \geq Q(F(z, w))\}$.

Finally, when $\mu = 0$, we have that $Q(z, w) = 2w$. We have that

$$Q(F(z, w)) - Q(z, w) = w(z^2 + (1 - \nu)^2) - 4z^2(1 - \nu) - w$$
$$= 4z^2(-1 + \nu) + w(z^2 - 2\nu + \nu^2).$$

It is straightforward to verify that the claimed results hold for this case. This completes the proof. $\square$

In the following result, we characterize the preimage map of $F$, which in general is a multi-valued map.

**Proposition 14** (Preimage structure). *Assume $0 \leq \nu < 1 - |\mu|$. Consider the sets*

$$\begin{cases} B = \{(z, w) \in \Omega^o : Q(z, w) > 6 - 4\nu\} \\ A_0 = \{(z, w) \in \Omega^o : z < \nu - 1\} \\ A_2 = \{(z, w) \in \Omega^o : z > 1 - \nu\}. \end{cases}$$

*The restrictions $F|_{\mathrm{cl}(A_0)}$, $F|_{\mathrm{cl}(A_2)}$ are homeomorphisms onto $\Omega$. Moreover, there exists homeomorphisms $G_0 : \Omega \to \mathrm{cl}(A_0)$, $G_1 : \mathrm{cl}(B) \to G_1(\mathrm{cl}(B)) \subset \{(z, w) \in \Omega : |z| \leq 1 - \nu\}$, $G_2 : \Omega \to \mathrm{cl}(A_2)$ such that $F \circ G_i$ is an identity map on the domain of $G_i$ for $i = 0, 1, 2$.*

**Proof.** Notice that the critical points of $F$ lie in the set

$$\{\det JF(z, w) = -(1 + z - \nu)(-1 + z + \nu)(1 - w - 3z^2 - 2z\mu - 2\nu + w\nu + \nu^2) = 0\}. \tag{14}$$

Since $|\mu| < 1 - \nu$, the bottom tip of $\Omega$, $(-\mu, 0)$, lies in $(\nu - 1, 1 - \nu)$. Therefore, $A_0$ is bounded by $w = -2(z + \mu)$ and $z = \nu - 1$. Notice that the parabola $1 - w - 3z^2 - 2z\mu - 2\nu + w\nu + \nu^2 = 0$ intersects $w = -2(z + \mu)$ at $z = (-2\mu - 1 + \nu)/3$ and we have

$$(-2\mu - 1 + \nu)/3 > \nu - 1 \Leftrightarrow \mu + \nu < 1,$$

which is satisfied by assumption. Therefore, $\det JF$ vanishes nowhere on $A_0$.

We next show that $F(A_0) = \Omega^o$. For an arbitrary $(z_0, w_0) \in \Omega^o$, $(z, w) \in F^{-1}(z_0, w_0)$ if $(z, w)$ solves the following system

$$\begin{cases} z^3 + \mu z^2 + ((1 - \nu)^2 - w + \nu w)z + \nu^2\mu - 2\mu\nu = z_0 \\ ((1 - \nu)^2 + z^2)w - 4z(1 - \nu)(z + \mu)) = w_0. \end{cases} \tag{15}$$

For $z \neq 0$, solving (15) is equivalent to solving

$$w = \frac{z^3 + \mu z^2 + (1-\nu)^2 z + \nu^2 \mu - 2\mu\nu - z_0}{(1-\nu)z} = \frac{4z(1-\nu)(z+\mu) + w_0}{z^2 + (1-\nu)^2},$$

which is equivalent to solving the following quintic equation

$$
\begin{aligned}
p(z) = z^5 + \mu z^4 - 2(\nu-1)^2 z^3 + ((-2\nu^2 + 4\nu - 3)\mu - z_0)z^2 \\
+ (\nu-1)(\nu^3 - 3\nu^2 + 3\nu + w_0 - 1)z + (\nu-1)^2(\mu\nu(\nu-2) - z_0) = 0.
\end{aligned}
\tag{16}
$$

Notice that $p(-\infty) = -\infty$ and $p(-1+\nu) = (\nu-1)^2(w_0 - 2(z_0 + \mu)) > 0$. Hence, $p$ has at least one root in $(-\infty, -1+\nu)$. By Lemma 12, in particular, by (10), we have that, when viewing $F$ as a map on $\mathbb{R}^2$:

$$F^{-1}(\partial\Omega) = \partial\Omega \cup \{z = \pm(1-\nu)\}, \text{ and } F^{-1}(\Omega) \subset \Omega.$$

Therefore, the above root of $p$ corresponds to one preimage in $A_0$. This means that $F(A_0) = \Omega^\circ$.

For any compact set $K \subset \Omega^o$. Note $K$ is also compact in $\Omega$. Since $F$ is proper by Proposition 10, $F^{-1}(K)$ is compact. Since $K \cap \partial\Omega = \varnothing$ and $\partial(\Omega) \supset F(\partial A_0)$, we have $F^{-1}(K) \cap \partial A_0 = \varnothing$. Therefore, $(F|_{A_0})^{-1}(K) = F^{-1}(K) \cap A_0 = F^{-1}(K) \cap \mathrm{cl}(A_0)$, which is a closed in $F^{-1}(K)$. As a closed subset of a compact space is compact, we have $(F|_{A_0})^{-1}(K)$ is compact. Hence, $F|_{A_0}$ is a proper map. Since $\Omega^o$ is simply-connected, by Hadamard Inverse Function theorem, we have that $F|_{A_0}$ is a homeomorphism from $A_0$ to $\Omega^o$.

We now show that $F$ maps $\partial A_0$ bijectively to $\partial\Omega$. Since $A_0 \subset \mathrm{cl}(A_0)$, we have that $F(A_0) = \Omega^o \subset F(\mathrm{cl}(A_0))$. Since $\mathrm{cl}(A_0)$ is compact, therefore, $F|_{\mathrm{cl}(A_0)} \colon \mathrm{cl}(A_0) \to \Omega$ is proper, and hence is closed (see, e.g., Lee, 2000, Theorem 4.95). Therefore, $F(\mathrm{cl}(A_0))$ is a closed set that contains $\Omega^o$. Hence, $\mathrm{cl}(\Omega^o) = \Omega \subset F(\mathrm{cl}(A_0))$. Since $F(A_0) = \Omega^o$, we have $\partial\Omega \subset F(\partial A_0)$, which means $F|_{\partial A_0}$ is onto $\partial\Omega$. We next show it is also injective. By Lemma 12, we know $F$ maps $\{z = \nu - 1\}$ to $\{w = 2(z + \mu)\}$ and maps $\{w = -2(z + \mu)\}$ to itself. When $z = \nu - 1$, the $w$-update under $F$ is given by

$$w' = w((1-\nu)^2 + (-1+\nu)^2) - 4(1-\nu)(-1+\nu)(-1+\mu+\nu),$$

which is linear in $w$. Therefore, $F|_{z=1-\nu}$ must be an injection. When $w = -2(z + \mu)$, the $w$-update under $F$ is given by

$$w' = w(\frac{w}{2} - 1 + \mu + \nu)^2.$$

As a function $w$, $w'$ have two critical points, $w = 2(1 - \mu - \nu)$ and $w = \frac{2}{3}(1 - \mu - \nu)$. Notice that $z = \nu - 1$ intersects $\partial\Omega$ at $(\nu - 1, 2(1 - \mu - \nu))$. Then when $(z, w) \in \mathrm{cl}(A_0)$, we have $w \geq 2(1 - \mu - \nu)$. Since $1 - \mu - \nu > 0$, we have that the above $w'$ is monotonic with $w$. Therefore, $F|_{\mathrm{cl}(A_0)}$ is an injection. It follows that, $F|_{\partial A_0}$ is a bijection to $\partial\Omega$ and $F|_{\mathrm{cl}(A_0)}$ is a bijection to $\Omega$. Since $F|_{\mathrm{cl}(A_0)}$ is a closed map, its inverse is continuous. Therefore, $F|_{\mathrm{cl}(A_0)}$ is a homeomorphism. The proof for $A_2$ is similar and thus is omitted.

Finally, we analyze the behavior of $F$, as a map onto $B$. We show that every points in $B$ are regular values. Note that,

$$
\begin{aligned}
6 - 4\nu - \frac{1 - 3z^2 - 2z\mu - 2\nu + \nu^2}{1 - \nu} > 0 \\
\Leftrightarrow 3z^2 + 2\mu z + (3\nu^2 - 8\nu + 5) > 0.
\end{aligned}
\tag{17}
$$

For this parabola of $z$, we have

$$
\begin{aligned}
(2\mu)^2 - 4 \cdot 3(3\nu^2 - 8\nu + 5) &< 0 \\
\Leftrightarrow |\mu|^2 &< 3(3\nu - 5)(\nu - 1) \\
\Leftarrow (1 - \nu)^2 &< -3(3\nu - 5)(1 - \nu) \\
\Leftarrow \nu &< \frac{7}{4}.
\end{aligned}
$$

Therefore, we have that (17) holds and that

$$Q(z, \frac{1 - 3z^2 - 2z\mu - 2\nu + \nu^2}{1 - \nu}) < 6 - 4\nu$$

$$\Leftrightarrow \sqrt{(\frac{1 - 3z^2 - 2z\mu - 2\nu + \nu^2}{1 - \nu})^2 - 16\mu z} < 6 - 4\nu - \frac{1 - 3z^2 - 2z\mu - 2\nu + \nu^2}{1 - \nu}$$

$$\Leftrightarrow -16\mu z + 2(6 - 4\nu) \cdot \frac{1 - 3z^2 - 2z\mu - 2\nu + \nu^2}{1 - \nu} - (6 - 4\nu)^2 < 0$$

$$\Leftrightarrow (6\nu - 9)z^2 + 2\mu(4\nu - 5)z + (2\nu^3 - 9\nu^2 + 13\nu - 6) < 0.$$

For this new parabola of $z$, we have its discriminant is negative if

$$\mu^2(5 - 4\nu)^2 - 12(3 - 2\nu)^2(\nu^2 - 3\nu + 2) < 0$$

$$\Leftrightarrow (1 - \nu)^2(5 - 4\nu)^2 - 12(3 - 2\nu)^2(\nu - 1)(\nu - 2) < 0$$

$$\Leftrightarrow 32\nu^3 - 182\nu^2 + 331\nu - 191 < 0.$$

By differentiation computation, we claim that the last equation holds when $\nu \in [0, 1]$. Therefore, we prove that the maximum $Q$ value on the parabola $\iota\colon 1 - w - 3z^2 - 2z\mu - 2\nu + w\nu + \nu^2 = 0$ is at most $6 - 4\nu$. Notice that all the critical values of $F$ is given by $\partial\Omega \cup F(\iota)$. But what we have shown and Lemma 13, we have

$$Q(z, w) \le 6 - 4\nu, \ \forall (z, w) \in F(\iota).$$

Therefore, we have that every points in $B$ are regular values.

Now we show that $|F^{-1}(z, w)| = 3$ for $(z, w) \in B$. To this end, we first consider a special point: $x^* = (-\mu + \mu(1 - \nu)^2, w^*(1 - \nu)^2) = F(0, w^*)$ for some $w^*$ such that $x^* \in B$. Notice that the $w$-coordinate of $x^*$ tends to infinity as $w^*$ tends to infinity. Hence, $w^*$ can be arbitrarily large while keeping $x^* \in B$, i.e., $Q(x^*) > 6 - 4\nu$. We show that $|F^{-1}(x^*)| = 3$. Plugging $z_0 = -\mu + \mu(1 - \nu)^2$ and $w_0 = w^*(1 - \nu)^2$ to (16) gives

$$0 = z^4 + \mu z^3 - 2(\nu - 1)^2 z^2 - 3\mu(\nu - 1)^2 z + (\nu - 1)^3(-1 + \nu + w^*)$$
$$\Leftrightarrow z^4 + \mu z^3 = (\nu - 1)^2(2z^2 + 3\mu z - (\nu - 1)(-1 + \nu + w^*)). \tag{18}$$

The left-hand side is a continuous function and thus has a finite upper bound when $z \in [-1 + \nu, 1 - \nu]$. The right-hand side is a parabola, whose symmetry axis is at $-3\mu/4$. Hence, it's global minimum is

$$-\frac{9}{8}\mu^2(\nu - 1)^2 - (\nu - 1)^3(-1 + \nu + w^*).$$

Notice that this quantity tends to $+\infty$ as $w^*$ tends to $+\infty$. Hence, for large enough $w^*$, equation (18) does not have a solution on $[-1 + \nu, 1 - \nu]$. It follows that $F^{-1}(x^*)$ does not have any element in $\{z \in [-1 + \nu, 1 - \nu]\}$ except $(0, w^*)$. By what we have shown, $F|_{cl(A_0)}$ and $F|_{cl(A_2)}$ are bijections onto $\Omega$. Hence, $F^{-1}(x^*)$ have exactly one element in $A_0$ and exactly one in $A_2$. Therefore, $|F^{-1}(x^*)| = 3$. Now consider any other point in $B$ and a path connecting $x^*$ and that point. Since every point in $B$ is a regular value, by the stack of records theorem, the function $|F^{-1}(\cdot)|$ is locally constant. (Stack of records theorem requires the domain to be compact and this can be achieved by confining $F$ on $w \le W$ for some large enough $W$ so that the image contains the path. This is guaranteed by properness of $F$.) Note the path is compact and thus $|F^{-1}(z, w)|$ is a constant on the entire $B$ and hence is 3.

Given $(z, w) \in B$, in $F^{-1}(z, w)$ we already know there are exactly one point in $A_0$ and exactly one point in $A_2$. Hence, the third point must lie in $\{(z, w) \in \Omega\colon |z| < 1 - \nu\}$. We define this map by $G_1\colon B \to \{(z, w) \in \Omega\colon |z| < 1 - \nu\}$ and let $A_1 = G_1(B)$. By what we have shown, $\det JF$ vanishes nowhere on $A_1$. Note by construction of $A_1$, $F|_{A_1}(A_1) = B$. With a similar treatment as we used for $A_0$, we can show $F|_{A_1}$ is proper. As $B$ is simply connected, we have that $F$, when restricted to $A_1$, is a homeomorphism to $B$. As we shown above, $F$ is a bijection from $|z| = \pm(1 - \nu)$ to $\partial\Omega$. Moreover, we claim that $G_1$ can be extended to $\{Q = 6 - 4\nu\}$ in a bijective manner, as one can choose $C$ slightly smaller than $6 - 4\nu$ and apply the same analysis for $\{(z, w) \in \Omega, |z| < 1, Q > C\}$ as we did for $B$. Hence, $F$, when restricted to $cl(A_1)$, is a bijection onto $cl(B)$. Note, $F|_{cl(A_1)}$ is proper and hence its inverse is continuous. Therefore, $F|_{cl(A_1)}$ is a homeomorphism. This completes the proof. $\square$

# E    PROOFS FOR SECTION 3

In this section, we present the proofs of our main results. The key idea is to first analyze the quotient dynamical system introduced in Appendix D, and then translate the conclusions back to the original gradient descent system.

## E.1    UNREGULARIZED PROBLEM

Preliminary results are first presented in Appendix E.1.1, and the proof of Theorem 1 is given in Appendix E.1.2.

### E.1.1    PRELIMINARY RESULTS

As discussed in Appendix D, the gradient descent system $\mathrm{GD}_\eta$ is semi-conjugate to the following system $F$:

$$F \begin{pmatrix} z \\ w \end{pmatrix} = \begin{pmatrix} z^3 + \mu z^2 - zw + z \\ (z^2 + 1)w - 4z(z + \mu) \end{pmatrix},$$

where $\mu = y\eta$ denote the parameter of the system and the state space of $F$ is

$$\Omega = \{w \geq 2|z + \mu|\}.$$

In the following several results, we characterize the long term behavior of the orbits of $F$. We will use the terms trajectory and orbit interchangeably. We say an orbit $\{z_k, w_k\}$ converges to a set $S$ if $d((z_k, w_k), S) \to 0$, where $d(x, S) = \inf_{y \in S} d(x, y)$. Unless stated otherwise, we use $(z', w')$ to denote $F(z, w)$.

**Proposition 15** (Long-Term Dynamics)**.** *Assume $|\mu| \leq 1$. Given an initial condition $(z_0, w_0) \in \Omega$, we have that:*

- *If $w_0 < \mu z_0 + 4$, the orbit stays in $\{w < \mu z + 4\}$ and converges to*

$$\{w = 2\mathrm{sgn}(\mu)(z + \mu), w \leq 4|\mu|\} \cup \{z = 0\}.$$

- *If $w_0 > \mu z_0 + 4$, the orbit either diverges, in the sense that $w_k \to \infty$, or converges to $\{z = 0\}$ in finite steps.*

- *If $w_0 = \mu z_0 + 4$, the orbit stays in $\{w = \mu z + 4\}$.*

**Proof.** Consider the function $\Delta Q(z, w) = Q(F(z, w)) - Q(z, w)$. Let $(z_k, w_k) = F^k(z_0, w_0)$ for $k \geq 1$. When $w_0 < \mu z_0 + 4$, by Lemma 13, we have $Q(z_{k+1}, w_{k+1}) \leq Q(z_k, w_k)$ for all $k \geq 0$. Hence, the trajectory stays in the region $\{Q < 8\} = \{w < \mu z + 4\}$. Since $Q(z_k, w_k)$ is monotonic and is non-negative, it converges to some finite value and $\Delta Q(z_k, w_k)$ converges to zero. By Lemma 13, we have that

$$\{\Delta Q = 0\} = Z = \{z = 0\} \cup \{w = \mu z + 4\} \cup \{w = 2\mathrm{sgn}(\mu)(z + \mu), w \leq 4|\mu|\}.$$

If $(z_k, w_k)$ does not converges to $Z$, there exists $\varepsilon_0$ such that for any $K$ there exists $k > K$ such that $d((z_k, w_k), Z) \geq \varepsilon_0$. Note that, the function $\Delta Q$, when restricted to the compact set $\{w \leq \mu z + 4 \colon d((z, w), Z) \geq \varepsilon_0\}$, is non-positive and continuous. Thus, it obtains its maximal value and the maximum is strictly negative. Hence, $d((z_k, w_k), Z) \geq \varepsilon_0$ implies that $\Delta Q(z_k, w_k) < -\delta$ for some $\delta > 0$, which contradicts the fact that $\Delta Q \to 0$. Therefore, $(z_k, w_k)$ converges to

$$Z \cap \{w < \mu z + 4\} = \{z = 0\} \cup \{w = 2\mathrm{sgn}(\mu)(z + \mu), w \leq 4|\mu|\}.$$

When $w_0 > \mu z_0 + 4$, similarly, we have that $Q(z_{k+1}, w_{k+1}) \geq Q(z_k, w_k)$. Hence, $(z_k, w_k)$ stays in the region $\{w > \mu z + 4\}$ for all $k \geq 0$. Hence, $Q(z_k, w_k)$ either diverges to infinity or converges to a finite value. If it diverges, $w_k$ must also diverge, since the function $Q$, when restricted to $\{w \leq \bar{w}\}$ for any fixed $\bar{w}$, is continuous and hence is upper bounded. If $Q(z_k, w_k)$ converges to some finite value, then $\Delta Q$ converges to zero and the trajectory must remain within the compact region $\{w \leq C\}$ for some $C > 0$. Using arguments similar to those above, we have that $(z_k, w_k)$ must converge to

$Z \cap \{w > \mu z + 4\} = \{z = 0\}$. Now assume the convergence is in infinite steps, i.e., $|z_k| \neq 0$ for all $k \in \mathbb{N}$. Then the sequence $|z_{k+1}/z_k|$ is well defined and converges to one. Notice that we have

$$|\frac{z_{k+1}}{z_k}| = |z_k^2 + \mu z_k - w_k + 1| \geq \left||z_k^2 + \mu z_k| - |w_k - 1|\right|. \tag{19}$$

Note as $z_k \to 0$, the above lower bound is dominated by $|w_k - 1|$. Since $\{w > \mu z + 4\} \cap \{z = 0\} = \{(0, w): w \geq 4\}$, $|\frac{z_{k+1}}{z_k}|$ is lower bounded by $1 + \delta$ for some $\delta > 0$. This contradicts the fact that $|z_{k+1}/z_k|$ converges to one. Hence, the convergence must occur within finite steps.

Finally, the result for the case $w_0 = \mu z_0 + 4$ directly comes from Lemma 13. This completes the proof. □

**Proposition 16** (Convergence). *When $|\mu| > 1$, almost all initializations does not converge to $\{z = 0\}$. When $|\mu| < 1$, almost all initializations with $Q(z, w) < 8$ converges and almost all initializations with $Q(z, w) > 8$ diverges.*

**Proof.** Consider $|\mu| < 1$. By Proposition 15, when $Q(z, w) > 8$, initializations either converge to $\{z = 0\}$ in finite steps or diverge. Notice that converging within finite steps means that the initialization lies in the set

$$\cup_{N=0}^{\infty} F^{-N}(\{z = 0\}).$$

As the Jacobian of $F$ has full rank almost everywhere, the above set is a measure-zero set (Ponomarev, 1987). Hence, almost all initializations with $Q > 8$ diverge.

When $Q < 8$, by Proposition 15, we have that the orbit converges to $\{z = 0\}$ or to $\{Q = 4|\mu|\} = \{w = 2\text{sgn}(\mu)(z + \mu), w \leq 4|\mu|\}$. Notice that, when $w = 2\text{sgn}(\mu)(z + \mu)$, the $w$-update under $F$ is given by

$$w' = \kappa(w) = \frac{1}{4}w\left(w - 2 - 2|\mu|\right)^2. \tag{20}$$

Consider $\kappa$ as a one-dimensional dynamical system defined on $[0, 4|\mu|]$. For this one-dimensional system, it is straight forward to obtain that, there are two fixed points: $w = 0$ and $w = 2|\mu|$, and also that, when $|\mu| < 1$, all orbits converge to $w = 2|\mu|$ except the one with initial value $w = 0$. Note that $w = 0$ corresponds to the fixed point $(-\mu, 0)$ of $F$. The Jacobian of $F$ at $(-\mu, 0)$ has eigenvalues $(1 + \mu)^2, (-1 + \mu)^2$. Therefore, $(-\mu, 0)$ is a hyperbolic fixed point. By the local stable manifold theorem and the fact that $\{w = -\text{sgn}(\mu)(z + \mu)\}$ is invariant under $F$, we have that the basin of attraction of $(-\mu, 0)$ can be given by

$$B(-\mu, 0) = \cup_{N=0}^{\infty} F^{-N}(O \cap \{w = -\text{sgn}(\mu)(z + \mu)\}),$$

for some small neighborhood $O$ of $(-\mu, 0)$. This set is a measure-zero set, since the Jacobian of $F$ has full rank almost everywhere. Therefore, for all initializations lies in $\{Q(z, w) < 8\} \setminus B(-\mu, 0)$, the orbit converges to $\{z = 0\}$ or, to $\{Q = 4|\mu|\} \setminus \{(-\mu, 0)\}$. Consider any fixed initialization $(z_0, w_0)$ in the second case. Since $(-\mu, 0)$ is a hyperbolic fixed point, the orbit does not have an accumulation point in some neighborhood of $(-\mu, 0)$. Therefore, the omega-limit set $\omega(z_0, w_0) \subset \{Q = 4|\mu|\} \setminus \{(-\mu, 0)\}$. The omega-limit set is non-empty, as the orbit is always bounded. Note, for any $m \in \omega(z_0, w_0)$, we have that $F^N(m) \to (0, -2|\mu|)$ as $N$ tends to infinity, as explained above. Since the omega-limit set is forward invariant under $F$ and closed, we have that $(0, -2|\mu|) \in \omega(z_0, w_0)$. It follows that the orbit visits an arbitrarily small neighborhood $O$ of $(0, -2|\mu|)$ at some time $k$. Take the neighborhood as $O = \{|Q(z, w) - 4|\mu|| < \varepsilon, |z| < \varepsilon\}$. Notice that

$$|\frac{z_{k+1}}{z_k}| = |z_k^2 + \mu z_k - w_k + 1| \leq |z_k^2 + \mu z_k| + |w_k - 1|.$$

Since $|z^2 + \mu z| + |w - 1| \to |2|\mu| - 1|$ as $(z, w) \to (0, 2|\mu|)$, and $|2|\mu| - 1| < 1$, we can choose $\varepsilon$ small enough such that $|\frac{z_{k+1}}{z_k}| < 1 - \delta$ for some $\delta > 0$. Therefore, $|z_{k+1}| < |z_k|$. Meanwhile, note that the $Q$ value monotonically decreases as the orbit is in $\{Q < 8\}$. Hence, for all $j \geq k$, $(z_j, w_j) \in O$ and $|\frac{z_{j+1}}{z_j}| < 1 - \delta$. It follows that $z_j \to 0$ and the orbit converges to $\{z = 0\}$. Therefore, we have that almost all initializations with $Q < 8$ converge to $\{z = 0\}$.

Finally, consider $|\mu| > 1$. Since $2|\mu| = 2\text{sgn}(\mu)(0 + \mu)$, we have that $\inf\{w: (0, w) \in \Omega\} = 2|\mu| > 2$. Using arguments similar to those in Proposition 15, in particular in (19), we have that converging to any global minimizer must occur within finite steps. As shown above, those initializations form a measure-zero set. This completes the proof. □

From the proofs of the preceding two propositions, we obtain the following corollary.

**Corollary 17.** *Consider gradient descent with step $\eta$ in problem* (2). *Any global minimizer with $\|u\|^2 + \|v\|^2 \geq 2/\eta$ is an unstable minimizer, i.e., it repels orbits in its neighborhood. Consequently, initializations that converge to such a minimizer form a measure-zero set. Moreover, when $|\mu| > 1$, i.e., $\eta|y| > 1$, all global minimizers are unstable.*

Next, we analyze dynamics on the invariant set $\{Q(z,w) = 8\} = \{w = \mu z + 4\}$. Observe that when restricted to this set, $F$ reduces to the following one-dimensional system:

$$\tilde{F}(z) = z^3 + \mu z^2 - z(\mu z + 4) + z = z^3 - 3z, \quad z \in [-2, 2].$$

The following result shows that $\tilde{F}$ is a chaotic dynamical system.

**Proposition 18** (Chaotic Boundary Dynamics). *Assume $|\mu| < 1$. The system $\tilde{F}$ on $I = [-2, 2]$ is Devaney chaotic and has topological entropy $\log 3$. Moreover, there exists periodic orbits with any period and thus $\tilde{F}$ is also Li-Yorke chaotic.*

**Proof.** We first seek a simpler system which is topologically conjugate to $\tilde{F}$. Notice that $\tilde{F}$ is a continuous map from $[-2, 2]$ to itself. Consider $\psi_0(z) = z/2$, which is a homeomorphism from $[-2, 2]$ to $[-1, 1]$, and $\tilde{F}_1(z) = 4z^3 - 3z$, which is a continuous map from $[-1, 1]$ to itself. We have that

$$\tilde{F}_1 \circ \psi_0(z) = 4(\frac{z}{2})^3 - \frac{3z}{2} = \frac{z^3}{2} - \frac{3z}{2} = \psi_0 \circ \tilde{F}(z).$$

Hence, $\tilde{F}$ is conjugate to $\tilde{F}_1$. Now consider $\psi(z) = \sin(\frac{\pi}{2} \cdot z)$, which is a homeomorphism from $[-1, 1]$ to $[-1, 1]$, and

$$\tilde{F}_2(z) = \begin{cases} 3z + 2, & x \in [-1, -1/3]; \\ -3z, & x \in (-1/3, 1/3); \\ 3z - 2, & x \in [1/3, 1], \end{cases}$$

which is a continuous map from $[-1, 1]$ to $[-1, 1]$. We have that for $z \in [-1, -1/3]$,

$$\tilde{F}_1 \circ \psi(z) = 4\sin^3(\frac{\pi}{2} \cdot z) - 3\sin(\frac{\pi}{2} \cdot z) = -\sin(\frac{3\pi}{2}z) = \sin(\frac{\pi}{2}(3z + 2)) = \psi \circ \tilde{F}_2(z).$$

Similarly, one can verify that $\tilde{F}_1 \circ \psi = \psi \circ \tilde{F}_2$ also holds on $(-1/3, 1/3)$ and $[1/3, 1]$. Hence, $\tilde{F}_2$ is topologically conjugate to $\tilde{F}$.

Note $\tilde{F}_2$ is a piecewise linear continuous map with slope equal to $\pm 3$. Hence, the topological entropy of $\tilde{F}_2$ is equal to $\log 3$ (De Melo and Van Strien, 2012, Corollary of Theorem 7.2). For a univariate map on a compact interval, positive topological entropy implies Devaney chaotic (Elaydi, 2007, Theorem 3.13). Due to the conjugacy, we have that $\tilde{F}$ is also Devaney chaotic. Also, conjugacy preserves topological entropy (Robinson, 1998, Theorem 1.7, Ch.8). Therefore, $h(\tilde{F}) = \log 3$.

We now show the existence of periodic orbit with any period. According to the Li-Yorke Theorem (Li and Yorke, 1975), a sufficient condition is that there exists a point $x$ such that $\tilde{F}_2^3(x) \leq x < \tilde{F}_2(x) < \tilde{F}_2^2(x)$. Consider $x = -5/7$. We have that $\tilde{F}_2(x) = -1/7$, $\tilde{F}_2^2(x) = 3/7$, and $\tilde{F}_2^3(x) = -5/7$. Due to the conjugacy, $\tilde{F}$ also has periodic orbit with any period and is Li-Yorke chaotic. This completes the proof. $\qquad\square$

We translate Proposition 18 to the original gradient system $\mathrm{GD}_\eta$.

**Proposition 19.** *Assume $|\mu| < 1$. We have that $h(\mathrm{GD}_\eta) \geq h(\mathrm{GD}_\eta|_{\partial \mathcal{D}'_\eta}) \geq \log 3$, and $\mathrm{GD}_\eta$ admits periodic orbits with any period.*

**Proof.** By Lemma 13, $\partial \mathcal{D}'_\eta$ is invariant under $\mathrm{GD}_\eta$. Since the map $T(u, v) = (z, w)$ is a semi-conjugacy between $\mathrm{GD}_\eta|_{\partial \mathcal{D}'_\eta}$ and $\tilde{F}$, we have that $h(\mathrm{GD}_\eta|_{\partial \mathcal{D}'_\eta}) \geq h(\tilde{F}) = \log 3$ (see, e.g., Robinson, 1998, Theorem 1.7, Ch 8), where the equality comes from Proposition 18. Meanwhile, since $\partial \mathcal{D}'_\eta$ is an invariant subset of $\mathbb{R}^{2d}$, we have that $h(\mathrm{GD}_\eta) \geq h(\mathrm{GD}_\eta|_{\partial \mathcal{D}'_\eta})$ (see, e.g., Vries, 2014, Proposition 8.1.7).

Next, we show the existence of all periodic orbits. As shown in Proposition 18, $\tilde{F}$ is conjugate to $\tilde{F}_2$:

$$\tilde{F}_2(z) = \begin{cases} 3z + 2, & x \in I_0 = [-1, -1/3]; \\ -3z, & x \in I_1 = (-1/3, 1/3); \\ 3z - 2, & x \in I_2 = [1/3, 1]. \end{cases} \tag{21}$$

Note $\tilde{F}_2$ is a piecewise linear map defined on $[-1, 1]$, with each piece mapping onto the whole interval $[-1, 1]$. With classical analyses in symbolic dynamics (see, e.g., Devaney and Eckmann, 1987, Ch 1.7), we have that $\tilde{F}_2$, when restricted to $[-1, 1] - \{x \in [-1, 1]: \tilde{F}_2^N(x) \neq \pm 1, \forall N\}$, is conjugate to the shift operator $\sigma$, defined on the following set

$$\{s \in \{0, 1, 2\}^{\mathbb{N}}: s \text{ is not eventually constant in } 0 \text{ or } 2\},$$

such that $\sigma(s) = \sigma(s_0 s_1 s_2 \cdots) = (s_1 s_2 \cdots)$. The conjugacy is given by the map $\zeta$:

$$\zeta(x) = (s_0 s_1 \cdots), \quad s_j = l \text{ if } \tilde{F}_2^j(x) \in I_l,$$

where $l \in \{0, 1, 2\}$ and $I_l$ is defined in (21).

Now we construct periodic orbits $(u_k, v_k)_{k \geq 0}$ with periodicity $K$ for arbitrary $K$. Consider $T(u_k, v_k) = (z_k, w_k)$. Consider new coordinates in $uv$-space, given by $(u + v, u - v)$, which preserves all dynamical behaviors of $\mathrm{GD}_\eta$. Notice that

$$\begin{aligned} u_{k+1} + v_{k+1} &= u_k - \eta z_k v_k + v_k - \eta z_k u_k = (1 - \eta z_k)(u_k + v_k), \\ u_{k+1} - v_{k+1} &= u_k - \eta z_k v_k - v_k + \eta z_k u_k = (1 + \eta z_k)(u_k - v_k). \end{aligned} \tag{22}$$

It follows that, during training $u_k + v_k$ and $u_k - v_k$ must lie in the one-dimensional space spanned by $u_0 + v_0$ and $u_0 - v_0$, respectively. Meanwhile, as shown in Proposition 8, if $T(u', v') = T(u, v)$, then $\|u' + v'\| = \|u + v\|$ and $\|u' - v'\| = \|u - v\|$. Also, it is clear that if $T(u', v') \neq T(u, v)$, $(u', v') \neq (u, v)$. Therefore, whenever $(z_k, w_k)_{k \geq 0}$ is a $K$-orbit and,

$$\# \{k \in \{0, \cdots, K - 1\} : 1 - \eta z_k < 0\}, \quad \# \{k \in \{0, \cdots, K - 1\} : 1 + \eta z_k < 0\}$$

are two even numbers, where $\#S$ is the number of elements in the set $S$, we have that $(u_k, v_k)_{k \geq 0}$ is a $K$-orbit. Notice that, under the conjugacy between $\tilde{F}$ and $\tilde{F}_2$, these two numbers correspond to the numbers of visits of the $\tilde{F}_2$-orbit to $I_2$ and $I_0$, i.e., the number of 2's and 0's appearing in the symbol $s$. For $K = 2$, take $(z_0, w_0) = (-2, 4 - 2\mu)$. Notice that $(-2, 4 - 2\mu)$ is a fixed point under $F$, and that, $1 - \eta z_0 - \eta\lambda > 0$ but $1 + \eta z_0 - \eta\lambda < 0$. Therefore, $(u_k, v_k)_{k \geq 0}$ is a 2-orbit. For $K = 3$, take $(z_0, w_0)$ be the element corresponding to $(001\,001\cdots)$, which is a 3-orbit in the symbolic system. As there are two 0's and zero 2's, $(u_k, v_k)_{k \geq 0}$ is a 3-orbit. For $K = 4$, take $(z_0, w_0)$ be the element corresponding to $(0011\,0011\cdots)$, and we have that $(u_k, v_k)_{k \geq 0}$ is a 4-orbit. For $K = 2N + 1$, take $(z_0, w_0)$ corresponding to the repetition of $2N$ 0's and one 1; for $K = 2N + 2$, take $(z_0, w_0)$ corresponding to the repetition of $2N$ 0's and two 1's. By this, we have that $\mathrm{GD}_\eta$ admits all periodic orbits, which completes the proof. $\qquad\square$

In the following, we show that, the original gradient descent system is not chaotic in the sense of Devaney when $d \geq 2$.

**Proposition 20.** *Assume that $\eta|y| < 1$ and $d \geq 2$. The system $\mathrm{GD}_\eta|_{\partial\mathcal{D}'_\eta}$ is not topological transitive.*

**Proof.** Let $(u_k, v_k)_{k \geq 0}$ be an orbit under $\mathrm{GD}_\eta$. Consider new coordinates $p_k = u_k + v_k$ and $q_k = u_k - v_k$ for $k \geq 0$. Recall (22) and notice that $z_k = u_k^\top v_k - y = -y + (\|p_k\|^2 - \|q_k\|^2)/4$. Thus, $(p_k, q_k)$ evolves autonomously. Since $(u, v) \mapsto (p, q)$ is a homeomorphism, the $uv$-system is topological transitive if and only if the $pq$-system is.

Notice that, if $4 = \eta(\|u\|_2^2 + \|v\|_2^2) - \eta^2 y(u^\top v - y)$, we have that

$$4 \geq \eta(\|u\|_2^2 + \|v\|_2^2) - \eta^2|y|\frac{\|u\|_2^2 + \|v\|_2^2}{2} = \eta(1 - \frac{\eta|y|}{2})(\|u\|_2^2 + \|v\|_2^2),$$

where we used that $|u^\top v| \leq (\|u\|_2^2 + \|v\|_2^2)/2$. Since $\eta|y| < 1$, we have that $\|u\|_2^2 + \|v\|_2^2 < 8/\eta$. Therefore,

$$(u,v) \in \partial \mathcal{D}'_\eta \Leftrightarrow \|u\|_2^2 + \|v\|_2^2 + \sqrt{(\|u\|_2^2 + \|v\|_2^2)^2 - 16y(u^\top v - y)} = \frac{8}{\eta}$$

$$\Leftrightarrow 4 = \eta(\|u\|_2^2 + \|v\|_2^2) - \eta^2 y(u^\top v - y) \tag{23}$$

$$\Leftrightarrow (u^\top \ v^\top) \begin{pmatrix} \eta I_d & -\eta^2 y/2 \cdot I_d \\ -\eta^2 y/2 \cdot I_d & \eta I_d \end{pmatrix} \begin{pmatrix} u \\ v \end{pmatrix} = 4 - \eta^2 y^2.$$

The last equation is a quadratic form and the eigenvalues of the coefficient matrix are $\eta \pm \frac{1}{2}\eta^2 y$, each with multiplicity $d$. Therefore, when $\eta|y| < 1$, the quadratic form is positive definite and defines a smooth ellipsoid of dimension $2d-1$. Notice that $(u,v) \mapsto (p,q) = (u+v, u-v)$ is a bijective linear transformation. Thus, $\partial \mathcal{D}'_\eta$ is still an ellipsoid in $pq$-coordinates.

Now fix any $(p_0, q_0) \in \partial \mathcal{D}'_\eta$ and an open neighborhood $U = U_p \times U_q$ of $(p_0, q_0)$ where $p_0 \in U_p \subset \mathbb{R}^d$ and $q_0 \in U_q \subset \mathbb{R}^d$. According to (22), we have that

$$\cup_{k=0}^\infty \mathrm{GD}_\eta^k(U) \subset \mathcal{C} \triangleq \{cp \colon p \in U_p, c \in \mathbb{R}\} \times \{dq \colon q \in U_q, d \in \mathbb{R}\} \subset \mathbb{R}^{2d}.$$

Clearly, when $d \geq 2$ and $U$ is small enough such that $(\mathbf{0}, \mathbf{0}) \notin U$, $\mathcal{C}$ is not dense in $\mathbb{R}^{2d}$ and there exists an open set $V$ such that $V \cap \partial \mathcal{D}'_\eta \neq \varnothing$ and $\mathcal{C} \cap V = \varnothing$. Therefore, $\mathrm{GD}_\eta|_{\partial \mathcal{D}'_\eta}$ is not transitive in $pq$-coordinates. This completes the proof. $\square$

We proceed to show that when the initialization is near the boundary, the orbit can visit any point in the state space.

**Proposition 21.** *Assume $|\mu| < 1$. Given any $(z^*, w^*) \in \Omega$ and any open set $O \subset \Omega$ such that $O \cap \{Q(z,w) = 8\} \neq \varnothing$, there exists $N \geq 0$ and $(z,w) \in O$ such that $F^N(z,w) = (z^*, w^*)$.*

**Proof.** As in Proposition 14, let $G_0 \colon \Omega \to \mathrm{cl}(A_0)$ denote the inverse of $F|_{\mathrm{cl}(A_0)}$. Let $m_0 = (z^*, w^*)$ and $m_k = G_0^k(m_0)$ for $k \geq 1$. We first show that $\lim_{k \to \infty} m_k = m^* = (-2, 4 - 2\mu)$. Note for all $k \geq 1$, $m_k \notin \{Q = 4|\mu|\}$. Therefore, by Lemma 13, we know that $Q(m_k)$ either stays at 8 or monotonically approaches 8. Hence, as we shown in the proof of Proposition 15, $m_k$ must converge to the compact set

$$Z = \{\Delta Q = 0\} = \{z = 0\} \cup \{w = \mu z + 4\} \cup \{w = 2\mathrm{sgn}(\mu)(z + \mu), w \leq 4|\mu|\}.$$

As $G_0(\Omega) \subset \mathrm{cl}(A_0)$, $m_k$ must converge to the set

$$Z \cap \mathrm{cl}(A_0) = \{z \leq -1, w = \mu z + 4\}.$$

It follows that the omega-limit set $\omega(m_0) \subset Z \cap \mathrm{cl}(A_0)$. Notice that, when restricted to $\{w = \mu z + 4\}$, the system $F$ reduces to $\tilde{F}(z) = z^3 - 3z$, $z \in [-2, 2]$. Then $G_0|_{Z \cap \mathrm{cl}(A_0)}$ corresponds to the branch of $\tilde{F}^{-1}$ whose image is $[-2, -1]$. Using the conjugacy between $\tilde{F}$ and $\tilde{F}_2$ as shown in Proposition 18, it's clear that all orbits under $G_0|_{Z \cap \mathrm{cl}(A_0)}$ converge to $z = -2$. Therefore, for any $q \in \omega(m_0)$, $G_0^N(q) \to (-2, 4 - 2\mu)$ as $N \to \infty$. Since the omega-limit set is invariant and closed, we have that $(-2, 4 - 2\mu) \in \omega(m_0)$, which implies that $m_k$ visits an arbitrarily small neighborhood of $(-2, 4 - 2\mu)$. Note, the eigenvalues of $JF(-2, 4 - 2\mu)$ are 9 and $5 - 2\mu$. Hence, $(-2, 4 - 2\mu)$ attracts all orbits of $G_0$ in some neighborhood of itself. Hence, $m_k \to m^*$ as $k \to \infty$.

Now consider any open set $O$ that satisfies $O \cap \{Q = 8\} = O \cap \{w = \mu z + 4\}$ is not empty. We show that there exists $x \in O$ and $n$ such that $F^n(x) = m^*$. Recall that $F|_{\{Q=8\}}$ is topologically conjugate to the piece-wise linear map $\tilde{F}_2$ as shown in Proposition 18. For $\tilde{F}_2$, we have that

$$\cup_{N=0}^\infty \tilde{F}_2^{-N}(\{\pm 1\}) = \cup_{k \geq 0}\{-1 + \frac{2j}{3^k} \colon j = 0, 1, \cdots, 3^k\},$$

which is dense in $[-1, 1]$. By symmetry arguments, we have that the preimage of $-1$ is dense. Therefore, using the conjugacy, we have that there exists $x \in O$ and $n$ such that $x \in F^{-n}(m^*)$, i.e., $F^n(x) = m^*$. Notice $\{Q = 8\} \subset \mathrm{cl}(B)$, by Proposition 14, we have that there exists $i_1, \cdots, i_{n_1} \in \{0, 1, 2\}$ such that $G_{i_{n_1}} \circ \cdots \circ G_{i_1}(m^*) = x$. Since the composition of $G_i$'s is continuous, there

exists a neighborhood $\tilde{O}$ of $m^*$ such that $G_{i_{n_1}} \circ \cdots \circ G_{i_1}(\tilde{O}) \subset O$. Since $m_k \to m^*$, there exists $n_2$ such that $m_{n_2} = G_0^{n_2}(m_0) \in \tilde{O}$. Taken together, we have that

$$G_{i_{n_1}} \circ \cdots \circ G_{i_1} \circ G_0^{n_2}(m_0) \triangleq \hat{m} \in O.$$

By the definitions of $G_i$'s as in Proposition 14, it follows that

$$F^{n_1+n_2}(\hat{m}) = m_0,$$

which completes the proof. $\qquad\square$

### E.1.2 PROOF OF THEOREM 1 AND THEOREM 2

*Proof of Theorem 1.* According to Proposition 7, any measure-zero event in system $F$ corresponds to a measure-zero event in system $\mathrm{GD}_\eta$. According to Proposition 9, the orbit of $\mathrm{GD}_\eta$ converges to $\{u^\top v = y\}$ if and only if the orbit of $F$ converges to $\{z = 0\}$, and the former converges to $(\mathbf{0}, \mathbf{0})$ if and only the latter converges to $(-y, 0)$. According to Proposition 16, when $|\mu| < 1$ and for almost all initializations $(z, w)$, the orbit converges to $\{z = 0\}$ if $Q(z, w) < 8$. Notice that $\mu = \eta y$ and due to the conjugacy (8),

$$Q(z,w) < 8 \Leftrightarrow \eta(\|u\|_2^2 + \|v\|_2^2) + \sqrt{\eta^2(\|u\|_2^2 + \|v\|_2^2)^2 - 16\eta y \cdot \eta(u^\top v - y)} < 8$$

$$\Leftrightarrow \|u\|_2^2 + \|v\|_2^2 + \sqrt{(\|u\|_2^2 + \|v\|_2^2)^2 - 16y \cdot (u^\top v - y)} < \frac{8}{\eta}.$$

Also, by Proposition 16, when $Q(z, w) > 8$ or $|\mu| > 1$, almost all initializations do not converge. This gives the critical step size (3).

We now show the sensitivity to initialization. Consider any open neighborhood $W \subset \mathbb{R}^{2d}$ such that $W \cap \partial \mathcal{D}'_\eta \neq \varnothing$. Notice that the Jacobian of the map $T$ drops rank if and only if $\{u = \pm v\}$. Also, as shown in (23), $\partial \mathcal{D}'_\eta$ defines a smooth ellipsoid of dimension $2d - 1$. Notice that, $\{u = \pm v\}$ is the union of two linear subspace with dimension $d$. Therefore, since $2d - 1 \geq d$ and an ellipsoid is curved everywhere, there always exists a point $\bar{\theta} \in W \cap \partial \mathcal{D}'_\eta \setminus \{u = \pm v\}$ and a neighborhood $\bar{W}$ of $\bar{\theta}$ such that $\bar{W} \subset W$ and $\bar{W} \cap \{u = \pm v\} = \varnothing$. The Jacobian of $T$ is full rank at all points in $\bar{W}$, so by constant rank theorem, $T(\bar{W})$ is an open set. Meanwhile, $T(\bar{\theta}) \in T(\partial \mathcal{D}'_\eta)$. Hence, under the conjugacy (8), we have $T(\bar{W}) \cap \{w = \mu z + 4\} \neq \varnothing$. According to Proposition 21, there exists $(z', w'), (z'', w'') \in T(\bar{W})$ such that $F^N(z', w')$ converges to $(0, w^*)$ with any $w^* \in [2|\mu|, \infty)$ and $F^N(z'', w'')$ converges to $(-\mu, 0)$, as $N$ tends to infinity. Therefore, there exists $\theta', \theta'' \in W$ such that $\mathrm{GD}_\eta^N(\theta')$ converges to a global minimizer with squared norm in $[2|y|, \infty)$ and $\mathrm{GD}_\eta^N(\theta'')$ converges to $(\mathbf{0}, \mathbf{0})$. Notice that

$$\|u\|^2 + \|v\|^2 \geq 2\|u\| \cdot \|v\| \geq 2|u^\top v|.$$

Therefore, the minimal squared norm at $\{u^\top v = y\}$ is $2|y|$. Also, notice that

$$\|uu^\top - vv^\top\|_F^2 = \mathrm{Tr}((uu^\top - vv^\top)(uu^\top - vv^\top))$$

$$= \|u\|^4 + \|v\|^4 - 2(u^\top v)^2$$

$$= (\|u\|^2 + \|v\|^2)^2 - 2\|u\|^2\|v\|^2 - 2(u^\top v)^2$$

$$\geq \frac{(\|u\|^2 + \|v\|^2)^2}{2} - 2(u^\top v)^2.$$

Hence, at any global minimizer, the imbalance is lower bounded by the squared norm. Hence, by what we have shown, arbitrarily large imbalance can be also attained by initializations in $\bar{W}$.

Finally, the lower bound for the topological entropy and the existence of periodic orbit of any period directly come from Proposition 19. This completes the proof. $\qquad\square$

Next, we present a basic property for the unregularized scalar factorization problem: the sharpness coincides with the squared norm at the set of global minimizers. This has been proved by Wang et al. (2022). Below we restate their results.

**Proposition 22** (Wang et al., 2022, Theorem F.2). *For the unregularized scalar factorization problem* (2), *the eigenvalues of the Hessian $\nabla^2 L$ are $\pm(u^\top v - y)$, each with multiplicity $d - 1$, and $\frac{1}{2}(\|u\|^2 + \|v\|^2 \pm \sqrt{(\|u\|^2 + \|v\|^2)^2 + 4(u^\top v - y)^2 + 8(u^\top v - y)u^\top v})$. Consequently, when $u^\top v = y$, we have that*

$$\lambda_{\max}(\nabla^2 L(u, v)) = \text{Tr}(\nabla^2 L(u, v)) = \|u\|^2 + \|v\|^2.$$

*Proof of Theorem 2.* Assume $E = (e, e') \subset [\gamma_{\min}, 2/\eta]$. Let $E' = (e\eta, e'\eta)$. We first prove that, in the $F$-system, there exists a positive-measure set of initializations that converge to $\{(0, w) : w \in E'\}$. Notice that

$$|\frac{z_{k+1}}{z_k}| = |z_k^2 + \mu z_k - w_k + 1| \le 1 \tag{24}$$

is equivalent to

$$w_k \ge z_k^2 + \mu z_k, \text{ and } w_k \le z_k^2 + \mu z_k + 2.$$

Notice that (i) the convex parabola $w = z^2 + \mu z + 2$ intersects the $w$-axis at $(0, 2)$, (ii) the level set $\{Q(z, w) = c\}$ are straight lines intersecting the $w$-axis at $(0, c/2)$. Therefore, there exists $\delta \in (0, 1), C \in (4|\mu|, 4)$ such that all points in

$$T = \{(z, w) \in \Omega : |z| < \delta, Q(z, w) < C\}$$

satisfies $|z_{k+1}/z_k| < \gamma$ for some constant $\gamma \in (0, 1)$. Notice that, this holds for all sufficiently small $\delta$ and thus we fix $\gamma$. For $(z_k, w_k) \in T$ we have that $|z_{k+1}| < \delta$. Meanwhile, since $Q(z_{k+1}, w_{k+1}) \le Q(z_k, w_k)$ by Lemma 13, we have that $(z_{k+1}, w_{k+1}) \in T$. It follows that if the initialization $(z_0, w_0)$ is in $T$, the entire trajectory remains within $T$. Hence, we have that $|z_k| \le |z_0|\gamma^k$ for all $k \ge 0$, and that

$$|w_{k+1} - w_k| = |z_k^2(w_k - 4) - 4z_k\mu| \le |z_k|^2|w_k - 4| + 4|\mu||z_k| \le C_1|z_k|$$

where $C_1$ is independent of $\delta$. We have that,

$$|w_\infty - w_0| \le C_1 \sum_{k=0}^{\infty} |z_k| \le \frac{C_1\delta}{1 - \gamma}.$$

Hence, $|w_\infty - w_0| \to 0$ as $\delta \to 0$. Then, there must exist a sufficiently small sub-interval $E'' \subset E'$ and a sufficiently small $\delta$ such that, for all initializations in

$$A = \{(z, w) \in \Omega : |z| < \delta, |w| \in E''\},$$

which is a positive-measure set, the trajectory converges to $\{(0, w) : w \in E'\}$.

Consider any open neighborhood $W \subset \mathbb{R}^{2d}$ such that $W \cap \partial \mathcal{D}'_\eta \ne \varnothing$. As shown in the above proof of Theorem 1, there always exits a point $\bar\theta \in W \cap \partial \mathcal{D}'_\eta \setminus \{u = \pm v\}$ and a neighborhood $\bar W$ of $\bar\theta$ such that $\bar W \subset W$, $T$ has full rank on $\bar W$, and $T(\bar W)$ is an open set satisfying $T(\bar W) \cap \{w = \mu z + 4\} \ne \varnothing$. According to Proposition 21, there exists $(z', w') \in T(\bar W)$ such that $F^N(z', w') \in A$. By continuity of $F^N$, there exists a neighbor $A'$ of $(z', w')$ satisfying $A' \subset T(\bar W)$ and $F^N(A') \subset A$. Since $T|_{\bar W} : \bar W \to T(\bar W)$ is continuous, $T|_{\bar W}^{-1}(A')$ is open in $\bar W$ and has positive measure. By the conjugacy (8) and what we have shown, for all points in $T|_{\bar W}^{-1}(A')$, the final norm lies in $E$. This completes the proof. $\square$

Lastly, we discuss the technical novelties in the proof of Theorem 1, in comparison with the techniques of Wang et al. (2022). While their analysis is also based on a careful study of the gradient descent system, it is restricted to a specific subset of the parameter space. In comparison, our approach is built on a global analysis of the system. The key ingredients are: (i) we identified an equivariance symmetry in $\text{GD}_\eta$ and the corresponding quotient dynamical system defined by the map $F$ (Proposition 9); (ii) we analyzed all the branches of the inverse map $F^{-1}$ (which is a multi-valued map) and carefully studied the inverse dynamics defined by $F^{-1}$ (Proposition 14 and Proposition 21); (iii) we used symbolic dynamics to show that $\text{GD}_\eta$, when restricted to the convergence boundary, is semi-conjugate to a Devaney chaotic system (Proposition 18) and that $\text{GD}_\eta$ admits periodic orbits with any period (Proposition 19).

### E.2 REGULARIZED PROBLEM

Similar to the previous section, preliminary results are first presented in Appendix E.2.1, and the proofs of Theorem 4 and Theorem 5 is given in Appendix E.2.2.

### E.2.1 PRELIMINARY RESULTS

Unless stated otherwise, we use $(z', w')$ to denote $F(z, w)$. We first show that when the step size is small enough, the quotient dynamics $F$ is predictable.

**Proposition 23.** *Assume $0 < \nu < 1 - |\mu|$. For almost all $(z, w) \in \Omega$, if $Q(z, w) < 8 - 4\nu$, we have that $F^N(z, w)$ converges to $T(\mathcal{M})$ as $N \to \infty$. If $Q(z, w) < 4 - 4\nu$, for any $(u, v)$ that satisfies $T(u, v) = (z, w)$, $\mathrm{GD}_\eta^N(u, v)$ converges to $p^-(u, v)$, as defined in Theorem 4, as $N \to \infty$.*

**Proof.** By Lemma 13, we have that $Q(F^N(z, w))$ monotonically decreases. Since $Q(F^N(z, w))$ is non-negative, it converges to some finite value. It follows that $F^N(z, w)$ converges to $\{\Delta Q = 0\} \cap \{Q < 8 - 4\nu\}$, which is equal to $\{Q = 4|\mu|\}$ according to Lemma 13. Note $\{Q = 4|\mu|\} = \{w = 2\mathrm{sgn}(\mu)(z + \mu), w \leq 4|\mu|\}$ by Lemma 11. The $w$-update under $F$ on $\{Q = 4|\mu|\}$ is given by

$$w' = \kappa(w) = w(\frac{w}{2} - 1 + v - |\mu|)^2.$$

When $|\mu| > \nu$, $\kappa(w)$ has two fixed points on $[0, 4|\mu|]$: $w = 0$ and $w = 2(|\mu| - \nu)$. It is straight forward to obtain that $w = 2(|\mu| - \nu)$ attracts all orbits on $[0, 4|\mu|]$ except the one with initial value $w = 0$. Note $w = 0$ corresponds to the saddle $(-\mu, 0)$, where the Jacobian of $F$ has eigenvalues: $(1 + \mu - \nu)^2$ and $(-1 + \mu + \nu)^2$. Hence, $(-\mu, 0)$ is a hyperbolic fixed point of $F$. Note also that, $w = 2(|\mu| - \nu)$ corresponds to $(-\mathrm{sgn}(\mu)\nu, 2(|\mu| - \nu))$, which is the only element of $T(\mathcal{M})$. The Jacobian of $F$ at this point has eigenvalues: $(-1 + 2\nu)^2$, and $1 - 2|\mu| + 2\nu$. Hence, $(-\mathrm{sgn}(\mu)\nu, 2(|\mu| - \nu))$ attracts nearby orbits. Then with the omega-limit set arguments similar to those in the proof of Proposition 16, we have that, for all initializations that do not lie in the basin of attraction of $(-\mu, 0)$:

$$\cup_{N=0}^\infty F^{-N}(O \cap \{w = -2\mathrm{sgn}(\mu)(z + \mu)\}),$$

where $O$ is a neighborhood of $(-\mu, 0)$, the orbit converges to $T(\mathcal{M})$. The above basin has measure-zero since the Jacobian of $F$ has full rank almost everywhere. Therefore, for almost all initializations with $Q(z, w) < 8 - 4\nu$, the orbit converges to $T(\mathcal{M})$. When $|\mu| \leq \nu$, $\kappa(w)$ has only one fixed point $w = 0$, which attracts all orbits $[0, 4|\mu|]$. Note in this case, $(-\mu, 0)$ attracts nearby orbits, since both eigenvalues of $JF$ have norm smaller than one. Then, with the omega-limit set arguments, we have that for all initializations with $Q(z, w) < 8 - 4\nu$, the orbit converges to $T(\mathcal{M})$.

Next, we identify the converged point under $\mathrm{GD}_\eta$ when $Q < 4 - 4\nu$. Without loss of generality, assume $y \geq 0$. Let $(z_k, w_k)_{k \geq 0}$ be the orbit and $T(u_k, v_k) = (z_k, w_k)$. Consider new coordinates $p = (u + v)/\sqrt{2}$ and $q = (u - v)/\sqrt{2}$. Note that when $y > \lambda$, the set of global minimizers is given by

$$\{u = v, \|u\|_2^2 = y - \lambda\} = \{q = 0, \|p\|^2 = 2(y - \lambda)\}.$$

Notice that

$$\sqrt{2} \cdot p_{k+1} = u_{k+1} + v_{k+1} = u_k - \eta z_k v_k - \eta \lambda u_k + v_k - \eta z_k u_k - \eta \lambda v_k$$
$$= (1 - \eta z_k - \eta \lambda)(u_k + v_k) = (1 - \eta z_k - \eta \lambda)\sqrt{2} \cdot p_k.$$

Under the conjugacy (8), we have

$$p_k = p_0 \Pi_{j=0}^{k-1}(1 - z_j - \nu). \tag{25}$$

Therefore, the converged point is either $(\sqrt{2(y - \lambda)}\frac{p_0}{\|p_0\|}, 0)$ or $(-\sqrt{2(y - \lambda)}\frac{p_0}{\|p_0\|}, 0)$. Since the global minimizer is given by $\{q = 0, \|p\|^2 = 2(y - \lambda)\}$, the former point is the minimal distance solution and the latter point is the maximal distance solution, under the $pq$-coordinates. Note the change of coordinates $p = (u + v)/\sqrt{2}$ and $q = (u - v)/\sqrt{2}$ is given by an orthogonal transformation which preserves distance. Therefore the same statement holds in the $uv$-coordinates.

Note that,

$$Q(z, w) = 4 - 4\nu \Leftrightarrow -2\mu z - 2(1 - \nu)^2 + (1 - \nu)w = 0.$$

The above line intersects with $\partial\Omega$ at $(1-\nu, 2(1-\nu+\mu))$ and $(\nu-1, -2(\nu-1+\mu))$. Therefore, for all initializations that satisfy $Q(z, w) < 4 - 4\nu$, we have $|z| < 1 - \nu$. Meanwhile, since $4 - 4\nu < 8 - 4\nu$, we have that the $\{Q < 4 - 4\nu\}$ is forward invariant by Lemma 13, i.e., $|z| < 1 - \nu$ holds on the entire orbit. This implies that, when $Q(z_0, w_0) < 4 - 4\nu$, we have that $1 \pm z_j - \nu > 0$ for all $j \geq 0$. By (25), the converged minimizer in $pq$-coordinate has to be $(\sqrt{2(y - \lambda)}\frac{p_0}{\|p_0\|}, 0)$, which is the minimal distance solution. This completes the proof. □

We now proceed to show the projected boundary $T(\partial\mathcal{D}''_\eta)$ is self-similar.

**Proposition 24** (Self-similarity). *Assume $0 < \nu < 1 - |\mu|$. The boundary $T(\partial\mathcal{D}''_\eta)$ is self-similar with degree three.*

**Proof.** We use $\mathcal{D}$ for $\mathcal{D}''_\eta$ for notation simplicity. First, we prove that $T(\partial\mathcal{D}) = \partial T(\mathcal{D})$. Notice that, by Proposition 9, orbit under $\mathrm{GD}_\eta$ converges to $\mathcal{M}$ or to the saddle if and only if the corresponding orbit under $F$ converges to $T(\mathcal{M})$ or to $(-\mu, 0)$, respectively. This implies that $T^{-1}(T(\mathcal{D})) = \mathcal{D}$ and $T(\mathcal{D}^c) = T(\mathcal{D})^c$. Since $T$ is continuous, we have that $\partial T^{-1}(T(\mathcal{D})) = \partial\mathcal{D} \subset T^{-1}(\partial T(\mathcal{D}))$. Hence, $T(\partial\mathcal{D}) \subset \partial T(\mathcal{D})$.

For the other direction, consider any point $y \notin T(\partial\mathcal{D})$. Then we have $T^{-1}(y) \cap \partial\mathcal{D} = \varnothing$. Note $T^{-1}(y)$ is connected and compact by Proposition 8. Thus, there exists an open neighborhood $O$ of $T^{-1}(y)$ such that $O \subset \mathcal{D}$ or $O \subset \mathcal{D}^c$. Hence, $T(O) \subset T(\mathcal{D})$ or $T(O) \subset T(\mathcal{D}^c) = T(\mathcal{D})^c$. We claim that for any $y'$ that is sufficiently close to $y$, $y' \in T(O)$. Notice that, if $y \notin \partial\Omega$, we have that $T^{-1}(y) \cap \{u = \pm v\} = \varnothing$. Thus, $JT$ is non-singular at $T^{-1}(y)$ and, by constant rank theorem, $T(O)$ contains a neighborhood of $y$, which gives the claim. If $y \in \partial\Omega$, without loss of generality, assume that $y = (\frac{w_0}{2} - \mu, w_0)$ for some $w_0 \geq 0$. Then $F^{-1}(y) = \{u = v, \|u\|^2 = w_0/2\}$. Consider any $(u', v') \in T^{-1}(y')$. As shown in the proof of Proposition 8, when $\|y' - y\|$ tends to zero, $\|u' - v'\|^2$ must tend to 0 and $\|u' + v'\|^2$ to $2w_0$. This implies that $(u', v')$ tends to $F^{-1}(y)$. Thus, if $y'$ is sufficiently close to $y$, we have $T^{-1}(y) \subset O$ and thus $y' \in T(O)$. Now, given that $T(O) \subset T(\mathcal{D})$ or $T(O) \subset T(\mathcal{D})^c$, we have $y \notin \partial T(\mathcal{D})$. Hence, $\partial T(\mathcal{D}) \subset T(\partial\mathcal{D})$ and $\partial T(\mathcal{D}) = T(\partial\mathcal{D})$.

Next, we prove that $F(\partial T(\mathcal{D})) = \partial T(\mathcal{D})$. Let $A = T(\mathcal{D})$. Notice that, by Proposition 23, $\{Q(z, w) < 8 - 4\nu\}$ is an attracting neighborhood of $T(\mathcal{M}) \cup \{(\mu, 0)\}$. Thus, the corresponding basin of attraction, $A$, is an open set due to the continuity of $F$. It follows that $\partial A \subset A^c \subset \{Q \geq 8 - 4\nu\}$. Now consider any point $x \in F^{-1}(\partial A)$. We claim that a small enough neighborhood $O$ of $x$ is mapped to a neighborhood of $F(x)$. Note the claim holds trivially if $x$ is a regular point of $F$. Assume that $\det JF(x) = 0$ and recall that the critical points of $F$ is given in (14). Since $F(x) \in \partial A \subset A^c$ and $F^{-1}(A^c) \subset A^c$, we have that $x \in A^c \subset \{Q \geq 8 - 4\nu\}$. Hence, as we shown in the proof of Proposition 14, $x \in \partial A_0$ or $x \in \partial A_2$, and the claim holds according to Proposition 14. Now, since $F(x) \in \partial A$, $F(O)$ contains a point $y \in A$ and a point $z \in A^c$. By the previous claim, $F^{-1}(y) \cap O \neq \varnothing$ and $F^{-1}(z) \cap O \neq \varnothing$. As $F^{-1}(A) \subset A$ and $F^{-1}(A^c) \subset A^c$, $x \in \partial A$ and $F^{-1}(\partial A) \subset \partial A$. Since $F$ is surjective by Proposition 14, we have $F \circ F^{-1}(\partial A) = \partial A$. It follows that

$$\partial A = F \circ F^{-1}(\partial A) \subset F(\partial A).$$

For the other direction, note that for any $y \in \partial A$, since $y \in \mathrm{cl}(A)$ and $F$ is continuous, we have that $F(y) \in \mathrm{cl}(F(A)) \subset \mathrm{cl}(A)$. Therefore, $y \in \mathrm{cl}(A) \setminus A^o = \mathrm{cl}(A) \setminus A$. Since $F^{-1}(A) \subset A$, $F(y) \notin A$. Then we have that $F(y) \in \mathrm{cl}(A) \setminus A = \partial A$ and $F(\partial A) \subset \partial A$. Therefore, $F(\partial A) = \partial A$.

Since $F(\partial A) = \partial A$ and $\partial A \subset \{Q \geq 8 - 4\nu\}$, Proposition 14 gives that

$$\partial T(\mathcal{D}''_\eta) = \cup_{k=0,1,2} G_i(\partial T(\mathcal{D}''_\eta)),$$

where $G_i$'s are homeomorphisms. As shown in Proposition 14, $G_i(\Omega^o) \cap G_j(\Omega^o)$ is empty whenever $i \neq j$. Therefore, $T(\partial\mathcal{D}''_\eta)$ is self-similar with degree three. This completes the proof. □

In the following, we show that $\mathcal{D}_\eta$ has an unbounded interior, up to a measure-zero set.

**Proposition 25** (Unboundedness). *When $\mu = 0, 0 \leq \nu < 1$, there exists $a, b > 0$ such that, for almost all initializations that lie in $\{(z, w) \in \Omega\colon |z| < a\exp(-bw)\}$, the orbit converges to the minimizer.*

**Proof.** Let $(z', w') = F(z, w)$. Note, when $\mu = 0$, the unique global minimizer of $L$ corresponds to $(0, 0)$. Let $\alpha = 1 - \nu$. Then $0 < \alpha < 1$. Assume that $|z| < a \exp(-bw)$ for some $a, b > 0$. We aim to show that $|z'| < a \exp(-bw')$. Notice that

$$
\begin{aligned}
|z'| &= |z^3 + (\alpha^2 - \alpha w)z| \\
&= |z|^3 + (\alpha^2 + \alpha w)|z| \\
&\leq a^3 \exp(-3bw) + a \exp(-bw)(\alpha^2 + \alpha w).
\end{aligned}
$$

Also, we have

$$
\begin{aligned}
w' &= w(z^2 + \alpha^2) - 4z^2 \alpha \\
&\leq (w + 4\alpha)a^2 \exp(-2bw) + w\alpha^2.
\end{aligned}
$$

Note $a \exp(-bw')$ decreases as $w'$ increases. Then,

$$
\begin{aligned}
&a \exp(-bw') > |z'| \\
\Leftarrow\ & a \exp\Big( - b\big((w + 4\alpha)a^2 \exp(-2bw) + w\alpha^2\big)\Big) > a^3 \exp(-3bw) + a \exp(-bw)(\alpha^2 + \alpha w) \\
\Leftarrow\ & \exp\Big( - b(w + 4\alpha)a^2 \exp(-2bw)\Big) \cdot \exp(-\alpha^2 bw) > a^2 \exp(-3bw) + \exp(-bw)(\alpha^2 + \alpha w) \\
\Leftarrow\ & \exp\Big( - b(w + 4\alpha)a^2 \exp(-2bw)\Big) > a^2 \exp((\alpha^2 - 3)bw) + \exp((\alpha^2 - 1)bw)(\alpha^2 + \alpha w) \\
\Leftarrow\ & 1 - b(w + 4\alpha)a^2 \exp(-2bw) > a^2 \exp((\alpha^2 - 3)bw) + \exp((\alpha^2 - 1)bw)(\alpha^2 + \alpha w).
\end{aligned}
\tag{26}
$$

Let

$$
p(w) = 1 - b(w + 4\alpha)a^2 \exp(-2bw), \quad q(w) = a^2 \exp((\alpha^2 - 3)bw) + \exp((\alpha^2 - 1)bw)(\alpha^2 + \alpha w).
$$

Note that

$$
\begin{aligned}
p'(w) &= -a^2 b \exp(-2bw)\Big(1 - 2b(w + 4\alpha)\Big) \\
&\propto 2bw - 1 + 8\alpha b.
\end{aligned}
$$

Therefore $p(w)$ decreases from $(-\infty, w_0)$ and increases on $[w_0, +\infty)$, where $w_0 = (1 - 8\alpha b)/(2b)$. Let $b > 1/(8\alpha)$. Then we have that

$$
\min_{w \in [0, +\infty)} p(w) = p(0) = 1 - 4\alpha a^2 b.
$$

Note also that

$$
\begin{aligned}
q'(w) &= \exp((\alpha^2 - 1)bw)\Big(a^2(\alpha^2 - 3)b \exp(-2bw) + (\alpha^2 - 1)b(\alpha^2 + \alpha w) + \alpha\Big) \\
&\propto a^2(\alpha^2 - 3)b \exp(-2bw) + (\alpha^2 - 1)b\alpha w + (\alpha^2 - 1)b\alpha^2 + \alpha \\
&\propto a^2(\alpha^2 - 3)b \exp(-2bw) + \Big((\alpha^2 - 1)\alpha w + (\alpha^2 - 1)\alpha^2\Big)b + \alpha.
\end{aligned}
$$

Note, that $a^2(\alpha^2 - 3)b \exp(-2bw) < 0$ always holds. Also, since $(\alpha^2 - 1)\alpha w + (\alpha^2 - 1)\alpha^2 \leq 0$ when $w \geq 0$, we can choose $b$ sufficiently large so that $q'(w) < 0$ holds on $[0, +\infty)$. Then

$$
\max_{w \in [0, +\infty)} q(w) = q(0) = a^2 + \alpha^2.
$$

Now fix $b$. Note $1 - 4\alpha a^2 b \to 1$ and $a^2 + \alpha^2 \to \alpha^2$ as $a \to 0$. We can always select $a$ small enough such that $q(0) < p(0)$. Then we have that $p(w) > q(w)$ holds for all $w \geq 0$ and hence (26) holds. Therefore, the set $\{|z| < a \exp(-bw)\}$ is forward invariant under $F$. Due to the exponential decay, we can always select $a$ small enough and $b$ large enough such that $\{|z| < a \exp(-bw)\} \subset \{Q(z, w) > Q(F(z, w))\}$, where the latter set is given in Lemma 13. Therefore, in this exponential cone, $Q$ decreases monotonically. Then using similar arguments to those in Proposition 23, we have that all almost all initializations in this cone converge to the minimizer. This completes the proof. $\square$

In the following, we show that when the initialization is near the boundary, the orbit can visit any point in the space.

**Proposition 26.** *Assume $0 \leq \nu < \min\{\frac{1}{2}, 1 - |\mu|\}$. Consider arbitrary point $m_0 = (z, w) \in \Omega$. When $\mu \geq 0$, $\lim_{N \to \infty} G_0^N(m_0) = (-2 + \nu, 4 - 2(\nu + \mu))$. When $\mu < 0$, $\lim_{N \to \infty} G_2^N(m_0) = (2 - \nu, 4 + 2(\mu - \nu))$.*

**Proof.** We prove the case $\mu \geq 0$. Note that the case of $\mu < 0$ can be proved via analogous procedures. Let $(z', w') = F(z, w)$ and $m_k = G_0^k(m_0)$ for $k \geq 1$. Consider the function $E(z, w) = w + 2(z + \mu)$. We have

$$
\begin{aligned}
E(F(z, w)) - E(z, w) &= w' + 2(z' + \mu) - w - 2(z + \mu) \\
&= (w + 2(z + \mu))(z + \nu - 2)(z + \nu).
\end{aligned}
\tag{27}
$$

Note for $(z, w) \in \Omega$, $w + 2(z + \mu) \geq 0$. Also, $2 - \nu > 0 > -\nu$. Then for $w > 2(z + \mu)$, we have that

$$
E(F(z, w)) - E(z, w) > 0 \Leftarrow z < -\nu.
$$

We have that $m_k \in \mathrm{cl}(A_0)$ for $k \geq 1$. Hence, the $z$-coordinate of $m_k$ is smaller than $\nu - 1$ for all $k \geq 1$. Since $\nu < 1/2$, $\nu - 1 < -\nu$. Hence, $m_k \in \{E(F(z, w)) > E(z, w)\}$ for all $k \geq 1$. Note $(m_k)_{k \geq 0}$ is a backward orbit. Thus, $E(m_k)$ monotonically decreases. Since $E$ is lower bounded by 0 on $\Omega$, $E(m_k)$ converges to some finite value $E^*$. For contradiction, assume $m_k$ is unbounded. Note that, given arbitrary $m_1 \in \mathrm{cl}(A_0)$, the set

$$
\{E(z, w) \leq E(m_1)\} \cap \{|z| < M\}
$$

is bounded for any $M > 0$. Hence, we have the $z$-coordinate of $m_k$ tends to negative infinity. Note that for sufficiently small negative $z$ and $(z, w) \in \Omega$, we have

$$
\begin{aligned}
w' > w &\Leftrightarrow 4z(z + \mu)(-1 + \nu) + w(z^2 + (-2 + \nu)\nu) > 0 \\
&\Leftarrow w > \frac{4z(z + \mu)(1 - \nu)}{z^2 + (-2 + \nu)\nu} \\
&\Leftarrow -2(z + \mu) > \frac{4z(z + \mu)(1 - \nu)}{z^2 + (-2 + \nu)\nu} \\
&\Leftarrow -(z^2 + (-2 + \nu)\nu) < 2z(1 - \nu) \\
&\Leftarrow z^2 + 2(1 - \nu)z + \nu(2 - \nu) > 0,
\end{aligned}
$$

which clearly holds as $z$ tends to $-\infty$. Therefore, $m_k$ must lie in the region $\{w' > w\}$ for all $k \geq K$ for some finite $K > 0$. Note this implies that the $w$-coordinate of $m_k$ starts to decrease from all $k \geq K$. This conflicts the fact that $m_k$ is unbounded, as $\Omega \cap \{w < C\}$ is bounded for any $C > 0$. Hence, $m_k$ is bounded.

According to (27), $m_k$ must converge to

$$
\mathrm{cl}(A_0) \cap \{(w + 2(z + \mu))(z + \nu - 2)(z + \nu) = 0\} = \mathrm{cl}(A_0) \cap \{(w + 2(z + \mu) = 0\}.
$$

Otherwise, assume $m_k \subset K$ for all $k$ and some compact set $K$. The function $E(F(z, w)) - E(z, w)$ is continuous, so if $m_k$ does not converge to its zero level set, $E(m_k) - E(m_{k-1})$ is bounded below and $E(m_k)$ can not converge.

Note, when restricting to $\{w = -2(z + \mu)\}$, the $w$-update under $F$ is given by

$$
w' = \kappa(w) = w(\frac{w}{2} - 1 + \nu + \mu)^2, \; w \geq 0.
$$

Solving $\kappa(w) = w$, we obtain that $\kappa$ has two fixed points on $w \geq 0$: $w = 0$ and $w = 4 - 2(\mu + \nu)$. It is straight forward to obtain that all backward orbits of $\kappa$ converges to $w = 4 - 2(\mu + \nu)$ except the one initialized at $w = 0$. Meanwhile, note that $w = 4 - 2(\mu + \nu)$ corresponds to $(\nu - 2, 4 - 2(\mu + \nu))$. The eigenvalues of the Jacobian of $F$ at this point are: $5 - 2\mu - 2\nu, (-3 + 2\nu)^2$. Therefore, the backward orbits of $F$, i.e., forward orbits of $G_0$, are locally attracted by $(\nu - 2, 4 - 2(\mu + \nu))$. Then, using omega-limit set arguments similar to those in the proof of Proposition 16, we have that $m_k$ converges to $(-2 + \nu, 4 - 2(\nu + \mu))$. This completes the proof. $\square$

Using Proposition 26, we show that for the gradient descent system, the converged minimizer is unpredictable when the initialization is near the boundary.

**Proposition 27.** *Assume $0 \leq \nu < \min\{\frac{1}{2}, 1 - |\mu|\}$. Consider $\xi_1 = (-2 + \nu, 4 - 2(\nu + \mu))$ and $\xi_2 = (2 - \nu, 4 + 2(\mu - \nu))$. For $i = 1, 2$, we have that $\cup_{N=0}^{\infty} F^{-N}(\xi_i)$ has infinitely many points and $\cup_{N=0}^{\infty} F^{-N}(\xi_i) \subset T(\partial \mathcal{D}''_\eta)$. When $y \geq 0$, for any open set $O$ such that $O \cap (\cup_{N=0}^{\infty} F^{-N}(\xi_1)) \neq \varnothing$, there exists $(z', w'), (z'', w'') \in O$ such that, for any $(u', v'), (u'', v'')$ that satisfy $T(u', v') = (z', w')$ and $T(u'', v'') = (z'', w'')$, we have $\mathrm{GD}_\eta^N(u', v')$ converges to $p^+(u', v')$ and $\mathrm{GD}_\eta^N(u'', v'')$ converges to $p^-(u'', v'')$. When $y < 0$, the same result holds for any open set $O$ such that $O \cap (\cup_{N=0}^{\infty} F^{-N}(\xi_2)) \neq \varnothing$.*

**Proof.** We prove the case $y \geq 0$. Note that the case of $y < 0$ can be proved via analogous procedures. We first show that $\cup_{N=0}^{\infty} F^{-N}(\xi_i)$ has infinitely many points and $\cup_{N=0}^{\infty} F^{-N}(\xi_i) \subset \partial T(\mathcal{D}''_\eta)$. Notice that, $\xi_1$ lies in the set $\{(z, w) \in \Omega : w = -2(z + \mu)\}$. By Proposition 12, this set is invariant under $F$, where the $w$-update under $F$ is given by

$$w' = \kappa(w) = w(\frac{w}{2} - 1 + \mu + \nu)^2.$$

By analyzing the one-dimensional cubic map $\kappa$, it is straight forward to obtain that, as $N \to \infty$, for all $w$ with $w < 4 - 2(\mu + \nu)$, $\kappa^N(w) \to 0$ and, for all $w$ with $w > 4 - 2(\mu + \nu)$, $\kappa^N(w) \to +\infty$. Note that $w$ converging to zero corresponds to $zw$-orbit converging to $(-\mu, 0)$ and $uv$-orbit converging to the $(\mathbf{0}, \mathbf{0})$. Therefore, we have $\xi_1 \in \partial T(\mathcal{D}''_\eta) = T(\partial \mathcal{D}''_\eta)$, where the equality was shown in the proof of Proposition 24. Note, $Q(\xi_1) = 8 - 4\nu > 6 - 4\nu$. Therefore, by Proposition 14, Therefore,

$$\cup_{N=0}^{\infty} F^{-N}(\xi_1) = \{G_{i_1} \circ \cdots \circ G_{i_k}(\xi_1) : \forall k \geq 1, i_j \in \{0, 1, 2\}, \forall j\}, \tag{28}$$

where $G_i$'s are homeomorphisms. By the construction of $G_i$, the cardinality of this set is infinity. Also, as each $G_i$ is a homeomorphism, any point in this set belongs to $T(\partial \mathcal{D}''_\eta)$.

Next, we show that for any open set $O$ such that $O \cap \cup_{N=0}^{\infty} F^{-N}(\xi_1) \neq \varnothing$, there exists $(z', w'), (z'', w'') \in O$ satisfying the claimed properties. When $y \leq \lambda$, $L$ has a unique minimizer $(\mathbf{0}, \mathbf{0})$ and the result holds according to Proposition 26. Now consider $y > \lambda$. Let $(u_k, v_k)_{k \geq 0}$ be a gradient descent orbit that converges to a global minimizer, and $(z_k, w_k) = T(u_k, v_k)$. As we shown in the proof of Proposition 23, the converged minimizer is the minimal distance solution if

$$\zeta(z_0, w_0) = \#\{j \geq 0 : 1 - z_j - \nu < 0\}$$

is an even number; and the converged minimizer is the maximal distance solution if the above is an odd number.

Let $m_* = (-\nu, 2(\mu - \nu))$, so that $\{m_*\} = T(\mathcal{M})$. Assume that $O \ni \{G_{i_1} \circ \cdots \circ G_{i_k}(\xi_1)\}$ for some fixed $i_1, \cdots, i_k$. By the continuity of $G_{i_1} \circ \cdots \circ G_{i_k}(\xi_1)$, there exists a neighborhood $\tilde{O} \ni \xi_1$ such that $G_{i_1} \circ \cdots \circ G_{i_k}(\xi_1)(\tilde{O}) \subset O$. By Proposition 26, there exists $N' > 0$ such that $G_0^{N'}(\Omega) \subset \tilde{O}$. Then we have that

$$(z', w') \triangleq G_{i_1} \circ \cdots \circ G_{i_k} \circ G_0^{N'}(m_*) \in O$$

and

$$(z'', w'') \triangleq G_{i_1} \circ \cdots \circ G_{i_k} \circ G_0^{N'} \circ G_2(m_*) \in O.$$

By construction, $F^{k+N'}(z', w') = m_*$ and $F^{k+N'+1}(z'', w'') = m_*$. Then it suffices to that one of $\zeta(z', w')$ and $\zeta(z'', w'')$ is odd and the other is even. To see this, notice that $\{w = -2(z + \mu)\}$ is forward-invariant. Then, as $m^* \notin \{w = -2(z + \mu)\}$, the orbit starting from $(z', w')$ and from $(z'', w'')$ can not visit $\{w = -2(z + \mu)\}$. Therefore, by noticing that $G_0(\Omega) \subset \{z < 1 - \nu\}$, $G_1(\Omega \setminus \{w = -2(z + \mu)\}) \subset \{z < 1 - \nu\}$, $G_2(\Omega \setminus \{w = -2(z + \mu)\}) \subset \{z > 1 - \nu\}$, and $m_* \in \{z < 1 - \nu\}$, we have that

$$\zeta(z', w') = \#\{1 \leq j \leq k : i_j = 2\}, \ \zeta(z'', w'') = \#\{1 \leq j \leq k : i_j = 2\} + 1.$$

This completes the proof. $\qquad\square$

### E.2.2 PROOFS OF THEOREM 4 AND THEOREM 5

*Proof of Theorem 4.* By Proposition 9, $\mathcal{S}_\eta = T^{-1}(T(\mathcal{S}_\eta))$, and $T(\mathcal{S}_\eta)$ is the basin of attraction of the point $(-\mu, 0)$ for system $F$. As shown in the proof of Proposition 23, $T(\mathcal{S}_\eta)$ has measure zero.

By Proposition 7, $\mathcal{S}_\eta$ also has measure zero. The projected boundary $T(\mathcal{D}''_\eta)$ is self similar with degree three by Proposition 24. The unboundedness is given by Proposition 25.

When $y \geq 0$, consider the set $H = T^{-1}(\cup_{N=0}^\infty F^{-N}(\xi_1))$, where $\xi_1$ is defined in Proposition 27. By Proposition 27 and since $T$ is surjective, $H$ has infinitely many elements. By Proposition 9, $T^{-1}(T(\partial\mathcal{D}''_\eta)) = \partial\mathcal{D}''_\eta$. By Proposition 27, $\cup_{N=0}^\infty F^{-N}(\xi_1) \subset T(\partial\mathcal{D}''_\eta)$. Together, we have that $H = T^{-1}(\cup_{N=0}^\infty F^{-N}(\xi_1)) \subset \partial\mathcal{D}''_\eta$.

Consider any open neighborhood $W \subset \mathbb{R}^{2d}$ such that $W \cap H \neq \varnothing$. We will show that $T(W)$ contains an open neighborhood $O$ such that $O \cap (\cup_{N=0}^\infty F^{-N}(\xi_1)) \neq \varnothing$. Notice the Jacobian of the map $T$ drops rank if and only if $u = \pm v$. If $W \cap \{u = \pm v\} = \varnothing$, then by constant rank theorem, $T$ is locally a projection, which gives the claim. If $W \cap \{u = \pm v\} \neq \varnothing$, then, without loss of generality, assume $W = B((u_0, u_0), \delta)$. Then $T(u_0, u_0) = (\|u_0\|^2, 2\|u_0\|^2)$. We show that for any point $(z', w') \in \Omega$ that is sufficiently close to $(\|u_0\|^2, 2\|u_0\|^2)$, there exists a preimage under $T$ in $W$. Note, as $T$ is surjective, there exists $(u', v')$ such that $T(u', v') = (z', w')$. Note whenever $(z', w')$ tends to $(\|u_0\|^2, 2\|u_0\|^2)$, we have $w' + 2z' = \|u' + v'\|^2$ tends to $4\|u_0\|^2$ and $w' - 2z' = \|u' - v'\|^2$ tends to 0. Therefore, $(u', v')$ tends to $\{u = v, \|u\| = \|u_0\|\}$. Note, the map $T$ is invariant under rotation. Therefore, with proper rotation, we can select $(u', v')$ such that, as $(z', w')$ tends to $(\|u_0\|^2, 2\|u_0\|^2)$, it tends to $\{u = v, u = u_0\} = (u_0, u_0)$. Thus, such $(u', v')$ lies in $W$. This gives the claim.

Finally, by Proposition 27, there exist $\theta', \theta'' \in W$ such that $\mathrm{GD}_\eta^N(\theta')$ converges to $p^+(\theta')$ and $\mathrm{GD}_\eta^N(\theta'')$ converges to $p^-(\theta')$. The case of $y < 0$ can be proved analogously using Proposition 27. This completes the proof. □

*Proof of Theorem 5.* The results directly come from Proposition 9 and Proposition 23. □

## F  NON-EXISTENCE OF CONTINUOUS DYNAMICAL INVARIANT

Consider the scalar factorization problems:

$$\min_{\theta=(u,v)} L(\theta) = \frac{1}{2}(uv - y)^2 + \frac{\lambda}{2}(u^2 + v^2), \tag{29}$$

where $\lambda \geq 0$ and $u, v, y \in \mathbb{R}$. We show that there is no simple quantity that remains invariant during training.

A *dynamical invariant* is a map defined on the parameter space of the model whose values remain unchanged along optimization trajectories. Formally, for gradient descent applied to problem (29), a map $I(u, v) \colon \mathbb{R}^2 \to \mathbb{R}^k$ with $k \geq 1$ is a $\delta$-approximate invariant if $\|I(\mathrm{GD}_\eta^N(\bar{u}, \bar{v})) - I(\bar{u}, \bar{v})\| \leq \delta$ holds for all $N \geq 1$ and initializations $(\bar{u}, \bar{v}) \in \mathbb{R}^{2d}$, where $\|\cdot\|$ is a norm on $\mathbb{R}^k$. When $\delta = 0$, $I$ becomes a strict invariant. Invariants and approximate invariants have been used extensively to analyze the optimization dynamics of gradient flow and gradient descent in non-convex optimization problems. Particularly, for problem (1) without regularization, the imbalance $I(U, V) = UU^\top - VV^\top$ is a well-known invariant of gradient flow (Du et al., 2018) and an approximate invariant of gradient descent with small step sizes (Arora et al., 2019; Ye and Du, 2021; Xu et al., 2023). In contrast, the following result shows that no simple invariants exist under large step sizes.

**Theorem 28** (Non-Existence of Simple Dynamical Invariants). *Consider gradient descent with step size $\eta$ applied to problem (29) with $0 \leq \lambda < \min\{(1/\eta) - |y|, 1/(2\eta)\}$. If $I(u, v) \colon \mathbb{R}^2 \to \mathbb{R}^k$ is a continuous $\delta$-approximate invariant, then $\sup_{(u,v),(u',v') \in \mathbb{R}^2} \|I(u, v) - I(u', v')\| \leq 2\delta$. Consequently, the only continuous invariants are the constant functions.*

**Proof.** We use the notation $F, \mu, \nu, z, w$ as stated in the conjugacy (8). Assume $I$ is a continuous $\delta$-approximate invariant. For any $\varepsilon > 0$, there exist $\theta', \theta''$ such that

$$\|I(\theta') - I(\theta'')\| > \sup_{(u,v),(u',v') \in \mathbb{R}^2} \|I(u, v) - I(u', v')\| - \varepsilon.$$

Without loss of generality, assume $y \geq 0$. Now fix any point $\theta \in T^{-1}(\lambda - 2/\eta, 4/\eta - 2(y + \lambda))$. Under the conjugacy (8), we have that $T(\theta) = (\nu - 2, 4 - 2(\mu + \nu))$. Then according to Proposition 26 for the regularized case and Proposition 21 for the unregularized case, in any neighborhood $O$ of

$T(\theta)$, there exists $\xi', \xi''$ and $N', N''$ such that $F^{N'}(\xi') = T(\theta')$ and $F^{N''}(\xi'') = T(\theta'')$. Using similar arguments as those in Appendix E.2.2 and in Appendix E.1.2, we have that there exists $\bar{\theta}', \bar{\theta}''$ such that $T(\mathrm{GD}_\eta^{N'}(\bar{\theta}')) = T(\theta')$ and $T(\mathrm{GD}_\eta^{N''}(\bar{\theta}'')) = T(\theta'')$. Next, we show that $\bar{\theta}'$ and $\bar{\theta}''$ can be chosen such that $\mathrm{GD}_\eta^{N'}(\bar{\theta}') = \theta'$ and $\mathrm{GD}_\eta^{N''}(\bar{\theta}'') = \theta''$. To see this, notice that, for any $(u, v), (s, t) \in \mathbb{R}^2$, $(u, v) = (s, t)$ if and only if $T(u, v) = T(s, t)$ and, the two pairs, $u + v$ and $s + t$, and, $u - v$ and $s - t$, have the same sign. Consider the change of coordinates $p = (u + v)/\sqrt{2}$ and $q = (u - v)/\sqrt{2}$. Let $(u_k, v_k)_{k \geq 0}$ denote an orbit under $\mathrm{GD}_\eta$. By direct computation, we have that

$$p_k = p_0 \Pi_{j=0}^{k-1}(1 - z_j - \nu).$$

Therefore, the sign of $p_{N'}$ is fully determined by whether

$$n_p = \# \left\{ j \in \{0, \cdots, N' - 1\} : 1 - \nu < z_j \right\}$$

is even or odd. Similarly, we have

$$q_k = q_0 \Pi_{j=0}^{k-1}(1 + z_j - \nu),$$

and the sign of $q_{N'}$ is fully determined by whether

$$n_q = \# \left\{ j \in \{0, \cdots, N' - 1\} : 1 - \nu > z_j \right\}$$

is even or odd. Notice that we can take $\xi'$ as follows

$$\xi' = G_0^{m_q} \circ G_2^{m_p}(T(\theta')),$$

where $G_0$ and $G_2$ are defined as in Proposition 14, and $m_p \in \{0, 1\}$, and $m_q$ is a sufficiently large number. Since the image of $G_0$ lies out side $\{z > 1 - \nu\}$, increasing $m_q$ does not affect $n_p$. Also, since the image of $G_2$ is contained in $\{z > 1 - \nu\}$, one can always select $m_p$ from $\{0, 1\}$ to make $n_p$ even or odd as needed. Now fix $m_p$. Since the image of $G_0$ is contained in $\{z < \nu - 1\}$, one can always select a sufficiently large $m_q$ to make $n_q$ even or odd as needed. Consequently, by appropriate choices of $m_p$ and $m_q$, the signs of $p_{N'}$ and $q_{N'}$ can be made arbitrary. This implies that, one can always select $\bar{\theta}'$ such that $\mathrm{GD}_\eta^{N'}(\bar{\theta}') = \theta'$. A similar statement holds for $\bar{\theta}''$.

Since $I$ is $\delta$-invariant, we have:

$$\|I(\bar{\theta}') - I(\bar{\theta}'')\| > \|I(\theta') - I(\theta'')\| - 2\delta > \sup_{(u,v),(u',v') \in \mathbb{R}^2} \|I(u, v) - I(u', v')\| - \varepsilon - 2\delta.$$

Notice that $\bar{\theta}', \bar{\theta}''$ can be arbitrarily close to $\theta$. Since $I$ is continuous at $\theta$ and $\varepsilon$ is arbitrary, we have that

$$\sup_{(u,v),(u',v') \in \mathbb{R}^2} \|I(u, v) - I(u', v')\| \leq 2\delta,$$

which completes the proof. $\qquad\square$

## G  GENERAL MATRIX FACTORIZATION

We present the extensions of the results in Section 3 to general matrix factorization.

In the following, we present the extension of Theorem 1 to unregularized matrix factorization.

**Theorem 29** (Unregularized Matrix Factorization). *Consider gradient descent with step size $\eta$ applied to problem* (1) *with $\lambda = 0$ and $d \geq d_y$. Let $Y = \mathrm{Diag}(y_1 \cdots, y_{d_y})$. Consider the set*

$$\mathcal{W} = \left\{ (U, V) \in \mathbb{R}^{2d \cdot d_y} : \langle u^i, u^j \rangle = \langle u^i, v^j \rangle = \langle v^i, v^j \rangle = 0, \ \forall i \neq j \right\}, \tag{30}$$

*where $u^i, v^i$ denote the ith column of matrices $U, V$. Assume the initialization $(\bar{U}, \bar{V}) \in \mathcal{W}$. The following holds:*

- *Critical Step Size: Define the critical step size*

$$\eta^*(\bar{U}, \bar{V}) = \min_i \min \left\{ \frac{1}{|y_i|}, \frac{8}{\|\bar{u}^i\|_2^2 + \|\bar{v}^i\|_2^2 + \sqrt{(\|\bar{u}^i\|_2^2 + \|\bar{v}^i\|_2^2)^2 - 16y_i((\bar{u}^i)^\top \bar{v}^i - y_i)}} \right\}.$$

*For almost all initializations (under surface measure on $\mathcal{W}$), the algorithm converges to a global minimum if $\eta < \eta^*(\bar{U}, \bar{V})$, and it does not converge to a global minimum if $\eta > \eta^*(\bar{U}, \bar{V})$. Therefore, when $\eta$ satisfies $\eta\|Y\|_2 < 1$, the convergence region restricted to $\mathcal{W}$, $\mathcal{D}_\eta \cap \mathcal{W}$, is equal almost everywhere (under surface measure on $\mathcal{W}$) to the following set:*

$$\mathcal{D}'_\eta = \left\{ (U, V) \in \mathcal{W}: \|u^i\|_2^2 + \|v^i\|_2^2 + \sqrt{(\|u^i\|_2^2 + \|v^i\|_2^2)^2 - 16y((u^i)^\top v^i - y_i)} < \frac{8}{\eta}, \ \forall i \right\}.$$

- **Sensitivity to Initialization**: *Fix a step size $\eta$ that satisfies $\eta\|Y\|_2 < 1$. Given arbitrary $\theta \in \partial\mathcal{D}'_\eta$ (here boundary is taken with respect to the subspace topology on $\mathcal{W}$), $\varepsilon > 0$ and $K_1, K_2 > 0$, there exist $\theta', \theta'', \theta''' \in B(\theta, \varepsilon)$ such that, as $N$ tends to infinity, $\mathrm{GD}_\eta^N(\theta')$ converges to a global minimizer with norm larger than $K_1$, $\mathrm{GD}_\eta^N(\theta'')$ converges to a global minimizer with $\|UU^\top - VV^\top\|_F > K_2$, and $\mathrm{GD}_\eta^N(\theta''')$ converges to a stationary point, which is saddle point when $\min\{|y_i|\} > 0$.*

- **Trajectory Complexity**: *Assume $\eta\|Y\|_2 < 1$. The topological entropy of the gradient descent system $\mathrm{GD}_\eta$ satisfies $h(\mathrm{GD}_\eta) \geq \log 3$. Moreover, $\mathrm{GD}_\eta$ has periodic orbits of any positive integer period.*

All of the above results follow directly from Theorem 1 and Proposition 31. We remark that, for a dynamical system $F: X \to X$, if $S \subset X$ is an invariant set, i.e., $F(S) \subset S$, then we have $h(F) \geq h(F|_S)$. This gives the result for topological entropy.

We now present the extensions of Theorem 4 and Theorem 5 to regularized matrix factorization.

**Theorem 30** (Regularized Matrix Factorization). *Consider gradient descent with step size $\eta$ for problem (4). Let $Y = \mathrm{Diag}(y_1, \cdots, y_{d_y})$. Assume that $0 < \lambda \leq \min_{i=1,\cdots,d_y}\{(1/\eta) - |y_i|, 1/(2\eta)\}$. Let $\mathcal{W}$ be defined as in (30). Assume the initialization $(\bar{U}, \bar{V}) \in \mathcal{W}$. Consider the map $T_i(U, V) = ((u^i)^\top v^i, \|u^i\|_2^2 + \|v^i\|_2^2)$. Let $\mathcal{S}_\eta$ denote the set of initializations $(U, V)$ that converges to $(\mathbf{0}, \mathbf{0})$. Let $\mathcal{D}''_\eta = \mathcal{D}_\eta \cup \mathcal{S}_\eta$. The following holds:*

- **Self-similarity:** *For any $i \in \{1, \cdots, d_y\}$, $T_i(\partial(\mathcal{D}''_\eta \cap \mathcal{W}))$ is self-similar with degree three (here boundary is taken with respect to the subspace topology on $\mathcal{W}$).*

- **Unboundedness:** *When $Y = 0$, there exist constants $a, b > 0$ such that almost all initializations $(\bar{U}, \bar{V}) \in \mathcal{W}$ (under surface measure on $\mathcal{W}$) with $|(\bar{u}^i)^\top \bar{v}^i| < a\exp(-b(\|\bar{u}^i\|_2^2 + \|\bar{v}^i\|_2^2))$ for all $i \in \{1, \cdots, d_y\}$ converge to a global minimizer.*

- **Sensitivity to Initialization:** *Let $(u_t^i, v_t^i)_{t \geq 0}$ denote the gradient descent trajectory of the pair $(u^i, v^i)$, with $(u_0^i, v_0^i) = (\bar{u}^i, \bar{v}^i)$. Let $\mathcal{M}_i$ denote the set of global minimizers for the scalar problem $L_i(u, v) = \frac{1}{2}(u^\top v - y_i)^2 + \frac{\lambda}{2}(\|u\|_2^2 + \|v\|_2^2)$. Then $\mathcal{M} \cap \mathcal{W} = \mathcal{M}_1 \times \cdots \times \mathcal{M}_{d_y}$, where $\mathcal{W}$ denotes the set of global minimizers for problem (4). We have that, for any $(U, V) \in \mathcal{D}_\eta \cap \mathcal{W}$ and any $i \in \{1, \cdots, d_y\}$, as $t$ tends to infinity, $(u_t^i, v_t^i)$ converges either to*

$$p^-(u_0^i, v_0^i) = \arg\min_{(u,v) \in \mathcal{M}_i} \|(u, v) - (u_0^i, v_0^i)\|^2,$$

*or to*

$$p^+(u_0^i, v_0^i) = \arg\max_{(u,v) \in \mathcal{M}_i} \|(u, v) - (u_0^i, v_0^i)\|^2.$$

*Moreover, there exist infinitely many points on $\partial(\mathcal{D}''_\eta \cap \mathcal{W})$ (here boundary is taken with respect to the subspace topology on $\mathcal{W}$) such that for any open set $O$ containing such a point, there exist $i \in \{1, \cdots, d_y\}$, $(U', V'), (U'', V'') \in O$ such that, as $t$ tends to infinity, $(u_t^{i,'}, v_t^{i,'})$ converges to $p^-(u_0^{i,'}, v_0^{i,'})$ and $(u_t^{i,''}, v_t^{i,''})$ converges to $p^+(u_0^{i,''}, v_0^{i,''})$.*

- **Stable Dynamics Under Small Step Size**: *Consider the function*

$$Q(u, v) = \|u\|_2^2 + \|v\|_2^2 + \sqrt{(\|u\|_2^2 + \|v\|_2^2)^2 - 16y(u^\top v - y)}.$$

*Then the following holds for almost all initializations $(\bar{U}, \bar{V}) \in \mathcal{W}$ (under surface measure on $\mathcal{W}$): If $\eta < \min_{i=1,\cdots,d_y} 8/(4\lambda + Q(\bar{u}^i, \bar{v}^i))$, then gradient descent converges to a global minimizer; If $\eta < \min_{i=1,\cdots,d_y} 4/(4\lambda + Q(\bar{u}^i, \bar{v}^i))$, then for all $i$, $(u_t^i, v_t^i)$ converges to $p^-(u_0^i, v_0^i)$.*

All of the above results follow directly from Theorem 4, Theorem 5 and Proposition 31. We remark that, while the above Theorems are presented for initializations in $\mathcal{W}$, chaotic phenomena are observed under generalization initializations. Experiments are provided in Appendix I.

**Proposition 31.** *Consider gradient descent with step size $\eta$ for problem* (1) *with $d \geq d_y$. Consider the set $\mathcal{W} = \left\{(U, V) \in \mathbb{R}^{2d \cdot d_y} \colon \langle u^i, u^j \rangle = \langle u^i, v^j \rangle = \langle v^i, v^j \rangle = 0, \ \forall i \neq j \right\}$, where $u^i, v^i$ denote the ith column of matrices $U, V$. The set $\mathcal{W}$ is forward-invariant, i.e., $\mathrm{GD}_\eta(\mathcal{W}) \subset \mathcal{W}$. Moreover, if the initialization $(\bar{U}, \bar{V}) \in \mathcal{W}$, then for $i = 1, \cdots, r$, the trajectory of the columns $(u^i, v^i)$ is identical to the trajectory of gradient descent applied to the scalar factorization problem $L_i(u, v) = \frac{1}{2}(u^\top v - y_i)^2 + \frac{\lambda}{2}(\|u\|_2^2 + \|v\|_2^2)$, with step size $\eta$ and initialization $(\bar{u}^i, \bar{v}^i)$.*

*Proof of Proposition 31.* Recall the update:

$$U_{t+1} = U_t - \eta V_t(V_t^\top U_t - Y^\top) - \eta\lambda U_t, \quad V_{t+1} = V_t - \eta U_t(U_t^\top V_t - Y) - \eta\lambda V_t.$$

For $(U_t, V_t) \in \mathcal{W}$, we have that

$$U_{t+1} = U_t - \eta V_t V_t^\top U_t + \eta V_t Y^\top - \eta\lambda U_t$$
$$= U_t - \eta \sum_{k=1}^{d_y} v_t^k (v_t^k)^\top \cdot U_t + \eta V_t Y^\top - \eta\lambda U_t.$$

Therefore, for $j = 1, \cdots, d_y$,

$$u_{t+1}^j = u_t^j - \eta \sum_{k=1}^{d_y} v_t^k (v_t^k)^\top u_t^j + \eta y_j v_t^j - \eta\lambda u_t^j$$
$$= u_t^j - \eta v_t^j (v_t^j)^\top u_t^j + \eta y_j v_t^j - \eta\lambda u_t^j$$
$$= u_t^j - \eta\big((v_t^j)^\top u_t^j - y_j\big)v_t^j - \eta\lambda u_t^j.$$

Therefore, the one-step $u^j$-update aligns with that in scalar factorization problem. Similarly, we can show this holds for $v^j$-update. Now it suffices to verify that $\mathcal{W}$ is forward invariant. Assume $(U_t, V_t) \in \mathcal{W}$. Notice that both $u_{t+1}^j$ and $v_{t+1}^j$ are linear combinations of $u_t^j$ and $v_t^j$. Then it clear that

$$\langle u_{t+1}^j, u_{t+1}^k \rangle = \langle u_{t+1}^j, v_{t+1}^k \rangle = \langle v_{t+1}^j, v_{t+1}^k \rangle = 0$$

whenever $j \neq k$. This completes the proof. $\square$

The gradient descent update map $\mathrm{GD}_\eta$ is non-invertible in general. Nevertheless, we show that the parameter space can be partitioned into small pieces, so that when restricted to each piece, $\mathrm{GD}_\eta$ has a simple behavior.

*Proof of Proposition 6.* For notational simplicity, we let $G = \mathrm{GD}_\eta$. Under the assumptions, $G$ is a map with polynomial coordinates and $\det JG$ is a polynomial. Then either $\det JG$ is the zero function or it has a measure-zero zero locus. To reject the first case, it suffices to have that $\det JG(0) = \det(I - \eta\nabla^2 L(0)) \neq 0$. Note $\nabla^2 L(0)$ is a fixed positive semi-definite matrix. Then if $\eta$ is not the inverse of one of the eigenvalues of $\nabla^2 L(0)$, we have $\det JG(0) \neq 0$. Therefore, for all $\eta > 0$ except for finitely many values, $\det JG$ has a measure-zero zero locus. For such $\eta$, $G$ is a non-constant map. By Jelonek (2002), there exists a semi-algebraic, measure-zero set $S \subset \mathbb{R}^p$ such that, $G|_{\mathbb{R}^p \setminus G^{-1}(S)} \colon \mathbb{R}^p \setminus G^{-1}(S) \to \mathbb{R}^p \setminus S$ is a proper map. Let $S' = G(\{\det JG = 0\})$ denote the set of critical values of $G$. Then $S'$ has measure zero by Sard's theorem and is also semi-algebraic. Since $\det JG$ is non-zero almost everywhere, by Ponomarev (1987), $\mathcal{K}_\eta = G^{-1}(S) \cup G^{-1}(S')$ is a measure-zero set. Since $\mathcal{K}_\eta$ is semi-algebraic, $\mathbb{R}^p \setminus \mathcal{K}_\eta$ has finitely many connected components. Fix a connected component $\mathcal{C}$. For any compact set $K \subset G(\mathcal{C})$, since $K \cap S = \varnothing$, $(G|_{\mathbb{R}^p \setminus G^{-1}(S)})^{-1}(K)$ is compact. Meanwhile, since $\partial \mathcal{C} \subset G^{-1}(S) \cup G^{-1}(S')$ and $K \cap (S \cup S') = \varnothing$, $(G|_{\mathcal{C}})^{-1}(K) = (G|_{\mathbb{R}^p \setminus G^{-1}(S)})^{-1}(K) \cap \mathrm{cl}(\mathcal{C})$ is compact. Therefore, $G|_{\mathcal{C}}$ is a proper map between connected manifolds that has full-rank Jacobian everywhere. Hence, $G|_{\mathcal{C}}$ is a smooth covering map (see, e.g., Lee, 2012). This completes the proof. $\square$

## H  EXPERIMENT DETAILS

For Figure 1 left panel, we consider the problem $L(u,v) = (u^\top v - 1)^2 + 0.3(\|u\|_2^2 + \|v\|_2^2)$ with $(u,v) \in \mathbb{R}^{10}$. We randomly sampled two orthogonal unit vectors in $\mathbb{R}^{10}$. Viewing the two vectors as new axes, we evenly sampled $600^2$ initial points in the range $[-4,4]^2$. We then ran gradient descent with step size 1 for 1000 iterations. The training stops if the loss is below $L_{\min} + 10^{-6}$, where $L_{\min}$ is the global minimum or if it is above 100. For Figure 1 right panel, we consider $L(u,v) = (uv - 1)^2$ with $(u,v) \in \mathbb{R}^2$. We evenly sampled $800^2$ initial points in the range $[-4.5, 4.5]^2$. We ran gradient descent with step size 0.2 for 6 iterations and recorded the final squared distances to the two minimizers, $m_1 = (1,1)$ and $m_2 = (2.9, 1/2.9)$. Viewing the final distances as functions of the initial point, we used the "contourf" function from the Matplotlib package (version 3.5.2) to draw the sublevel sets of the distances. For the minimizer $m_1$, we drew the sublevel set of $[0, 0.15)$ to get the preimage of $\mathrm{GD}^{-6}(B(m_1, \sqrt{0.15}))$. For the minimizer $m_2$, we drew the sublevel set of $[0, 0.25)$ to get the preimage of $\mathrm{GD}^{-6}(B(m_2, \sqrt{0.25}))$.

For Figure 2, we consider $L(u,v) = (uv - 1)^2$ with $(u,v) \in \mathbb{R}^2$. For the left panel, we evenly sampled $800^2$ initial points in the range $[-4.5, 4.5]^2$ and ran gradient descent with step size 0.2 for 6 iterations. To visualize the basin for unstable minimizers, note, as shown in Corollary 17, converging to unstable minimizers can only occur within finitely many steps. We therefore recorded the final loss value and used the "contour" function from the Matplotlib package (version 3.5.2) to collect points in the level set of 0 for the loss. Those points correspond to convergence to a global minimizer within 6 or less steps. We then filtered out and visualized points that converge to an unstable global minimizer, i.e., a minimizer with squared norm larger than $2/\eta$ (see Corollary 17).

In Figure 2, to visualize the basin for the saddle $(\mathbf{0}, \mathbf{0})$, note, as shown in the proof of Proposition 16, this basin can be given by $\cup_{N=0}^{\infty} F^{-N}(O \cap \{u = -\mathrm{sgn}(y)v\})$ for some neighborhood of $(\mathbf{0}, \mathbf{0})$. Then we also recorded the final distance to the set $\{u = -v\}$ and used the "contour" function from the Matplotlib package (version 3.5.2) to collect points in the level set of 0 for the distance. Then we filtered out the points that lie in $\mathcal{D}_\eta'$ (as defined in Theorem 1). This yield the basin associated with the saddle. To justify this procedure, note, as shown in Proposition 15, any point outside $\mathcal{D}_\eta'$ either converges to a minimizer within finite steps or diverges. Also note, by the analysis in Lemma 12, points on $\{u = -\mathrm{sgn}(y)v\}$ either converge to the saddle or diverge. For the right panel of Figure 2, we evenly sampled $800^2$ points in $[-0.9, -0.6] \times [-4.55, -4.25]$. We ran gradient descent with step size 0.2 for 250 iterations. The training stops if the loss value is below $10^{-8}$ or above 100.

For Figure 3, we consider $L(u,v) = (uv - 0.5)^2/2 + 0.1(u^2 + v^2)$ where $(u,v) \in \mathbb{R}^2$. For the left panel, we consider the dynamical system defined by $F$ (see Proposition 3) with $\eta = 1, \lambda = 0.2$ and $y = 1$. In the $zw$-space, we evenly sampled $2000^2$ initial points in $[-2.5, 3] \times [0, 10]$ and filtered out those in $\{w \geq 2|z|\}$. We applied $F^{200}$ to those sampled points and filtered out initial points that lead to loss value below $L_{\min} + 10^{-5}$ where $L_{\min}$ is the global minimum of $L$. Those points come from the projected convergence region $T(\mathcal{D}_\eta'')$. Then we used the "ndimage.binary_erosion" function from the SciPy package (version 1.9.1) to find the boundary of those points. The coloring of the boundary is based on the preimage structure of $F$, which is described in Proposition 14. For the middle panel, we evenly sampled $800^2$ initial points in $[-4,4]^2$ and ran gradient descent for 100 iterations. For the right panel, we estimated the box-counting dimension for the boundary points found in the left panel. We first normalized these points to fit within $[0,1]^2$. We then computed the number of boxes $N(\epsilon)$ needed to cover all the points, with the box width $\varepsilon$ ranging from $1/2^2$ to $1/2^8$. We then performed linear regression on $\log N(\varepsilon)$ versus $\log(1/\varepsilon)$.

## I  ADDITIONAL EXPERIMENTS ON MATRIX FACTORIZATION

**The folding behavior of GD in scalar factorization**  In Figure 6, we illustrate the folding behavior of the map $\mathrm{GD}_\eta$ in $L(u,v) = (uv - 1)^2/2$ with $(u,v) \in \mathbb{R}^2$ and $\eta = 0.2$. The map $\mathrm{GD}_\eta$ is a 3-covering map from the light blue region $\mathcal{C}$ in the left panel to the light orange region $\mathrm{GD}_\eta(\mathcal{C})$ in the middle panel. Notice that $\mathcal{C} \subset \mathrm{GD}_\eta(\mathcal{C})$ and $\mathcal{C}$ contains the convergence boundary (black ellipsoid). The right panel shows that $\mathrm{GD}_\eta$ is transitive on the boundary: the trajectory of an initialization on this boundary appears to wander along it in a seemingly random manner. For theoretical justifications of these behaviors, see Proposition 14 and Proposition 18.

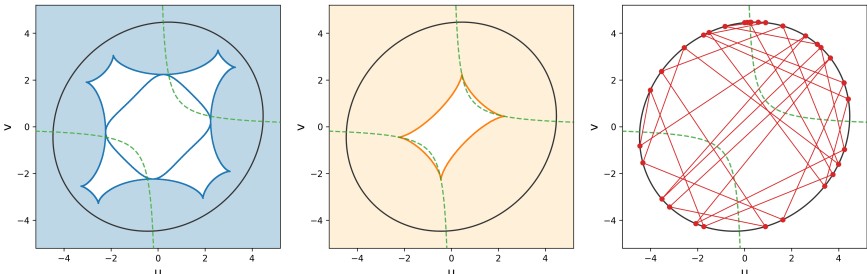

Figure 6: Folding behavior of the map $\mathrm{GD}_\eta$ in $L(u,v) = (uv-1)^2/2$ with $(u,v) \in \mathbb{R}^2$ and $\eta = 0.2$. In all panels, the black ellipsoid is the convergence boundary and the green hyperbola the set of global minimizers. The orange line in the middle panel is the set $\mathrm{Crit}$, which consists of all critical values of $\mathrm{GD}_\eta$. The blue lines in the left panel are the set $\mathrm{GD}_\eta^{-1}(\mathrm{Crit})$, i.e., the preimage of critical values. The map $\mathrm{GD}_\eta$ is a 3-covering map from the light blue region in the left panel to the light orange region in the middle panel. The right panel shows the training trajectory for an initialization on the convergence boundary.

**Comparison with Wang et al. (2022)**    Figure 7 provides a visual comparison between our Theorem 1 and Wang et al. (2022, Theorem 3.1), for the objective $L(u,v) = (uv-y)^2/2$ with $(u,v) \in \mathbb{R}^2$. Recall that the convergence condition provided in the work of Wang et al. (2022) is:

$$\eta < \eta_1^*(\bar{u}, \bar{v}) = \min\left\{\frac{1}{3|y|}, \frac{4}{\|\bar{u}\|^2 + \|\bar{v}\|^2 + 4|y|}\right\}. \tag{31}$$

The left penal of Figure 7 shows the case $\eta = 1, y = 0.3$. Here, the first condition in (31) is satisfied. As shown, initializations satisfying the second condition $\eta < 4/(u^2 + v^2 + 4|y|)$ form a strict subset of the convergence region $\mathcal{D}'_\eta$ in Theorem 1. The right panel shows the case $\eta = 1, y = 0.9$. Note that this setting falls outside the analysis of Wang et al. (2022), as $\eta > 1/(3|y|)$. Meanwhile, initializations satisfying $\eta < 4/(u^2 + v^2 + 4|y|)$ form an even smaller subset of $\mathcal{D}'_\eta$.

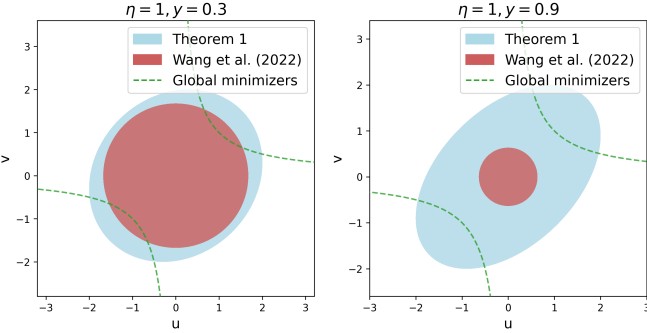

Figure 7: Comparison between Theorem 1 and Wang et al. (2022, Theorem 3.1). Gradient descent with step size $\eta = 1$ is applied to $L(u,v) = (uv-y)^2/2$ with $(u,v) \in \mathbb{R}^2$. In both panels, the light blue region is the convergence region $\mathcal{D}'_\eta$ described in Theorem 1, and the red region is the set of initializations satisfying $\eta < 4/(u^2 + v^2 + 4|y|)$, a conditions required by Wang et al. (2022).

**How convergence boundary and basin of saddle evolve with $\lambda$**    In Figure 8, we illustrate how the convergence boundary and the basin of attraction of the saddle point evolve as the regularization parameter $\lambda$ increases in the scalar factorization problem. As shown in the figure, and consistent with our theoretical results, the convergence boundary is smooth (in the almost everywhere sense) when $\lambda = 0$. When $\lambda$ is just above zero, the boundary is close to a smooth and bounded set, with the fractal spikes so thin that they are barely visible. As $\lambda$ increases, the fractal structure becomes more pronounced, and the spikes gradually get wider. Also, the basin of attraction of the saddle does not separate points inside the convergence region from points outside.

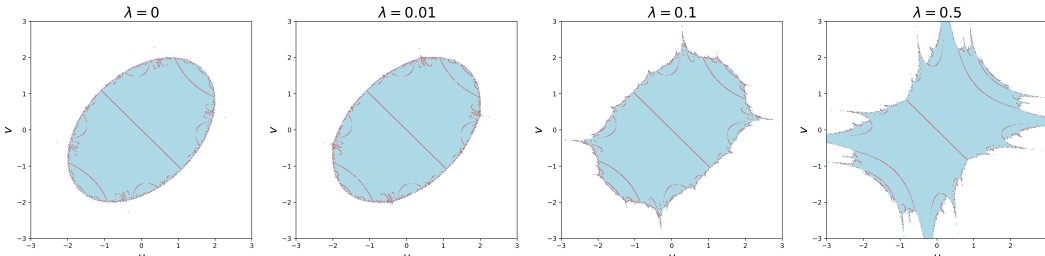

Figure 8: Gradient descent is applied to $L(u, v) = (uv - 0.8)^2/2 + \frac{\lambda}{2}(u^2 + v^2)$, where $u, v \in \mathbb{R}$ and $\lambda \in \{0, 0.01, 0.1, 0.5\}$. Blue points represent initializations that converge to a global minimizer; uncolored points represent initializations that do note converge. Red lines represent the basin of attraction of the saddle point $(0, 0)$.

**Unregularized matrix factorization with general initialization**   In Figure 9, we ran gradient descent for shallow matrix factorization without regularization under general initialization. We observe that, on a random slice of the parameter space, the convergence boundary is non-smooth, suggesting that a smooth convergence boundary is a special property of the invariant subspace $\mathcal{W}$. This also implies that, globally, the critical step size might depend intricately on the initialization. However, sensitivity to initialization is common: on all the random slices, the converged minimizer is unpredictable near the convergence boundary. This suggests that chaotic dynamics always exists near the global convergence boundary.

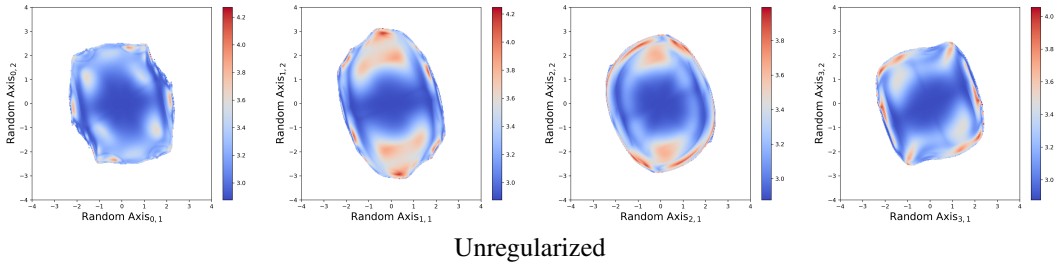

Unregularized

Figure 9: Gradient descent is applied to $L(U, V) = \|U^\top V - Y\|_F^2/2$, where $U, V \in \mathbb{R}^{5 \times 4}$ and $Y$ is a diagonal matrix whose diagonal elements are randomly sampled from $[0, 1]$. Four randomly sampled two-dimensional slices of the parameter space $\mathbb{R}^{40}$ are shown. The points are colored according to the squared Frobenius norm of the converged minimizer; uncolored points represent initializations that do not converge.

**Regularized matrix factorization with general initialization**   In Figure 10, we ran gradient descent for shallow matrix factorization with regularization under general initialization. We observe that in the random slices of the parameter space, the convergence boundary exhibits fractal-like geometry, although it appears to be less spiky than the boundary in scalar factorization. Also, as shown in the figure, sensitivity to initialization persist under general initialization. Together, Figure 10 and Figure 9 suggest that chaos and unpredictability are global properties of gradient descent in shallow matrix factorization.

**Deep matrix factorization**   In Figure 11, we ran gradient descent in depth-three matrix factorization under generalization initialization. For deep matrix factorization we observe that already for the unregularized problem, the convergence boundary exhibits fractal-like structure, as fine-scale structures emerge. We report how the squared norm of the converged minimizer depends on the initialization, for two random slices of the parameter space. As shown in the figure, while points near the origin converge to minimizers of small norm, sensitivity to initialization occurs in the vicinity of the boundary. For the regularized problem, we observe that not only the convergence boundary has a fractal-like structure, but the convergence region also becomes disconnected, with intricate connected components. The disconnectedness can be explained by the emergence of local minimizers, which attracts nearby trajectories, and non-strict saddles, which trap trajectories for long periods before they

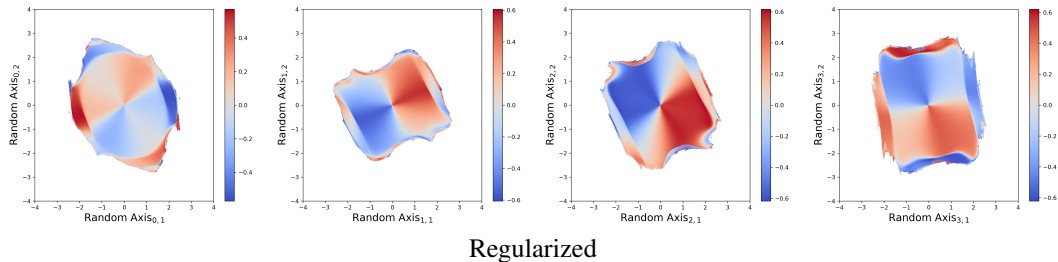

Regularized

Figure 10: Gradient descent is applied to $L(U, V) = \|U^\top V - Y\|_F^2/2 + 0.25(\|U\|_F^2 + \|V\|_F^2)$ where $U, V \in \mathbb{R}^{5\times 2}$ and $Y = \mathrm{Diag}(0.9, 0.8)$. Four randomly sampled two-dimensional slices of the parameter space $\mathbb{R}^{20}$ are shown. The points are colored according to the one coordinate value of the converged minimizer; uncolored points represent initializations that do not converge.

escape. For a detailed discussion of the landscape geometry of regularized deep matrix factorization, see the work of Chen et al. (2025). Additionally, we observe sensitivity to initialization near the convergence boundary.

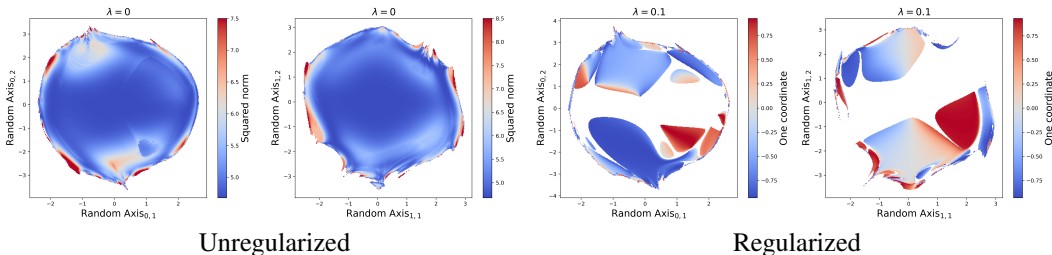

Unregularized          Regularized

Figure 11: Gradient descent is applied to $L(U, V, W) = \|UVW - Y\|_F^2/2 + \frac{\lambda}{2}(\|U\|_F^2 + \|V\|_F^2 + \|W\|_F^2)$, where $U, V, W \in \mathbb{R}^{2\times 2}$ and $Y = \mathrm{Diag}(0.9, 0.5)$. The left two panels show two randomly sampled two-dimensional slices of the parameter space $\mathbb{R}^{12}$ for the unregularized problem. Points are colored according to the squared norm of the converged minimizer. The right two panels show the same random slices for the regularized problem. Points are colored according to one coordinate of the converged minimizer. In all panels, uncolored points represent initializations that do not converge to a global minimizer.

## J    ADDITIONAL EXPERIMENTS ON REAL-WORLD DATA

**Chaotic regime vs. small-step-size regime** In Figures 4 and 12, we considered a 2-class subset of CIFAR-10, each containing 25 randomly sampled images. We trained a neural networks with two hidden layers, each with 100 neurons, for 5000 epochs. Whenever Polyak momentum is used, $\beta$ is set as 0.9. For mean-squared error, the training stops if the training loss is below $10^{-6}$ or is above $10^4$; For cross-entropy, the training stops if the training loss is below $10^{-2}$ or is above $10^4$. The results for cross entropy are shown in Figure 12. Note when the cross-entropy is employed, the final sharpness is lower than the $2/\eta$ curve, which aligns with the observations of Cohen et al. (2021). We also clearly observe two distinct step-size regimes, associated with EoS and Chaos, respectively, similar to what we had seen for the mean squared error in Figure 4.

**Fractal structure in parameter space** In Figures 5 and 13, we considered a 2-class subset of CIFAR-10, each containing 25 randomly sampled images. We trained a neural networks with two hidden layers, each with 25 neurons, for 3000 epochs under mean squared error. The weight decay parameter for Figures 5 and 13 is set as $10^{-3}$ and 0, respectively. The step size is set as $\eta = 0.05$. We randomly sampled two orthogonal unit vectors in the parameter space and kept them fixed as the coordinate axes for the random slice. On that slice, we evenly sampled $300 \times 300$ points from $[32, 42] \times [16, 24]$ (with respect to the random axes) and used them as initializations. The training stops if the training loss is below $5 \times 10^{-4}$ or is above $10^4$. Notice that in Figure 13, the fractal structures are qualitatively similar to those in Figure 5.

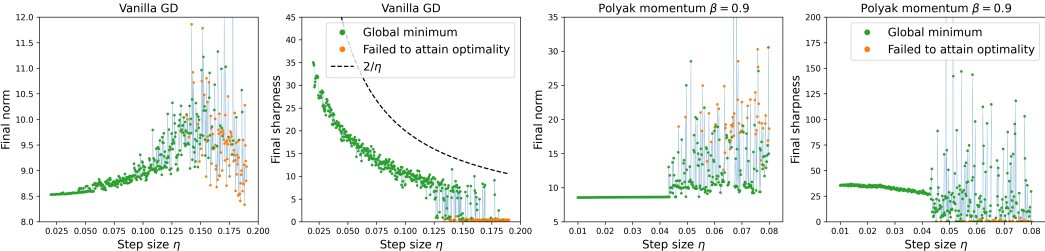

Figure 12: GD without or with momentum in training a depth-3 ReLU network on a subset of CIFAR-10 for 5000 iterations using the cross-entropy loss. The initialization is randomly sampled once and then kept fixed across all panels. At large step sizes, the final norm and final sharpness are highly sensitive to the step size.

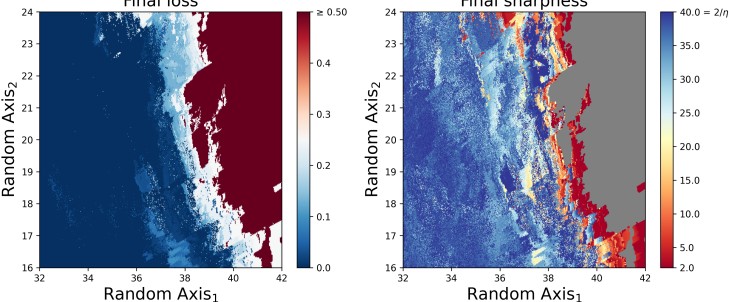

Figure 13: GD without weight decay in training a depth-3 ReLU network on a subset of CIFAR-10 for 3000 iterations. Shown is a random two-dimensional slice of the parameter space. Each point is a parameter initialization, colored according to the final value of the loss (left) and sharpness (right). Gray points are initializations from which the algorithm diverges.

