# OpenReview forum: "Gradient Descent with Large Step Sizes: Chaos and Fractal Convergence Region"
_ICLR.cc/2026/Conference — ICLR 2026 Poster_

### Official Review · Reviewer_F5Q8 · 2025-10-29

**Soundness:** 2
**Presentation:** 3
**Contribution:** 2
**Rating:** 4
**Confidence:** 3

**Summary:**

In this paper, the authors theoretically examine the dynamics of full-batch gradient descent (GD) with large step-sizes on a shallow matrix factorization problem

$$ 1/2 \Vert UV^T - Y \Vert_F^2 $$

In contrast to previous work, they particularly focus on critical learning rates, i.e., learning rates for which GD converges just about and exhibits chaotic behavior. Their main results are as follows:

For the simplified setting of scalar factorization (the matrix $Y$ is a scalar $y$ which is decomposed into the inner product of two vectors $u,v$) and fixed learning rate $\eta$, Theorem 1

- characterizes the region of initializations $D_\eta$ for which GD converges,
- shows that around any point on the boundary of D one finds initializations such that GD converges to arbitrarily large/unbalanced solutions or to the saddle $(u,v) = (0,0)$, and
- lower bounds the topological entropy of the dynamical system defined by GD characterizing it to be chaotic

In the setting of Theorem 1, but with additional $\ell_2$-regularization of $u$ and $v$, and fixed learning rate $\eta$, Theorem 3 shows that

- the outer boundary of $D_\eta$ has fractal structure,
- for $y = 0$, the set of initializations converging to global optimality is unbounded (although the set of global minimizers is bounded), and
- on the boundary of $D_\eta$ the limit is highly sensitive to the choice of initialization.

Finally, in Proposition 5 they show that Theorem 1 and 3 can be lifted to matrix valued $y$ by restricting to initializations in a particular slice of the high dimensional space.

Empirical simulations support their results.

**Strengths:**

+ Examining this extreme region of initialization/learning rate leads to beautiful connections between GD and fractals/chaotic systems

**Weaknesses:**

- The rigorous analysis is strongly restricted to shallow scalar factorization (Proposition 5 only extends the insights to a null set of the matrix factorization space)
- The results are only relevant for a (quasi-)null set of those initializations for which GD converges

**Questions:**

On the one hand, I think it is an interesting line of study to understand the connection of GD with large learning rates and chaotic systems. On the other hand, the results of the paper are strongly limited to scalar factorization and even in this case, most of the effects can only be observed on a (quasi)-null set of the successful initializations which is apparently confirmed by parts of the discussion (see points below). Furthermore, parts of the results raise doubts on whether topological entropy is the right measure to identify chaotic behavior (see points below). Consequently, I lean towards not accepting the submission. Due to time constraints, I could not check the proofs in detail.

The following points might be considered to improve the draft:

- l. 092: „the map GD sends a region C containing multiple minimizers onto a larger region that contains C“ -> Is this a general statement or restricted to a specific objective function? It seems to me that this does not hold in general. Consider, e.g., a suitably scaled version of the objective function $f(x) = 1/4 x^4 - 1/2 x^2$ with derivative $f’(x) = x^3 - x = (x-1)(x+1)x$, and C = [-2,2]. Then the map GD should map C to a subset instead of a superset.

- ll. 184/185: „…the supremum of the step sizes that allow convergence, where…“ -> shouldn’t it be „…supremum of the step sizes that allow convergence to global optimizers…“? At least this is the formal definition of $\eta(U,V)$. Furthermore, I would suggest to formally define $\mathcal{M}$ at least once in clear math notation.

- ll. 238/239: The explicit form of $D_{\eta}'$ seems to be incorrect for $y = 0$. In this case according to (3), convergence should hold for a.e. initialization and every $\eta$ such that $D_{\eta}'$ should be equal to the whole space a.e.

- l. 247: The statement claims that for $y = 0$ and any learning rate (or $y \neq 0$ and arbitrarily small learning rates) GD is a chaotic system. This appears counter-intuitive to me. Shouldn’t GD for extremely small learning rates be absolutely non-chaotic? Either there is a mistake in the statement or the use of topological entropy to measure chaos of GD seems inappropriate.

- ll. 293-311: The discussion about instability of points with norm larger than $2/\eta$ and the empirical observation that converging GD realizations always reach stationary points with norm less than $2/\eta$ aligns with existing work on sharpness regularization due to large step-sizes (since in this toy setting sharpness and parameter norm coincide). It thus considerably weakens the relevance of the theoretical statement in Theorem 1 that one can reach limit points with arbitrary high norm/sharpness.

- l. 298: „…the basin of the saddle…“

- l. 325: „…the presence of a basin…“

- l. 340: The notation {z = y} is very imprecise

- l. 358: In the definition of $p^-$ and $p^+$, the argmin/argmax is not always a singleton, or is it? If not, it would be cleaner to write $p^- \in$ argmin …

- l. 415: „…when the step size…“

- l. 436: „…applied to the scalar factorization…“

- l. 480: „..development of such a program.“

---

> ### Author Response · Authors · 2025-11-24
> **Initial response to Reviewer F5Q8 (part 1)**
>
> Thank you for taking your time to review our manuscript and for your valuable feedback. We address your comments and questions below.
>
> > The results are only relevant for a (quasi-)null set of those initializations for which GD converges.
>
> > The discussion about instability of points with norm larger than $2/\eta$ and the empirical observation that converging GD realizations always reach stationary points with norm less than $2/\eta$ aligns with existing work on sharpness regularization due to large step-sizes (since in this toy setting sharpness and parameter norm coincide). It thus considerably weakens the relevance of the theoretical statement in Theorem 1 that one can reach limit points with arbitrary high norm/sharpness.
>
> It is true that in order to attain a very high final norm or sharpness, the initializations must lie near to a null set.
> However, here the key insight is that these pathological initializations are *dense* near the convergence boundary.
> Meanwhile, although the decision boundary is a null set, its neighborhood is not a null set and can have a large volume.
> This already implies a strong form of unpredictability,
> and an important implication is as follows:
> Existing implicit bias literature typically describes the bias through certain simple rules, such as norm minimizing in the flow regime, or forcing the final sharpness close to $2/\eta$ in the EoS regime.
> Our results demonstrate that such simple descriptions are almost impossible at large step sizes.
> Specifically, since the pathological initializations are dense, **any rule to predict the final sharpness, norm, or balancedness of the converged solution (i.e., any function mapping an initialization to this quantity) must be *discontinuous almost everywhere* near the boundary**.
>
> We have also added a new theoretical result showing that **GD is unpredictable also at the distributional level**.
> Recall that the final sharpness almost surely lies in $[\gamma_{\min}, 2/\eta]$.
> We proved that for *any* sub-interval $I\subset [ \gamma_{\min}, 2/\eta]$ and any neighborhood $U$ on the convergence boundary, there is a positive-measure set of initializations within $U$ that yields final sharpness in $I$.
> In other words, for initializations randomly drawn from a neighborhood $U$ on the convergence boundary, **no matter how small $U$ is, even after removing pathological null sets, the distribution of the final sharpness (or norm) must be supported on the *entire* interval $[\gamma_{\min}, 2/\eta]$.**
> Previously, we had only illustrated this experimentally (Figure 2), and now we are including a rigorous proof (Theorem 2).
> Crucially, this suggests that the chaotic regime is fundamentally different from the EoS regime, as in the EoS regime the final sharpness is anticipated to be approximately $2/\eta$.
> We thank you for your feedback that led to this result.
>
> We also emphasize that, all the unpredictability results in Theorem 3 (Theorem 4 in the revised manuscript) for the regularized problem are not confined to null sets.
> First, we show that the convergence boundary forms a fractal, making it difficult to determine which side an initialization lies on when it is close to the boundary.
> Since both sides are open sets, this phenomenon involves sets of positive measure.
> Second, we show that an infinitesimal perturbation can switch the converged solution from $p^+$ to $p^-$, or vice versa.
> Since the fixed points $p^+$ and $p^-$ attract nearby points, their basins of attraction are open sets (a classical statement in dynamical system).
> Thus, such perturbations also occur on positive-measure sets.

---

> ### Author Response · Authors · 2025-11-24
> **Initial response to Reviewer F5Q8 (part 2)**
>
> > The rigorous analysis is strongly restricted to shallow scalar factorization. (Proposition 5 only extends the insights to a null set of the matrix factorization space).
>
> We acknowledge that the matrix factorization model we consider for our theoretical results is relatively simple, but would like to emphasize that we are advancing theory for a setting where existing theoretical results are extremely limited.
> Our work focuses on how the step size affects the GD dynamics and the training outcomes, which is an important but poorly understood question.
> Most analyses of the GD dynamics are conducted under sufficiently small step sizes.
> To our knowledge, existing rigorous study under large step sizes have all focused on highly simplified models, such as scalar factorization or width-one linear networks, (see discussion before Section 3.1 for references).
>
> In this regard, our paper offers a substantial advance of the theory for understanding GD's behavior at large step sizes.
> The key message of our work is that: **GD can be inherently unpredictable at large step sizes**.
> Also, our new theoretical result indicates that, **even at the distribution level, the chaotic regime is fundamentally different from the EoS regime**, as in the EoS regime the final sharpness should be close to $2/\eta$.
>
> In the revised manuscript, **we included new experiments verifying the unpredictability on real-world data**. We trained a three-layer ReLU network on a subset of CIFAR-10 with MSE/cross-entropy, using GD with/without momentum.
> In all cases, we observed that at large step sizes the training outcome is highly sensitive to the step size (Figure 4 and 12).
> Moreover, **we include experiments reporting fractal structures in the space of the network parameters** (Figure 5 and 13), meaning that the training outcome is also highly sensitive to the parameter initialization.
> **The difference between the chaotic regime and the EoS regime is also verified in this real-world setting** (Figure 5 and 13).
>
> We believe that these insights are both interesting and crucial in understanding the effects of step sizes, and to our knowledge, they have not been addressed by any prior work.
>
> Lastly, for the matrix factorization problem, we point out that our results describing chaos in the subspace $\mathcal{W}$ also suggest unpredictability outside $\mathcal{W}$. Due to the continuity of the map $\mathrm{GD}_\eta$, initializations in the vicinity of $\mathcal{W}$, which is a positive-measure set, will inherit the sensitivity to initial conditions, at least during the initial phase of training before potentially escaping $\mathcal{W}$ (a phenomenon known as *transient chaos* in dynamical systems).
>
> > Parts of the results raise doubts on whether topological entropy is the right measure to identify chaotic behavior (see points below).
>
> > This appears counter-intuitive to me. Shouldn’t GD for extremely small learning rates be absolutely non-chaotic? Either there is a mistake in the statement or the use of topological entropy to measure chaos of GD seems inappropriate.
>
> That is a great question!
> A short answer is that: chaos always exists, and the step size only determines where it manifests.
> Note the convergence boundary in Theorem 1 takes a form of $g(u,v)=8/\eta$.
> As $\eta\to 0$, the boundary moves farther from the origin and tends to infinity.
> So, for extremely small step size, chaos still exists, but manifests only at initializations with extremely large norms.
> Similarly, regardless of the step size, GD has a topological entropy larger than $\log 3$.
> This is because the topological entropy is a global quantity: if a system exhibits chaos anywhere in the space, its entropy is positive.
> We are confident that our statement is correct and that topological entropy is a proper measure of the chaos.
> We also highlight that this counter-intuitive result is in fact a novel implication of our work: **globally, gradient flow is qualitatively different from GD with arbitrarily small step size**.

---

> ### Author Response · Authors · 2025-11-24
> **Initial response to Reviewer F5Q8 (part 3)**
>
> > "The map GD sends a region $C$ containing multiple minimizers onto a larger region that contains $C$" -> Is this a general statement or restricted to a specific objective function? It seems to me that this does not hold in general. Consider, e.g., a suitably scaled version of the objective function $f(x)=1/4x^4+1/2x^2$ with derivative $f'(x)=x^3-x$, and $C = [-2,2]$. Then the map GD should map $C$ to a subset instead of a superset.
>
> The discussion of the mechanism behind chaos that we offer is for general settings, and we hypothesize that the conditions sketched in Section 4 should be sufficient for chaos to emerge under general objectives.
>
> For the specific objective $f(x)=\frac{1}{4}x^4+\frac{1}{2}x^2$, regarding your comment "GD should map $C$ to a subset instead of a superset", some caution is needed.
> Here, the GD update map is $\mathrm{GD}\_\eta(x)=x-\eta f'(x)=x-\eta(x^3-x)$.
> Thus, whether $\mathrm{GD}_\eta([-2,2])$ contains or is contained in $[-2,2]$ depends on the step size.
>
> Nevertheless, our conditions indeed apply to this objective $f$.
> Recall our statement is that chaos emerges if there exists a component $C$ such that (i) $\mathrm{GD}\_\eta$ behaves as a covering map on $C$ and (ii) $\mathrm{GD}\_\eta$ sends $C$ to a superset of $C$.
> When $\eta=2$, note that $\mathrm{GD}\_\eta$ maps $C=[-\sqrt{2},\sqrt{2}]$ to itself (which is a superset), and also that $\mathrm{GD}\_\eta$ bijectively maps $[-\sqrt{2}, -\frac{1}{\sqrt{2}}], [-\frac{1}{\sqrt{2}}, \frac{1}{\sqrt{2}}], [\frac{1}{\sqrt{2}},\sqrt{2}]$ to $C$, which means $\mathrm{GD}\_\eta$ behaves as a $3-$fold covering map on $C$.
> Accordingly, this specific system $x\mapsto -2x^3+3x$ is known to be a chaotic system [1].
>
> We confirm that in our condition we require $\mathrm{GD}_\eta$ to send $C$ to a superset instead of a subset.
> In the revised manuscript, we have made this clearer by elaborating how precisely GD folds the parameter space in problem $L(u,v)=(uv-1)^2$ with $(u,v)\in \mathbb{R}^2$, with a visualization provided in Figure 6.
>
> Finally, to extend the mechanism to broader settings, we have included a new theoretical result showing that, **for deep linear networks with arbitrary depth and width and any polynomial loss, gradient descent is a smooth covering map outside a measure zero set in the parameter space** (Proposition 6). This new result suggests that the folding behavior of GD, and hence chaotic dynamics, may persist in more general settings.
>
> [1] Rogers, Thomas D., and David C. Whitley. "Chaos in the cubic mapping." Mathematical Modelling 4.1 (1983): 9-25.
>
> > The explicit form of $\mathcal{D}\_\eta'$ seems to be incorrect for $y=0$. In this case according to (3), convergence should hold for a.e. initialization and every $\eta$ such that $\mathcal{D}_\eta'$ should be equal to the whole space a.e.
>
> Please note that when $y=0$, the convergence region becomes $\mathcal{D}_\eta'=\\{\\|u\\|^2+\\|v\\|^2 < 4/\eta \\}$, which is a ball with radius $2/\sqrt{\eta}$, not the full space.
> Accordingly, the critical step size becomes $\eta^*(\bar{u},\bar{v})=4/(\\|\bar{u}\\|^2+\\|\bar{v}\\|^2)$, so any step size above this value leads to non-convergence almost surely.
>
> > "the supremum of the step sizes that allow convergence, where..." -> shouldn't it be "supremum of the step sizes that allow convergence to global optimizers"? At least this is the formal definition of $\eta(U,V)$. Furthermore, I would suggest to formally define $\mathcal{M}$ at least once in clear math notation.
>
> > Q6-Q12
>
> Thank you for the suggestions and for pointing out the typos!
> In the revised manuscript we have specified "convergence to a global optimizer" in defining critical step size;
> included a formal definition of $\mathcal{M}$;
> and changed $\\{z=y\\}$ to $T(\mathcal{M})$ in Proposition 2 (Proposition 3 in the revised manuscript).
> For the definition of $p^+(\theta)$ and $p^-(\theta)$,
> the argmin and argmax are indeed always singleton sets if $\theta$ is in the convergence region $\mathcal{D}_\eta$ (which is required by Theorem 3).
> We have included a footnote regarding this.
> We have also fixed all other typos.
> Thank you again, we appreciate your help!
> We hope the above updates adequately address your concerns and remain attentive to any further suggestions you might have.

---

> ### Comment · Reviewer_F5Q8 · 2025-11-27
>
> I thank the authors for their clarifications! I still feel that the analyzed setting is strongly limited and that the paper overstates the theoretical contribution at places, but I acknowledge the value of the findings which have been further strengthened by the new Theorem 2 and additional experiments in Figures 4,5,12,13. I thus raised my score.
>
> Regarding the convergence radius for y=0: I misread the min in (3) for a max. Sorry, my bad…

---

### Official Review · Reviewer_V6Q9 · 2025-10-30

**Soundness:** 4
**Presentation:** 3
**Contribution:** 2
**Rating:** 4
**Confidence:** 3

**Summary:**

The authors examine the dynamics of gradient descent with step sizes at (or near) the critical value associated with a given initialization. In particular, they analyze the scalar factorization problem in both its regularized and unregularized forms.
In the unregularized case, they rigorously determine the critical step size for any given initialization and show that the behavior near the convergence boundary is chaotic, both in the norm of the minimizer, the imbalance between the two factorization vectors, and the nature of the final point (minimum or strict saddle). They also derive a lower bound for the topological entropy of gradient descent in this setting.
In the regularized case, they rigorously show that the convergence boundary is fractal, implying that convergence itself is unpredictable near the boundary, and provide an intuitive explanation of the mechanism, based on a partition of the parameter space. Here too, they demonstrate that the dynamics initialized near the boundary are chaotic, showing in particular that gradient descent converges either to the nearest or the farthest global minimizer, and that infinitesimal perturbations can flip this selection.
Finally, they show that in the general matrix factorization problem, there exists a non-trivial set of initializations for which the problem decomposes into parallel sub-dynamics of scalar factorizations, allowing the extension of the previous results to this case.

**Strengths:**

The setting studied in this work allows for a rigorous analysis of gradient descent when critical step sizes are used, with a thorough characterization of the chaotic nature of the dynamics and of the unpredictability of both convergence itself and the final point.
As stated by the authors, this can be connected with broader lines of research in realistic machine learning settings, such as the edge
of stability, and to the best of my knowledge, this is the first rigorous work on the characterization of chaos for gradient descent.

**Weaknesses:**

The setting studied in this paper is very specific, namely the scalar version of the matrix factorization problem. As stated by the authors, this can be extended to the general matrix factorization only with orthogonal initializations, which correspond to the case where the problem actually decomposes into many parallel scalar ones.
It would strengthen the impact to clarify how these mechanisms extend in more general settings, or even how the intuition provided by this simple setting is useful when moving to more realistic scenarios where gradient descent with critical step sizes is used.

**Questions:**

Beyond the numerical experiments for general initializations in the appendix, can you provide some results or intuition on whether the qualitative mechanisms and intuitions developed in this paper for a simple optimization landscape persist in more complex settings, where, for example, many local minima are present (either in theoretical settings or in experimental findings in realistic machine learning settings)?

Do you have any insights on whether similar chaotic or fractal behaviors might appear for adaptive or momentum-based optimizers (e.g., Adam) near critical hyperparameters?

In Theorem 3, unboundedness is shown only in the case of $y = 0$, but it is then stated in the main text that, in general, the convergence region has an unbounded interior. Even though the caption of Figure 3 states that this behavior is qualitatively observed in general, I think it should be made clearer in the main text that this property, in the general $y \neq 0$ setting, is only observed numerically.

In the figure captions, the parameter space is referred to as $(x, y)$ in the unbounded case. I think this causes confusion, since in the main text $(u, v)$ is used in both cases, and $y$ is used for the target of the factorization.

---

> ### Author Response · Authors · 2025-11-24
> **Initial response to Reviewer V6Q9 (part 1)**
>
> Thank you for taking your time to review our manuscript and for your valuable feedback. We address your comments and questions as detailed below.
>
> > It would strengthen the impact to clarify how these mechanisms extend in more general settings, or even how the intuition provided by this simple setting is useful when moving to more realistic scenarios where gradient descent with critical step sizes is used.
>
> > Beyond the numerical experiments for general initializations in the appendix, can you provide some results or intuition on whether the qualitative mechanisms and intuitions developed in this paper for a simple optimization landscape persist in more complex settings, where, for example, many local minima are present (either in theoretical settings or in experimental findings in realistic machine learning settings)?
>
> > Do you have any insights on whether similar chaotic or fractal behaviors might appear for adaptive or momentum-based optimizers (e.g., Adam) near critical hyperparameters?
>
> Thanks for the great questions!
> In our initial manuscript, we traced the general mechanism underlying the chaos to a folding behavior of gradient descent.
> In the revised manuscript, we have made this discussion clearer by elaborating how precisely GD folds the parameter space in problem $L(u,v)=(uv-1)^2$ with $(u,v)\in \mathbb{R}^2$, with a visualization provided in Figure 6.
> Moreover, extending the discussion to broader settings, we have included a new theoretical result showing that, **for deep linear networks with arbitrary depth and width and any polynomial loss, gradient descent is a smooth covering map outside a measure zero set in the parameter space** (Proposition 6).
> This new result suggests that, even under a more complex landscape with non-global minimizers and non-strict saddles, GD still exhibits a folding behavior, and accordingly, we expect chaos persists in those cases.
> We thank you for your feedback that led to this result.
>
> Beyond the theory, we have also included new experiments with more complex models and real-world data, where we validated the presence of chaos. Please see our next response for more details of these experiments.

---

> ### Author Response · Authors · 2025-11-24
> **Initial response to Reviewer V6Q9 (part 2)**
>
> > The setting studied in this paper is very specific. As stated by the authors, this can be extended to the general matrix factorization only with orthogonal initializations, which correspond to the case where the problem actually decomposes into many parallel scalar ones.
>
> While we acknowledge that our setting is relatively specific when compared with other available models, we emphasize the significant limitations of existing theory to study large step sizes in general.
> Our work focuses on how the step size affects the GD dynamics and the training outcomes, which is an important but poorly understood question.
> To our knowledge, existing theoretical work on GD dynamics with large step sizes have all focused on highly simplified models, such as scalar factorization or width-one linear networks (see discussion before Section 3.1 for references).
> In light of this, we believe that our paper offers a substantial advancement in understanding GD's behavior at large step sizes.
>
> In the revised manuscript we have included new experiments on real-world data, and we highlight two key implications of our work for broader scenarios:
>
> * The key message of our work is that: **GD can be inherently unpredictable at large step sizes**. This chaotic regime stands in sharp contrast to the previously studied settings where GD has predictable implicit biases. In the revised manuscript, **we included experiments verifying this unpredictability on real-world data**. We trained a three-layer ReLU network on a subset of CIFAR-10 with MSE/cross-entropy, using GD with/without Polyak momentum. In all cases, we observed that at large step sizes the training outcome is highly sensitive to the step size (Figure 4 and 12). Moreover, **we report fractal structures in the space of the network parameter** (Figure 5 and 13), meaning that the training outcome is also sensitive to the parameter initialization.
>
> * In the revised manuscript, we included a new theoretical result showing that **GD is unpredictable also at the distributional level**. We showed that, for initializations randomly drawn from an *arbitrarily small* neighborhood intersecting the convergence boundary, the distribution of the final sharpness (or norm) is supported on the *entire* interval $[\gamma_{\min}, 2/\eta]$, where $\gamma_{\min}$ is the smallest sharpness (or norm) over all minimizers. Previously, we had only illustrated this experimentally (Figure 2), and have now added a rigorous proof (Theorem 2). This implies that, **at large step sizes, even at the distributional level, the best prediction of the final sharpness is the *entire* interval $[\gamma_{\min}, 2/\eta]$; no smaller range is possible**. Crucially, this suggests that **the chaotic regime is fundamentally different from the EoS regime**, as in the EoS regime the final sharpness is anticipated to be approximately $2/\eta$. The difference between these two regimes is also verified on real-world settings (Figure 4 and Appendix 12).
>
> We believe that the above phenomena are crucial in understanding the effects of step sizes, and to our knowledge, they have not been addressed by any prior work. We rigorously showed these phenomena on a simplified yet very non-trivial model and empirically verified them in real-world settings.
>
> > In Theorem 3, unboundedness is shown only in the case of $y=0$, but it is then stated in the main text that, in general, the convergence region has an unbounded interior. Even though the caption of Figure 3 states that this behavior is qualitatively observed in general, I think it should be made clearer in the main text that this property, in the general setting, is only observed numerically.
>
> > In the figure captions, the parameter space is referred to as $(x,y)$ in the unbounded case. I think this causes confusion, since in the main text $(u,v)$ is used in both cases, and is used for the target of the factorization.
>
> Thanks for pointing these out!
> We have revised our statements to explicitly mention $y=0$ when discussing the theoretical result on unboundedness.
> We also have revised the figure caption to use $(u,v)$ consistently.
> We hope the above updates adequately address your concerns and remain attentive to any further suggestions you might have.

---

### Official Review · Reviewer_t5Rh · 2025-10-31

**Soundness:** 3
**Presentation:** 2
**Contribution:** 3
**Rating:** 4
**Confidence:** 3

**Summary:**

This paper analyzes gradient descent (GD) for both regularized and unregularized (shallow) matrix factorization, which simplifies to a scalar factorization problem under some orthogonal conditions. For the unregularized case, the authors prove the critical step size needed to converge to a global minimizer, as well as the conditions under which GD fails to converge to a minimizer. They state that this new result also tightens existing ones found in the literature.
Furthermore, they show how GD is sensitive to initialization for this problem, and how at the critical step size, infinitesimal perturbations of the initialization can send the trajectory to a minimizer with arbitrarily large norm, sharpness, or imbalance, or to a saddle point. For the regularized problem, they show that the global dynamics of GD become even more unpredictable than for the unregularized problem. This unpredictability depends on the geometry of the boundary of the convergence region, which they show is self-similar and hence has a fractal structure.

**Strengths:**

This paper presents new insights into the geometry of the convergence boundary for matrix factorization, which simplifies to a scalar factorization problem. It also presents a dichotomy in the geometry between regularized and unregularized scalar factorization: for the regularized problem, the convergence region (up to a measure-zero set) has an unbounded interior, whereas for the unregularized case, it coincides almost everywhere with a bounded domain. To my knowledge, this is novel.

Furthermore, the authors present critical step size regions that determine convergence or failure to converge. I am not sure exactly how novel this part is, however (which I discuss in the following section). Nonetheless, I believe that the mathematical tools used to derive the results could be useful for the broader research community.

**Weaknesses:**

- I am somewhat unsure about the novelty of the step size bound for convergence in the unregularized case. Looking at the first term in Equation (3), I believe it comes from the inverse of the sharpness. If we then consider the results from Ghosh et al. (2025), specifically the case where $L=2$ and the target matrix is rank-1, this seems to reduce to the scalar factorization case. For this, they then prove that using a learning rate of $\eta > 1 / |y|$ induces oscillations (and thus prevents convergence). Could the authors verify if this is the case? I do think that the second term in Equation (3) could be new, and it warrants more discussion than simply stating it depends on the convergence region. There should also be more discussion on what is done differently to improve the Wang et al. (2022) results, as mentioned in Line 255.

- The paper's language is sometimes overly strong; for example, in line 257, the phrase "precise description of chaos in gradient descent." This is a description of GD on a specific problem, not a general one.

- I found the mathematical language to be overly complicated at times. For example, if I understand correctly, Proposition 5 seems to state that if we start from an orthogonal initialization, GD is invariant. This allows a simplification to the loss function in lines 162-163, which then reduces to the scalar factorization case. If so, this section could be simplified to improve readability. I also found lines 363-373 difficult to follow, though this may just be my interpretation.

- Finally, a limitation is the simplicity of the studied model. While I do not expect derivations for the settings in lines 478-480, experimental results would be beneficial. For example, does adding regularization to deep matrix factorization alter the results of Ghosh et al. (2025), and do any of the results from this paper translate to that setting?

**Questions:**

I have listed a few questions in the weaknesses section.

---

> ### Author Response · Authors · 2025-11-24
> **Initial response to Reviewer t5Rh (part 1)**
>
> Thank you for taking your time to review our manuscript and for your valuable feedback. We address your comments and questions as detailed below.
>
> > I am somewhat unsure about the novelty of the step size bound for convergence in the unregularized case. Looking at the first term in Equation (3), I believe it comes from the inverse of the sharpness. If we then consider the results from Ghosh et al. (2025), specifically the case where $L=2$ and the target matrix is rank-1, this seems to reduce to the scalar factorization case. For this, they then prove that using a learning rate of $\eta>1/|y|$ induces oscillations (and thus prevents convergence). Could the authors verify if this is the case?
>
> > I do think that the second term in Equation (3) could be new, and it warrants more discussion than simply stating it depends on the convergence region. There should also be more discussion on what is done differently to improve the Wang et al. (2022) results, as mentioned in Line 255.
>
> We agree that the first term in the critical step size is not novel and is indeed related to the oscillation results by Ghosh et al. (2025). In fact, we view this as a standard result from stability analysis: when the step size exceeds a threshold at which every minimizer becomes Lyapunov-unstable (in our case, $1/|y|$), GD fails to converge to any minimizer almost surely.
>
> We emphasize however that the second term, and therefore the entire critical step size, are highly non-trivial. **The critical step size provides a *necessary and sufficient* condition for convergence,
> for all step sizes $\eta$ and initializations $\theta_{0}$** (except for a null set in the $(\eta, \theta_0)$-space). Obtaining such a global convergence result is notoriously difficult in non-convex optimization, which, to our knowledge, is very rare in existing work. The only result in this direction we are aware of is [1], which studied an objective function on $\mathbb{R}$.
>
> Wang et al. (2022) only considered step sizes in the range $\eta<1/(3|y|)$, and initializations in a strict subset of our convergence region; see Figure 7 for a visualization. In their setting, since the initializations are far from the convergence boundary, the dynamics are tame and non-chaotic.
>
> In contrast, our proof relies on a global analysis of the dynamical system $\mathrm{GD}\_{\eta}$.
> Our key technical novelties are: (i) we identified an equivariance symmetry in $\mathrm{GD}\_{\eta}$ and the corresponding quotient dynamics; (ii) we analyzed all branches of the inverse map $\mathrm{GD}\_{\eta}^{-1}$ (which is a multi-valued map) using tools from differential topology, and thoroughly studied the inverse dynamics defined by $\mathrm{GD}\_{\eta}^{-1}$; (iii) we used symbolic dynamics to show that $\mathrm{GD}\_{\eta}$ is topologically transitive on the convergence boundary.
> We believe these techniques are novel in machine learning optimization and are valuable for future research.
>
> We thank the reviewer for the valuable suggestions and have included the above discussions with more details in the revised manuscript (Figure 7 and Appendix E.1.2).
>
> [1] Chen, Xuxing, et al. "From Stability to Chaos: Analyzing Gradient Descent Dynamics in Quadratic Regression." Transactions on Machine Learning Research.

---

> ### Author Response · Authors · 2025-11-24
> **Initial response to Reviewer t5Rh (part 2)**
>
> > a limitation is the simplicity of the studied model. While I do not expect derivations for the settings in lines 478-480, experimental results would be beneficial.
>
> > For example, does adding regularization to deep matrix factorization alter the results of Ghosh et al. (2025), and do any of the results from this paper translate to that setting?
>
> Please note that experiments for deep matrix factorizations under general initializations have already been provided in Appendix I, where fractal boundaries and chaos can be observed in both regularized and unregularized settings.
>
> In the meantime, **we have included experiments verifying the chaos for GD on more complex models and real-world data**. We trained a three-layer ReLU network on a subset of CIFAR-10 using MSE/cross-entropy and GD with/without momentum.
> In all cases, we observed that at large step sizes the training outcome is highly sensitive to the step size (Figure 4 and 12).
> Furthermore, **we report fractal structures in the space of parameters** (Figure 5 and 13), meaning that the training outcome is also sensitive to the parameter initialization.
>
> Regarding the relation to the results of Ghosh et al. (2025),
> we first point out that their analysis relies on a simplification similar to ours.
> They impose an assumption on the initialization, termed the "Singular Vector Stationary Set", which requires the singular vectors of all weight matrices to remain fixed throughout training.
> Under this assumption, the dynamics decouples into several independent systems on the singular values, each taking the form $L(\sigma_1,\cdots,\sigma_L)=(\Pi_{i=1}^L\sigma_i - \sigma_*)^2$ with $\sigma_i,\sigma_*\in \mathbb{R}$.
> Note, this is essentially a depth-$L$, width-$1$ scalar factorization problem, whereas in contrast we considered the depth-$2$, width-$d$ setting.
> In this sense, our setting is orthogonal to theirs rather than being a special case.
> Also, we point out that for $L=2$, the specific initializations considered in their work (equations (3) and (4) in the work of Ghosh et al. (2025)) satisfy our orthogonal condition, and therefore our results directly apply in that case.
>
> Finally, in regard to the simplicity of the matrix factorization model, we emphasize the limitations of existing theory for large step sizes in general.
> Our work focuses on how the step size affects the GD dynamics and the training outcomes, which is an important but poorly understood question.
> To our knowledge, existing analysis of GD dynamics have either focused on sufficiently small step sizes, or on highly simplified models, such as scalar factorization or width-one linear networks (see discussion before Section 3.1 for references).
> In this regard, we believe that our paper substantially advances the theoretical understanding of GD's behavior at large step sizes.
> The key message of our work is that: **GD can be inherently unpredictable at large step sizes**.
> This chaotic regime is in sharp contrast to the previously studied settings where GD has predictable implicit biases.
> We believe this message is crucial in understanding the effects of step sizes, and to our knowledge, it has not been addressed by any prior work.
> Here we rigorously proved it on a simplified but non-trivial model and empirically verified it in more complex models and real-world data.
>
> > The paper's language is sometimes overly strong: for example, in line 257, the phrase "precise description of chaos in gradient descent." This is a description of GD on a specific problem, not a general one.
>
> > I found the mathematical language to be overly complicated at times. For example, if I understand correctly, Proposition 5 seems to state that if we start from an orthogonal initialization, GD is invariant. This allows a simplification to the loss function in lines 162-163, which then reduces to the scalar factorization case. If so, this section could be simplified to improve readability. I also found lines 363-373 difficult to follow, though this may just be my interpretation.
>
> Thanks for the valuable suggestions!
> We have revised our statements to make them more precise within the specific settings.
> We have removed Proposition 5 and instead used a description as follows:
> "with initialization in $\mathcal{W}$, the trajectory remains in $\mathcal{W}$, and the dynamics decouple column-wise: each pair $(u^i,v^i)$ evolves independently according to the scalar factorization dynamics with target $y_i$, the $i$th diagonal entry of $Y$."
> To help clarify the geometry of the convergence boundary, we have expanded the illustrative example for the case $d = 1$. We hope these updates adequately address your concerns, and we remain attentive to any further suggestions you might have.

---

### Official Review · Reviewer_Nc47 · 2025-11-01

**Soundness:** 3
**Presentation:** 3
**Contribution:** 3
**Rating:** 6
**Confidence:** 4

**Summary:**

This paper studies regularized matrix factorization problems with gradient descent (GD). The authors investigate what happens when GD is run with large step sizes, a regime often associated with improved generalization in deep learning, yet poorly understood theoretically.

They first start from a simplified matrix factorization, in which the target matrix is reduced to a scalar. In particcular, for the unregularized scalar matrix factorization the authors characterize the critical step size below which the GD converges almost surely, sensitivity to initialization, and topological entropy of the GD system. For the regularized case the GD training dynamics exhibit fractal convergence boundadry.

Then they discuss the possibility to extend their results to general matrix factorization problems. The paper is concluded with some discussions on limitations and future directions.

**Strengths:**

The authors provide a solid theoretical analysis for matrix factorization with gradient descent. The main results reveal that the training dynamics can exhibit chaos and fractal geometry when stepsizes vary.

**Weaknesses:**

1. The main results are limited to gradient descent on matrix factorization problems. In practice there is still a gap between this type of optimizatin problem and large-scale machine learning problems with stocahstic gradient methods.

2. For the general matrix factorization problems, the authors only provide partial results -- the initialization of the matrices should satisfy a special orthogonality condition, under which the trajectories of the column vectors can be characterized using the scalar factorization case. This type of condition might be a bit strong and not covering all cases. This may also indicate the intrinsic difficulties of extending the scalar case to the high dimensional case.

3. The experiments provided are limited to synthetic simulations, and there seems no detailed discussions between the theoretical results on chaos, fractal geometry and empirical observations from real-world problems.

**Questions:**

1. Although Section 4 covers the case when initialization matrices are rescaling of identities, there are still many cases not covered and the orthogonality conditions seems to be strong. Could the authors could provide some comments on the main challenges of extending the one-dimensional case to the high-dimensional case.

2. Could the authors provide some discussions on whether or not the chaos or factal geometry can be observed in real-world machine learning problems?

3. Any challenges of extending the GD dynamics to SGD dynamics?

---

> ### Author Response · Authors · 2025-11-24
> **Initial response to Reviewer Nc47 (part 1)**
>
> Thank you for taking your time to review our manuscript and for your valuable feedback. We address your comments and questions below.
>
> > The main results are limited to gradient descent on matrix factorization problems. In practice there is still a gap between this type of optimization problem and large-scale machine learning problems with stochastic gradient methods.
>
> > The experiments provided are limited to synthetic simulations, and there seems no detailed discussions between the theoretical results on chaos, fractal geometry and empirical observations from real-world problems.
>
> > Could the authors provide some discussions on whether or not the chaos or fractal geometry can be observed in real-world machine learning problems?
>
> Our theoretical results focus on matrix factorization, which is a fundamental problem in optimization and machine learning, but we agree that more general settings are also of interest and we believe similar results are plausible.
>
> We think it is important to stress that GD with *arbitrary step sizes*, with a particular focus on the large-step-size regime, is a direction that has only rarely been studied in prior work, as analyzing discrete-time dynamics in non-convex optimization, such as matrix factorization, is mathematically very challenging.
> To our knowledge, existing theoretical analyses on GD dynamics with large step sizes in non-convex problems have all focused on highly simplified models, such as scalar factorization or width-one linear networks, and have been *restricted to some specific ranges of step sizes* (see discussion before Section 3.1 for references).
> In this regard, our analysis presents a substantial advancement in understanding large-step-size phenomena.
>
> In response to your comment, we have now included new experiments evaluating more complex models and real-world data, as well as new theoretical results. We highlight two key implications of our work for broader scenarios:
>
> * The key message of our work is that: **GD can be inherently unpredictable at large step sizes**. This chaotic regime stands in sharp contrast to the previously studied settings where GD has predictable implicit biases. In the revised manuscript, **we included new experiments verifying this unpredictability on real-world data**. We trained a three-layer ReLU network on a subset of CIFAR-10 with MSE/cross-entropy using GD with/without momentum. In all cases, we observed that at large step sizes the training outcome is highly sensitive to the step size (Figure 4 and 12). Moreover, **we report fractal structures in the space of network parameters** (Figure 5 and 13), meaning that the training outcome is also sensitive to the parameter initialization.
>
> * In the revised manuscript, we included a **new theoretical result showing that GD is unpredictable also at the distributional level** (Theorem 2). We showed that for initializations randomly drawn from an *arbitrarily small* neighborhood on the convergence boundary, the distribution of the final sharpness (or norm) must be supported on the *entire* interval $[\gamma_{\min}, 2/\eta]$, where $\gamma_{\min}$ is the smallest sharpness (or norm) over all minimizers. In our initial manuscript this was only experimentally illustrated (Figure 2), and the new result offers a rigorous proof. This implies that, **at large step sizes, even at the distributional level, the best prediction of the final sharpness is the *entire* interval $[\gamma_{\min}, 2/\eta]$; no smaller range is possible**. Crucially, this suggests that **the chaotic regime is fundamentally different from the EoS regime**, as in EoS regime the final sharpness is anticipated to be approximately $2/\eta$. The difference between these two regimes is also verified on real-world settings (Figure 4 and 12).
>
> We believe that the above phenomena are both interesting and crucial in understanding the effects of step sizes, and to our knowledge, they have not been addressed by any prior work. Here we rigorously proved them on a simplified but quite non-trivial model and empirically verified them on real-world settings.

---

> ### Author Response · Authors · 2025-11-24
> **Initial response to Reviewer Nc47 (part 2)**
>
> > For the general matrix factorization problems, the authors only provide partial results -- the initialization of the matrices should satisfy a special orthogonality condition, under which the trajectories of the column vectors can be characterized using the scalar factorization case. This type of condition might be a bit strong and not covering all cases. This may also indicate the intrinsic difficulties of extending the scalar case to the high dimensional case.
>
> > Although Section 4 covers the case when initialization matrices are rescaling of identities, there are still many cases not covered and the orthogonality conditions seems to be strong. Could the authors could provide some comments on the main challenges of extending the one-dimensional case to the high-dimensional case.
>
> > Any challenges of extending the GD dynamics to SGD dynamics?
>
> Thanks for the great questions! Indeed, extending the results to general initialization is challenging. In fact, understanding gradient descent dynamics for matrix factorization under general (non-structured) initialization has remained a longstanding open problem, even in the regime of small step sizes. For $L(U,V)=\\|U^\top V-Y\\|_F^2$, the main challenge is to analyze how the (left) singular vectors of $U$ and $V$ evolve during training. Under orthogonal initialization, it suffices to analyze the evolution of the $i$th pair of (left) singular vectors of $U$ and $V$, independently for each $i$. In general, one needs to consider the alignment between all pairs of singular vectors, which is much more challenging.
>
> Nonetheless, we point out that the presence of chaos on the subspace $\mathcal{W}$ (defined by our orthogonality conditions) already suggests that the global dynamics can be unpredictable: Due to the continuity of the map $\mathrm{GD}_\eta$, initializations in the vicinity of $\mathcal{W}$ will inherit the sensitivity to initial conditions, at least during the initial phase of training before potentially escaping $\mathcal{W}$ (a phenomenon known as *transient chaos* in dynamical systems).
>
> Regarding SGD, stochasticity brings additional complexity. The dynamics of SGD are essentially stochastic processes, where one may consider how a distribution over the parameter space evolves over time. So, the dynamics take place in the space of distributions, which is infinite dimensional. Many existing theoretical approaches consider the continuous-time diffusion approximation. Given that GD already exhibits chaos in finite-dimensional settings, as we have shown, we expect that analyzing SGD under large step size could be difficult. The only work on this direction we know of is [1], which focused on objectives defined on $\mathbb{R}$ or $\mathbb{R}^2$. We think this is an interesting future direction, and hope that the methods and insights developed in our paper will be valuable for exploring it further.
>
> [1] Ziyin, Liu, et al. "SGD with a constant large learning rate can converge to local maxima." ICLR 2022.

---

### Author Response · Authors · 2025-11-30
**Summary of the rebuttal (part 1)**

We sincerely thank the reviewers for their valuable suggestions and feedback which have helped us make important improvements. We have responded to each of the raised concerns in detail, and we summarize the main points below for the convenience of the AC.

**1: Simplicity of the studied model**

Reviewers perceived the simplicity of the model we analyzed as a weakness. Our theoretical results focus on matrix factorization. We acknowledge that this model is relatively simple, but we also emphasize that we are advancing theory for an important setting where existing theoretical results are extremely limited: GD with arbitrary, particularly large, step sizes. To our knowledge, all existing rigorous analyses of GD dynamics under large step sizes in non-convex problems have focused on highly simplified settings, such as scalar factorization or width-one linear networks, and have been restricted to some specific ranges of step sizes. From this standpoint, our analysis offers an important advancement in understanding large-step-size phenomena. Compared to existing work, we identified a previously unknown, but rather important behavior of GD: **GD can be inherently unpredictable at large step sizes**. This chaotic regime stands in sharp contrast to the previously studied settings where GD has predictable implicit biases, such as the flow regime or the Edge of Stability (EoS) regime.

In response to the reviewers' concern about whether our findings extend to more realistic settings, we have included new experiments on more complex models and real-world data. We trained three-layer ReLU networks on CIFAR-10 with different losses and variants of GD, and empirically verified the presence of chaos and fractal convergence boundary.


**2: Convergence to a high-norm solution occurs on a null set of initializations**

Reviewer F5Q8 states that in Theorem 1 convergence to a solution with norm greater than $2/\eta$ occurs only on a null set of initializations (the basin of attraction of unstable minimizers). However, we highlight that the key insight here is that these pathological initializations are *dense* near the convergence boundary, which already implies a strong form of unpredictability. As a consequence, **any rule to predict the final sharpness, norm, or balancedness of the converged solution must be discontinuous almost everywhere near the boundary**.

Additionally, we have added a new theoretical result showing that **GD is unpredictable also at the distributional level** (Theorem 2): for initializations randomly drawn from an *arbitrarily small* neighborhood, even after removing pathological null sets, the best prediction of the final sharpness is the *entire* interval $[\gamma_{\min}, 2/\eta]$; no smaller range is possible. This reiterates that **the chaotic regime is fundamentally different from the EoS regime**, as in the EoS regime the final sharpness is anticipated to be approximately $2/\eta$. We also verified the distinction between the chaotic and the EoS regimes in the real-world experiments.


**3: Generality and intuition for the mechanism behind chaos**

We traced the general mechanism underlying chaos to a folding behavior of gradient descent. Reviewer V6Q9 asked to what extent this mechanism extends to broader settings; Reviewer F5Q8 asked clarification questions. In response, we have made our discussion clearer by elaborating how precisely GD folds the parameter space in a two-dimensional example. Additionally, we have included a new theoretical result showing that,
**for general deep networks with polynomial activations and under any polynomial loss, gradient descent is a smooth covering map outside a measure zero set in the parameter space** (Proposition 6). This new result indicates that the folding behavior of GD, and thus chaos, persists in broader settings.

Note: in our initial response to the reviewers we had stated this new result for deep linear network, and we have since further extended it to networks with polynomial activations. The proof only relies on the GD update map having polynomial coordinates, and thus applies to polynomial activations.

---

> ### Author Response · Authors · 2025-11-30
> **Summary of the rebuttal (part 2)**
>
> **4: Novelty of the critical step size**
>
> Reviewer t5Rh asked about the novelty of the critical step size in Theorem 1. We emphasize that the statement provides a necessary and sufficient condition for convergence, for all step sizes $\eta$ and initializations $\theta_0$ (except for a null set in the $(\eta,\theta_0)$-space). Such global convergence result is notoriously difficult to obtain in non-convex optimization, which, to our knowledge, is very rare in existing work.
>
> **5: Orthogonal initialization**
>
> Reviewer Nc47, V6Q9 and F5Q8 raised concerns about the initialization conditions. We acknowledge the difficulty in extending our theoretical results to general initialization, and at the same time emphasize that understanding GD dynamics for matrix factorization under general initialization has remained a longstanding open problem even in the very well studied regime of small step sizes. We also explained how, as a consequence of the continuity of the GD update map, the presence of chaos on our considered subspace of initializations already implies unpredictability in a broader sense.
>
> We hope to have addressed all the concerns by the reviewers. We note that Reviewer F5Q8 "acknowledged the value of the findings" and raised their score, while other reviewers had yet to respond to our initial rebuttal. We believe that our findings on the unpredictability of GD are both interesting and crucial in understanding the effects of step sizes, and to our knowledge, they have not been addressed by any prior work.

---

### Public Comment · ~Shuang_Liang6 · 2026-03-05
**Camera-Ready Revisions**

We thank the Reviewers and Area Chairs for their valuable feedback. In the camera-ready version of the paper we have made the following revisions:

* Minor correction: During the rebuttal phase, we added Proposition 6 (Section 4) to convey that the mechanism underlying chaos may persist in more general settings. In the rebuttal version, the result was stated for all step sizes $\eta>0$. In the camera-ready version, this statement is corrected to "all $\eta>0$ except for at most finitely many values". The excluded values correspond to degenerate cases in which the GD map $\mathrm{GD}(\theta)=\theta-\eta\nabla L(\theta)$ has a measure-zero image. This refined condition holds with probability one and therefore does not weaken the generality of the result. We also note that Proposition 6 is independent of the main results in Section 3.

* Polishing: We corrected several typos, clarified the proofs of Lemma 12, Propositions 20, 24, and 27, and updated the related-work discussion.

---

### Meta-Review · Area_Chair_zhDf · 2025-12-26

**Summary:**

The reviewers had the following concerns:
1. Limited practical insights. Since the experiments were all synthetic, the reviewers are concerned about that the theoretical insights and the empirical observations might not be able to generalize to more practical settings.
2. Simple model assumption. Since the scalar matrix factorization problem is restrictive the reviewers are concerned about whether the analysis can be extended to general matrix factorization problems and beyond.
3. The technical novelty of the step size bound is unclear, when comparing it with prior works that studied similar problems.

**Reviewer Concerns:**

I think the first concern on practical insights were sufficiently addressed by the rebuttal: authors provided additional experiments with MLPs trained on CIFAR-10 dataset and elaborated the practical implications from their theoretical results. Likewise, the authors have explained the novelty of the step size bound in the rebuttal, the third concern is also addressed.

Although the second concern is still outstanding and the authors have acknowledged the challenges in extending their results to other problems, the restrictive assumption now seems to be not an major issue as new empirical results have shown that similar phenomenon does appear in more general problems.

**Reviewer Scores:**

I personally found the author rebuttal satisfactory, and believe some reviewers would feel the same. I think Reviewer Nc47 is like to maintain their score (6). Reviewer t5Rh and V6Q9 initially had scores of 4s, I think it is likely that at least one of them would increase their score to 6. I think Reviewer F5Q8 would increase their score to 6

---

### Decision · Program_Chairs · 2026-01-26

Accept (Poster)